# Characteristics and process controls of statistical flood moments in Europe – a data based analysis

David Lun[1], Alberto Viglione[2], Miriam Bertola[1], Jürgen Komma[1], Juraj Parajka[1], Peter Valent[1,3], Günter Blöschl[1]

[1]Institute of Hydraulic Engineering and Water Resources Management, Vienna University of Technology, Vienna, Austria

[2]Department of Environment, Land and Infrastructure Engineering, Politecnico di Torino, Turin, Italy

[3]Department of Land and Water Resources Management, Faculty of Civil Engineering, Slovak University of Technology, Bratislava, Slovakia

*Correspondence to*: David Lun (lun@hydro.tuwien.ac.at)

**Abstract**. Many recent studies have sought to characterize variations of the annual maximum flood discharge series over time and across space in Europe, including some that have elucidated different process controls on different statistical properties of these series. To further support these studies, we conduct a pan-European assessment of process controls on key properties of this series, including the mean annual flood (MAF), coefficient of variation (CV) and skewness (CS) of flood discharges. These annual maximum flood discharge series consist of instantaneous peaks and daily means observed in 2370 catchments in Europe without strong human modifications covering the period 1960-2010. We explore how the estimated moments MAF, CV and CS vary due to catchment size, climate and other controls across Europe, where their averages are 0.17 m³ s$^{-1}$ km$^{-2}$, 0.52 and 1.28, respectively.

The results indicate that MAF is largest along the Atlantic coast, the high-rainfall areas of the Mediterranean coast and in mountainous regions, while it is smallest in the sheltered parts of the East European plain. The CV is largest in southern and eastern Europe, while it is smallest in the regions subject to strong Atlantic influence. The pattern of CS is similar, albeit more erratic, in line with the greater sampling variability of CS. In the Mediterranean, MAF, CV and CS decrease strongly with catchment area, suggesting that floods in small catchments are relatively very large, while in Eastern Europe this dependence is much weaker mainly due to more synchronized timing of snow melt over large areas.

The process controls on the flood moments in five predetermined hydroclimatic regions are identified through correlation and multiple linear regression analyses with a range of covariates and the interpretation is aided by a seasonality analysis. Precipitation-related covariates are found to be the main controls of the spatial patterns of MAF in most of Europe except for regions in which snowmelt contributes to MAF, where air temperature is more important. The Aridity Index is, by far, the most important control on the spatial pattern of CV in all of Europe. Overall, the findings suggest that, at the continental scale, climate variables dominate over land surface characteristics, such as land use and soil type, in controlling the spatial patterns of flood moments.

Finally, to provide a performance baseline for more local studies, we assess the estimation accuracy of regional multiple linear regression models for estimating flood moments in ungauged basins.

## 1. Introduction

Understanding the spatial distribution of statistical flood characteristics is important from both practical and scientific perspectives, assisting in estimating design floods in gauged and ungauged catchments, and shedding light on the regional processes of flood generation from a probabilistic perspective (Rosbjerg et al., 2013).

Much research has been conducted on identifying process variables and mechanisms controlling the magnitudes of flood characteristics. Catchment area is usually the main control on the specific mean annual flood (MAF) as smaller basins tend to have larger specific MAFs than larger ones (Rosbjerg et al., 2013) because a large basin is less likely to be fully covered by a storm than a small basin. This tends to reduce the variance of extreme catchment-average precipitation and thus the MAF (Viglione et al., 2010). Additionally, there is an important space-time effect that explains the attenuation of specific floods as catchment areas increase. Event-scale catchment response times tend to increase with area (Gaál et al., 2012), which leads to a greater attenuation of the flood peaks. Convective events, limited in duration and spatial extent, are most relevant for producing floods in small catchments with fast response times (Gaál et al., 2015), whereas long duration stratiform precipitation becomes more relevant as catchment size increases (Merz and Blöschl, 2009). Meanwhile the effect of catchment area on the CV (the ratio of standard deviation and annual means) tends to be more complex. For example, Smith (1992), based on data in the Appalachian region, found an increase in CV with catchment area up to about 100 km², and subsequently a decrease, which he attributed to the spatial organisation of precipitation and downstream changes in the floodplain system. Blöschl and Sivapalan (1997) suggested that space-time scale interactions may be the main reason for this scale dependence, while Merz and Blöschl (2003) noted that the strength of the dependence of CV on area will differ between regions with different prevailing flood generation mechanisms such as floods from synoptic-scale precipitation events (e.g. frontal weather systems) and snowmelt-driven floods. Based on an analysis of flood data in Slovakia, Austria and Italy, Salinas et al. (2014) found both CV and CS to decrease with catchment area which they interpreted as the result of aggregation effects of the spatial heterogeneity of rainfall and the interaction between the spatial and temporal scales of rainfall and catchment size.

Runoff generation and thus flood moments are also controlled by soil characteristics, geology and land-use. While USGS regional flood frequency studies based on observed data have revealed non-climatic controls (Parrett et al., 2011, Paretti et al., 2014, England et al., 2019), most knowledge on these effects comes from process-based simulation studies. For example, Gioia et al. (2012) demonstrated that infiltration and soil storage strongly affect the flood moments, and Brath et al. (2006) performed a similar analysis on land use. The role of these variables can to some extent be inferred from their use as covariates in flood frequency regionalisation models (see, e.g. Zaman et al., 2012; Rosbjerg et al., 2013; Miller and Brewer, 2018) , including nonstationary ones. However, the type of soil, geology and land-use data available at the regional scale is often not consistent with the level of detail required for attributing runoff generation processes, and therefore correlations with flood characteristics tend to be low (Weingartner et al., 2003).

Another important control is climate. Mechanistically, one would expect extreme precipitation over timescales (e.g. 1 day) to represent flood characteristics as it is usually the main driver at the event scale (Viglione et al., 2009). However, many studies have shown that mean annual precipitation (MAP) is a better predictor of MAF (e.g. Madsen et al., 1997; Reed et al., 1999; Merz and Blöschl, 2009) for which a number of reasons have been suggested: MAP is an important control of antecedent soil moisture on a seasonal scale (e.g. Grillakis et al., 2016)

and is usually highly correlated with event precipitation. Moreover climate, vegetation, soils and land forms may co-evolve with MAP, thus exerting a longer-term influence which may increase or decrease floods (Gaál et al., 2012, Perdigão and Blöschl, 2014). Based on data from around the world, Farquharson et al. (1992) found CV of annual peak flows (variability between years) to increase with the Aridity Index (the ratio of potential evaporation and MAP). This dependency may be the result of at least two processes. On the one hand, the lower and more variable runoff coefficients of arid regions tend to increase the flood CV far beyond that of rainfall (Viglione et al., 2009). On the other hand, the CV of rainfall is also sometimes larger in arid regions than in more humid regions (Fatichi et al., 2012). Merz and Blöschl (2009) found potential evaporation to be an excellent predictor of both MAF and CV in the lowlands of Austria (the greater the PET, the higher the CV), which they interpreted in terms of the increasing non-linearity of the rainfall-runoff process with aridity. Iacobellis et al. (2002), found that CV behaviour is controlled mainly by the long-term climate and the infiltration characteristics at the catchment scale.

Climate controls, including rainfall, soil moisture and snowmelt are usually subject to strong seasonality. An analysis of the seasonality of floods (Bayliss and Jones, 1993; Merz and Blöschl, 2003) has therefore been an efficient way to shed light on the interaction of these processes. For example, based on the seasonality of 4262 catchments in Europe, Blöschl et al. (2017) identified extreme winter precipitation in northwestern Europe, snowmelt in northeastern and eastern Europe and summer precipitation and snowmelt in the Alpine regions of Europe to be important flood drivers. Using their data set, Berghuijs et al. (2019) found that soil moisture excess explained flood seasonality better than other hydroclimatic variables, such as extreme precipitation, particularly in the western part of Europe, in line with a similar study in the United States (Berghuijs et al., 2016).

While substantial understanding of regional flood controls has been achieved in the past, few studies have analysed large, consistent data sets of flood discharges in terms of their statistical moments. Large data sets provide the opportunity to obtain more robust and generalisable findings than studies containing a smaller number of catchments. The aim of this paper is therefore to identify patterns of flood moments and their controls across Europe. We use a data set of flood discharges of 2370 catchments across Europe for the period 1960-2010 and apply correlation and regression analyses to identify climatic and catchment process controls on the moments.

## 2. Data and Methodology

### 2.1 Data

This study uses a subset of the data set of European flood discharges of Blöschl, Hall et al. (2019), for which stricter selection criteria were applied than for their entire data set (see their section on datasets and their supplementary material for the data). It consists of 2370 annual maximum discharge series from 33 countries derived from instantaneous peak flows and daily mean flows for each calender year. Catchment areas range from 5 to 100000 km$^2$, with a median of 383 km². The observation period is 1960 to 2010, and record lengths range from 30 to 51 years with a median of 51 years. The time series were manually checked for strong human modifications such as reservoirs (Blöschl, Hall et al., 2019 and Hall et al., 2015) and include both rainfall-generated floods and snowmelt-generated floods (Kemter et al., 2020).

To analyse process controls, a range of catchment attributes and climatic indicators are used. In addition to catchment area (A), we used catchment-averaged climate indicators, including long-term mean annual

precipitation (MAP), long-term mean potential evaporation (PET) and the aridity index (AI), PET/MAP. Extreme precipitation is quantified by the daily rainfall rate that is not exceeded in 95% of the days of the year (P95), and the long-term mean of the maximum 2-day precipitation of each year (Pmax). While the duration of event precipitation to examine varies with catchment size and characteristics due to differences in response times (Gaál et al, 2012) we chose a constant value of two days here for consistency. As a proxy for snowmelt we used the mean air temperature in spring (Tspr) and winter (Twin). Soil moisture (SM) was taken from the CPC Soil moisture database, which contains model-calculated soil moisture values. Fan and Van Den Dool (2004) discuss some biases of the soil moisture data set, which may distort some of the findings here. We used the mean of the annual maximum monthly values over the observation period. Topographical indicators include the mean catchment elevation (Elev) and the mean topographical slope (Slope) of each catchment. Land use was quantified as a percentage of total catchment area and includes forest areas (LUF) and water bodies (LUW). Soil characteristics were quantified in terms of five soil-texture categories (Stex). The data used are summarised in Table 1.

Table 1: Data used in this study including quantiles of the variables and source information. Sources:
[1]Data Base on European Floods: https://github.com/tuwhydro/europe_floods
[2]CCM River and Catchment Database. Vogt et a., 2017. https://data.europa.eu/
[3]E-OBS gridded dataset (v18.0e), 0.1 deg. Cornes at al., 2018. https://www.ecad.eu/
[4]Global Aridity Index and Potential Evapo-Transpiration (ET0) Climate Database V2. Trabucco and Zomer, 2019 https://figshare.com/articles/Global_Aridity_Index_and_Potential_Evapotranspiration_ET0_Climate_Database_v2/7504448/3
[5]CPC Soil Moisture (V2), NOAA Climate Prediction Center. Fan and Dool, 2004. https://www.cpc.ncep.noaa.gov/
[6]Global Multi-resolution Terrain Elevation Data GMTED2010, 7.5 arc-seconds. https://www.usgs.gov/
[7]Corine Land Cover (CLC) 2000, Version 20b2, https://land.copernicus.eu/
[8]European Soil DataBase (ESDB), Soil Geographical DataBase (SGDB), TEXT_SRF_DOM, 10x10km. Panagos et al., 2012. https://esdac.jrc.ec.europa.eu

| Variable group | Symbol | Data Description | Units | 25%-quantile | 50%-quantile | 75%-quantile | Source |
|---|---|---|---|---|---|---|---|
| Flood Moments | MAF | Mean annual specific flood | m³ s⁻¹ km⁻² | 0.06 | 0.11 | 0.22 | Data base on European floods[1] |
| | $MAF_\alpha$ | Mean annual specific flood normalized to catchment area of $\alpha$ =100km² | m³ s⁻¹ km⁻² | 0.08 | 0.16 | 0.28 | Data base on European floods[1] |
| | CV | Coefficient of variation of annual maximum flood peaks | - | 0.36 | 0.46 | 0.61 | Data base on European floods[1] |
| | CS | Coefficient of skewness of annual maximum flood peaks | - | 0.62 | 1.09 | 1.69 | Data base on European floods[1] |
| Catchment Area | A | Catchment area | km² | 135.90 | 382.80 | 1264.80 | CCM River and Catchment Database[2] |
| Precipitation | MAP | Long-term mean annual precipitation | mm yr⁻¹ | 621.28 | 798.69 | 1057.76 | EOBS[3] |
| | P95 | Daily precipitation rate, that is higher than what is observed on 95% of days in the observed period | mm d⁻¹ | 8.40 | 10.54 | 13.45 | EOBS[3] |
| | Pmax | Mean of maximum of 2-day precipitation of each year | mm d⁻¹ | 18.00 | 22.45 | 28.51 | EOBS[3] |
| Air temperature | Tspr | Mean daily temperature in MAM (Celsius) | °C | 5.03 | 7.15 | 8.42 | EOBS[3] |

| | | | | | | | |
|---|---|---|---|---|---|---|---|
| | Twin | Mean daily temperature in DJF (Celsius) | °C | -3.35 | -1.16 | 1.05 | EOBS[3] |
| Soil moisture | SM | Mean of annual maximum monthly soil moisture | mm | 368.38 | 424.93 | 507.20 | CPC Soil Moisture (V2)[5] |
| Evaporation | PET | Long-term mean potential evapotranspiration | mm yr$^{-1}$ | 749.02 | 817.73 | 897.98 | Global Aridity Index and Potential Evapo-Transpiration (ET0) Climate Database V2[4] |
| Aridity | AI | Aridity index (PET/MAP) | unitless | 0.72 | 1.00 | 1.25 | Global Aridity Index and Potential Evapo-Transpiration (ET0) Climate Database V2[4] |
| Topography | Elev | Mean catchment elevations | m a.s.l. | 199.04 | 472.58 | 833.12 | GMTED2010[6] |
| | Slope | Mean topographic slope (mean of tangent of angle of slope) | unitless | 0.03 | 0.08 | 0.18 | GMTED2010[6] |
| Land use | LUF | Fraction of catchment area covered by forest and seminatural areas | % | 31.66 | 54.59 | 79.62 | CORINE[7] |
| | LUW | Fraction of catchment area covered by water bodies | % | 0 | 0.05 | 0.57 | CORINE[7] |
| Soils | Stex | Dominant surface textural class of the STU (Soil Typological Unit), mean value of categories (1=coarse, 5=fine) | class | 1.77 | 2.00 | 2.25 | ESDB[8] |

## 2.2 Hydroclimatic regions

For the statistical analyses, Europe was subdivided into five regions based on the eleven biogeographic regions of Roekaerts (2002) and guided by the flood seasonalities of Blöschl et al. (2017). The aim of the partitioning was to represent a small number of contiguous regions that are to some extent hydro-climatologically homogeneous, without considering their effect on predicting flood moments. In the Northeastern region, floods are mainly due to snowmelt during spring and early summer. The Atlantic region is characterised by mild, wet winters and cool, humid summers; floods mainly occur in winter following rain events. The Central-Eastern region has a continental climate with cold winters and warm summers and floods mainly occur in spring with snow-melt contributions (resulting in a mixture of rainfall and snowmelt). The Alpine region comprises the Alps and the Carpathians, where floods mainly occur in summer due to summer storms and/or snow melt. The Mediterranean region is characterised by hot, dry summers and mild, wet winters; floods occur in autumn and winter. For simplicity, each catchment was allocated to one of the regions according to the location of its stream gauge. The latter is usually representative of the entire catchment, as only for 65 catchments the difference between stream gauge elevation and mean catchment elevation is more than 1000m.

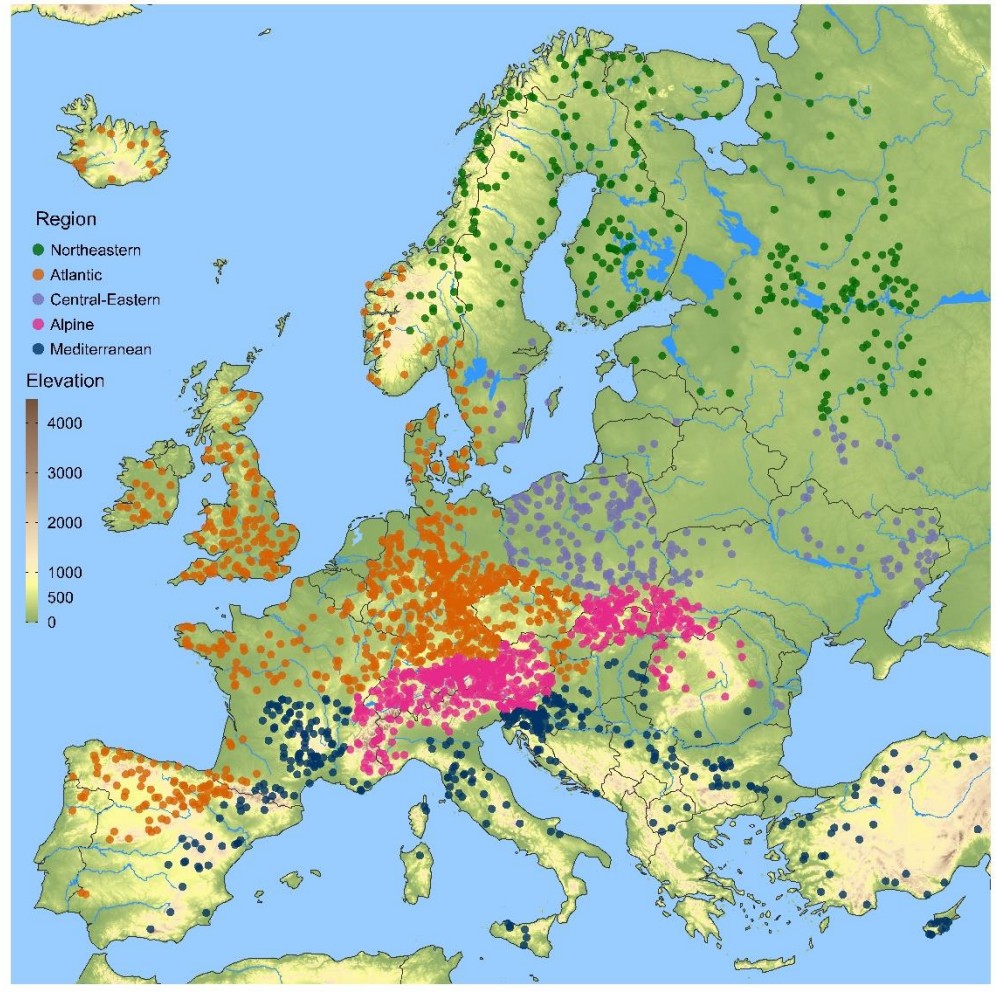

 Figure 1: Location of the 2370 hydrometric stations analyzed. Colours of dots indicate five hydro-climatic regions (Northeastern, Atlantic, Central-Eastern, Alpine, Mediterranean). Background colour is elevation (m a.s.l.).

### 2.3 Analysis method

The statistical flood moments, the specific mean annual flood (MAF), the coefficient of variation (CV) and the coefficient of skewness (CS), were estimated from the annual maximum peak discharges series by:

$$MAF = \frac{1}{n}\sum_{i=1}^{n} Q_i \tag{1}$$

$$S^2 = \frac{1}{n-1}\sum_{i=1}^{n}(Q_i - MAF)^2 \tag{2}$$

$$CV = \frac{S}{MAF} \tag{3}$$

$$CS = \frac{n\sum_{i=1}^{n}(Q_i - MAF)^3}{(n-1)(n-2)S^3} \tag{4}$$

where $Q_i$ is the annual maximum peak discharge ($m^3\,s^{-1}$) of a record in year $i$, divided by the catchment area ($km^2$). While in some cases log-transformed variables are used in flood frequency analysis (Griffis and Stedinger, 2007), here we analyze real space moments of flood series, in line with European practice (e.g. Merz and Blöschl, 2009). Estimation uncertainty of the statistical flood moments decreases with record length and increases with the moment order. While the estimation uncertainty of the mean is small, the uncertainty and bias of the estimators of CV and

CS (equations 3 and 4) can be substantial. Ye et al. (2020) illustrate the uncertainty and bias in the estimation of CV. The bias in the estimation of CV is relatively small for ranges of CV and CS as in this study (using their equation 2: the bias is at most 0.065 in absolute value, in the case of CV ranging from 0.25 to 0.97 and CS ranging from 0.09 to 3.18, which encompasses 90% of the observed values in this study) making it reasonable to use the common product moment estimator of the CV. Meanwhile the standard error and bias of the CS estimate are about 0.56 and 0.22 (based on a simulation study), respectively, for a record length of 50 years and a series with the average estimated moments of the entire dataset (MAF=0.17 m³ s-1 km-2, CV=0.52, CS=1.28), which is about half and one sixth of the underlying population moment (assuming a GEV-distribution as the data-generating process). The bias and uncertainty for the estimation of skewness are well documented in Wallis et al. (1974), Bobee and Robitaile (1975) and Carney (2016), for example. The estimation uncertainties need to be accounted for when interpreting the process controls on the flood moments. Additionally, combining regional with local information can help reduce the estimation uncertainty of statistical moments of flood series, as demonstrated by the weighted skewness approaches of Griffis and Stedinger (2009) and the flood frequency hydrology approach of Viglione et al. (2013), but this is beyond the scope of this paper. In interpreting the results, we do not account for any non-stationarities of the flood moments, as the focus is on the aggregated behaviour during the observation period. Any autocorrelation that may be present will increase the uncertainty of the estimates, although they are usually small in annual flood data, and are therefore rarely considered in flood frequency estimation (Hosking and Wallis, 2005).

Since the specific mean annual flood is often strongly controlled by catchment area, which may mask other controls that vary regionally, we also considered the specific mean annual flood, $MAF_\alpha$, normalised to a catchment area of $\alpha=100\text{km}^2$

$$MAF_\alpha = MAF \cdot A^{-\beta_{MAF}} \cdot \alpha^{\beta_{MAF}} \qquad (5)$$

$MAF_\alpha$ and $\beta_{MAF}$ were found by ordinary least squares regression in the logarithmic space.

Our analysis encompasses the following steps:

1. We estimated what fraction of the spatial variability of the estimated flood moments can be explained by subdividing Europe into five regions (Figure 1), using a simple one-way analysis of variance (ANOVA). This can be interpreted as a simple regression model, where the dependent variables are the estimated flood moments and the only explanatory variables are indicators corresponding to the regional partition. The coefficient of determination of this model corresponds to the fraction of variance explained by the partition over the total variance in estimates of the flood moments.

2. We evaluated the role of catchment area, since it is almost always the main control on the mean annual flood when examining a sample of catchments varying by orders of magnitude, and it reflects the aggregation behaviour of the floods and their climatic and catchment controls. Specifically, we estimated the dependence of MAF, CV and CS on catchment area from equation (5) and analogous equations for CV and CS, transforming all variables logarithmically.

3. We conducted a seasonality analysis to assist in the process interpretations. We represented the date of occurrence, *D*, of the maximum annual flood as a number from 1 to 365 (Julian dates) in polar coordinates

on a unit circle with angle $\theta = D \frac{2\pi}{365}$. For a flood series, the direction $\bar{\theta}$ of the average vector from the origin indicates the mean date of occurrence of the flood events around the year. The length of the vector from the origin k is a measure of the variability of the date of occurrence, ranging from 0 (uniformly distributed across the year) to 1 (all events on the same day). It is calculated as the Euclidean distance between the origin and the mean flood date (mean of the sine and cosine of flood dates in polar coordinates), see e.g. Burn (1997). In the spirit of Blöschl et al. (2017) we used the seasonality of floods to identify dominant flood-generating mechanisms, e.g. spring snowmelt vs winter storms which to some extent explain variations in the flood moments (Merz and Blöschl, 2003).

4. We analysed the effects of individual hydro-climatic controls (see Table 1) on the spatial distribution of the flood moments. To assist in the interpretation, we first evaluated the Spearman rank correlation coefficients between the attributes, followed by an analysis of the Spearman rank correlation coefficients between the flood moments and the attributes. We used Spearman's rank correlation coefficients, as non-linear associations between variables are possible. The corresponding significance tests for the estimates of Spearman's rho are employed with the assumption of an asymptotic t-distribution under the null hypotheses (Gibbons and Chakraborti, 2010). The correlations were evaluated for all of Europe and the five regions separately.

5. We evaluated the effect of multiple controls on the flood moments. Multiple linear regression models were fit to the estimated moments. Since the emphasis was on obtaining parsimonious models, the number of selected explanatory variables for a regression equation was limited to four. Both predictor and response variables were log-transformed when they had skewed distributions. MAF, CV, A and P95 were log-transformed, as their distributions were skewed. We use the logarithm of the real-space moments for the regression models. The attributes were selected using a stepwise selection procedure (Weisberg, 2005). The criterion for model comparison was Mallow's Cp

$$C_p = \frac{RSS_p}{\hat{\sigma}^2} + 2p - n \qquad (6)$$

where $p$ refers to the number of coefficients in the current model including the intercept and $n$ to the number of observations. $RSS_p$ is the residual sum of squares of the model being considered with $p$ covariates and $\hat{\sigma}^2$ is the residual error variance of the model including all possible covariates. For the comparisons of information criteria such as $C_p$, a complete set of observations of explanatory variables is required, therefore 22 catchments, where some observations of covariates were not available, were excluded from this part of the analysis (see Table S1 in the supplementary materials).

In order to assess which of the covariates in each regression provided the most explanatory power, a dominance analysis was conducted (Azen and Budescu, 2003). We used the measure for general dominance, which summarizes the average increase in the measure for the goodness of fit, when a given covariate is included in a regression model, for all possible model subsets for a fixed set of predictors. The sum of all measures for general dominance of each variable results in the $R^2$ of the full regression. The general dominance measure (average contributions) provides a ranking of the variables in terms of their contributions to the fit of the models ($R^2$). However, this is only valid in the context of the model and the selected variables. These contributions can and most likely will change when variables are added

245    to or subtracted from the regressions. To facilitate the interpretation of results, in section 3.5 we present the general dominance measure of individual covariates, divided by the $R^2$ of the regression, and refer to this as the normalised general dominance measure. This does not affect the ranking of the variables.

6. The predictive accuracy of the fitted regression models was assessed in a leave-one-out cross-validation. The errors are evaluated on the scale of the data, instead of the scale of the regression models (i.e. log
250    scale) and normalized, e.g. in the case of the MAF:

$$ANE_{MAF} = \left| \frac{\widehat{MAF} - MAF}{MAF} \right| \tag{7}$$

ANE stands for absolute normalized error. $\widehat{MAF}$ are the predictions of the regression models and MAF are the at-site estimates. The ANE for CV are computed in the same way.

In addition, ordinary kriging was used for interpolating the at-site estimates of flood moments (Cressie,
255    1993) and the interpolated values were contrasted with the predictions of regional regression models.

## 3. Results

### 3.1 Characteristics of flood moments

Table 2 shows the characteristics of the estimated flood moments for the five hydroclimatic regions and the entire
260 data set. On average over the entire data set, the mean specific annual flood is 0.17 m³ s⁻¹ km⁻², while the mean specific annual flood normalised to a catchment area of 100 km² ($MAF_\alpha$) is 0.21 m³ s⁻¹ km⁻². The latter is somewhat larger, as 100 km² is smaller than the median catchment size of 383 km². On average over the entire data set, the CV and CS are 0.52 and 1.28, respectively. The regions differ in terms of the moments. The largest average $MAF_\alpha$ occurs in the Alpine region and in the Mediterranean (0.30 m³ s⁻¹ km⁻²). The smallest average $MAF_\alpha$ (0.05 m³ s⁻¹
265 km⁻²) occurs in the Central Eastern region, but this is also the region where the average CV and CS are largest (0.69 and 1.59, respectively). On the other hand, the Northeastern region has the smallest average CV and CS and below average MAF (0.39, 0.82 and 0.13, respectively). The regional coefficients of variation in Table 2 (every other column) reflect the within-region variability of the observed flood moments. They are generally higher for MAF and $MAF_\alpha$ than for CV, both within individual regions and for all of Europe.

270 The coefficient of determination $R^2$ of the one-way ANOVA is an indicator of how much of the spatial variability is explained by the partitioning into the five regions. The partitioning explains 17% of the spatial variance of $MAF_\alpha$, but only 9% and 5% of the spatial variance of CV, CS, respectively. This means that the spatial variability of CV and CS within each region is almost as great as the spatial variability over all of Europe. While some of that variability may arise from sampling uncertainty, but Figure 2 indicates that, at least for $MAF_\alpha$ and CV, there are
275 clear spatial patterns that are not aligned with the regions, i.e. much of the variability lies within the regions and not between them. As the regions have not been chosen to reflect flood magnitudes but rather a range of hydro-climatic processes, this is not surprising.

The largest $MAF_\alpha$ occur in wet, mountainous regions, including the Alps, the Appenines (Italy), the Carpathian Flysch Belt, adjacent to the Ligurian and Adriatic Seas, the Languedoc, and the western coasts of Great Britain
280 and Norway. The lowest $MAF_\alpha$ occurs in much flatter regions, such as the North and East European Plains, Dnieper

Lowland, Finnish Lakeland and southern Sweden and south-eastern England (Fig. 2c). $MAF_\alpha$ exhibits slightly more homogeneous spatial patterns across Europe than MAF, as the effect of catchment size has been removed. The spatial distribution of CV is, to some degree, a mirror image of that of $MAF_\alpha$, as CV and $MAF_\alpha$ are slightly negatively correlated with a Spearman correlation coefficient of r=-0.12 (-0.06 for CV vs MAF). However, there

are deviations from this general pattern. Along the Mediterranean coast between Genoa, Italy and Valencia. CVs are rather large even though $MAF_\alpha$ are large as well, partly because of flashy mountainous catchments with high rainfall. The spatial coherence of CS is less apparent (which can also be seen from the low $R^2$ in Table 2), as roughly half the spatial variability is likely attributable to sampling variability (the standard error of the CS estimate for average parameters is $\sigma_{CS} = 0.56$). However, there seems to exist a general pattern of larger than average CS

along the Mediterranean coast and the mountainous areas of Europe, and smaller than average CS in Scandinavia and northern Russia. There is a strong positive Spearman correlation between CS and CV (r=0.63) which points towards non-linear runoff generation processes affecting both CS and CV (Rogger et al., 2013), although, again the correlation may be partly due to the sampling variability resulting in correlations between the estimators of CV and CS (see chapter 10 in Kendall and Stuart, 1969).

Table 2: Regional mean (m) and regional coefficient of variation (cv) of the mean annual specific flood (MAF, (m³ s$^{-1}$ km$^{-2}$)), mean annual specific flood normalised to catchment area of 100km$^2$ ($MAF_\alpha$, (m³ s$^{-1}$ km$^{-2}$)), coefficient of variation (CV) and coefficient of skewness (CS) for the entire data set and the five regions. n is the number of stations per region. $R^2$ is the fraction of the spatial variance explained by the partitioning into the five regions.

| | Europe (n=2370) | | Northeastern (n=288) | | Atlantic (n=875) | | Central-Eastern (n=236) | | Alpine (n=622) | | Mediterrane-an (n=349) | | $R^2$ |
|---|---|---|---|---|---|---|---|---|---|---|---|---|---|
| | m | cv | m | cv | m | cv | m | cv | m | cv | m | cv | |
| MAF | 0.171 | 1.114 | 0.126 | 0.939 | 0.137 | 0.981 | 0.037 | 1.095 | 0.264 | 0.816 | 0.222 | 1.223 | 0.140 |
| $MAF_\alpha$ | 0.211 | 0.961 | 0.176 | 0.655 | 0.166 | 0.875 | 0.047 | 0.942 | 0.304 | 0.728 | 0.304 | 0.938 | 0.173 |
| CV | 0.518 | 0.492 | 0.386 | 0.347 | 0.494 | 0.435 | 0.695 | 0.583 | 0.516 | 0.381 | 0.571 | 0.529 | 0.090 |
| CS | 1.278 | 0.773 | 0.821 | 0.779 | 1.216 | 0.882 | 1.59 | 0.727 | 1.466 | 0.565 | 1.263 | 0.785 | 0.047 |

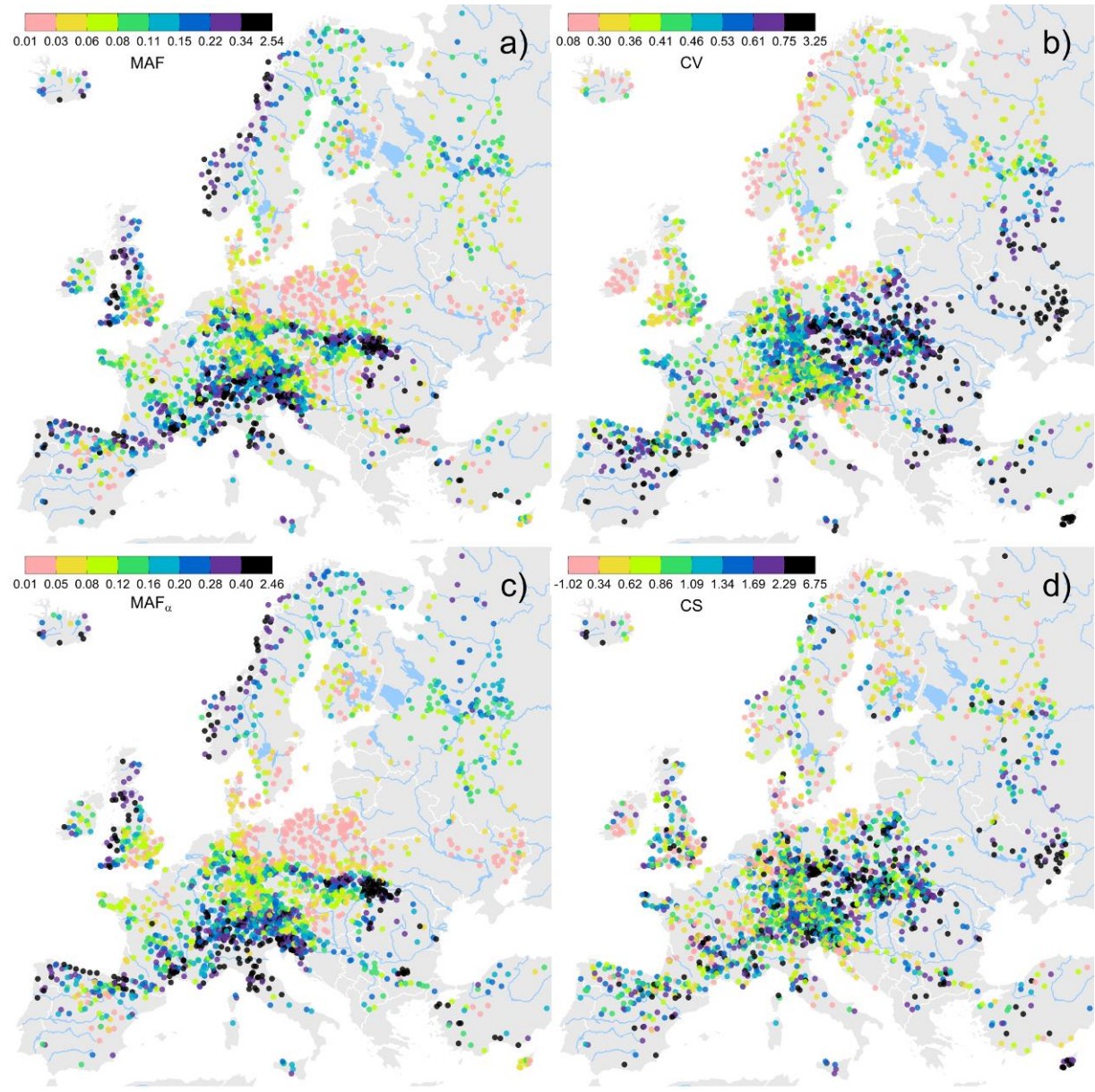

Figure 2: Mean specific flood (MAF (m³ s⁻¹ km⁻²)) (a), Coefficient of variation (CV) (b), Mean specific flood normalised to a catchment area of 100km² (MAF$_\alpha$ (m³ s⁻¹ km⁻²)) (c), and Coefficient of skewness (CS) (d). Colours represent the estimate partitioned into eight classes of equal frequency.

### 3.2 Seasonality and flood moments

As an indicator of flood processes, the average direction of seasonality $\bar{\theta}$ and the strength of seasonality $k$ are plotted in Figure 3 for each catchment (see Figure 3 in Blöschl et al., 2017, for the spatial distribution). The closer the points are to the edge of the circle, the stronger the seasonality. Additionally, the magnitudes of MAF$_\alpha$, CV and CS are indicated as colours as in Figure 3. The red circles in Figure 3 highlight geographic regions that are roughly homogeneous with respect to seasonality and the magnitude of the estimated flood moments, which were identified by examining seasonality maps (Blöschl et al., 2017).

In Northeastern Europe floods mainly occur in spring and early summer with a strong seasonality reflecting the role of snowmelt. MAF$_\alpha$ are in a medium range (green in Fig. 3) with the exception of the Norwegian coast (circle

1) where $MAF_\alpha$ are much larger but with little seasonality because there is a mix of spring snow melt floods and winter rain floods (Kemter et al., 2020). The CVs of these catchments are small. On the other hand, floods in western Russia almost always occur in April with large CV (circle 3). The June floods further in the North, adjacent to the White Sea, have much smaller CV because of a more consistent snow melt influence (Kemter et al., 2020, circle 2).

In the northwestern part of the Atlantic region (Ireland, west coast of the UK), floods tend to occur in December and the estimates of $MAF_\alpha$ are large (circle 4). East of the Atlantic region (Southern Germany, Czech Republic, circle 5) the average occurrence of floods is in late spring, although the seasonality is weak but CVs are large.

In the Central-Eastern region (Poland, Ukraine, circle 6) where floods usually occur in spring, the $MAF_\alpha$ are generally low and the CVs are large.

The catchments in the Alpine region with summer floods (circle 9) show high $MAF_\alpha$ due to rainfall enhancement of the Alps. The catchments with autumn floods in Southern Austria and Slovenia (circle 7) exhibit very high $MAF_\alpha$ due to the stronger influence of the Mediterranean Sea. This region also contains the Carpathians and adjacent midlands (circle 8) where floods tend to occur in spring with lower MAF but rather high CV, likely because of a mix of snow, rain-on-snow and rainfall floods.

The Mediterranean region contains catchments with mostly winter floods with rather high $MAF_\alpha$. The autumn flood catchments in Slovenia have particularly high $MAF_\alpha$ (circle 10) and low CV while the catchments with spring floods on the Balkans (circle 11) possess medium $MAF_\alpha$ and CV.

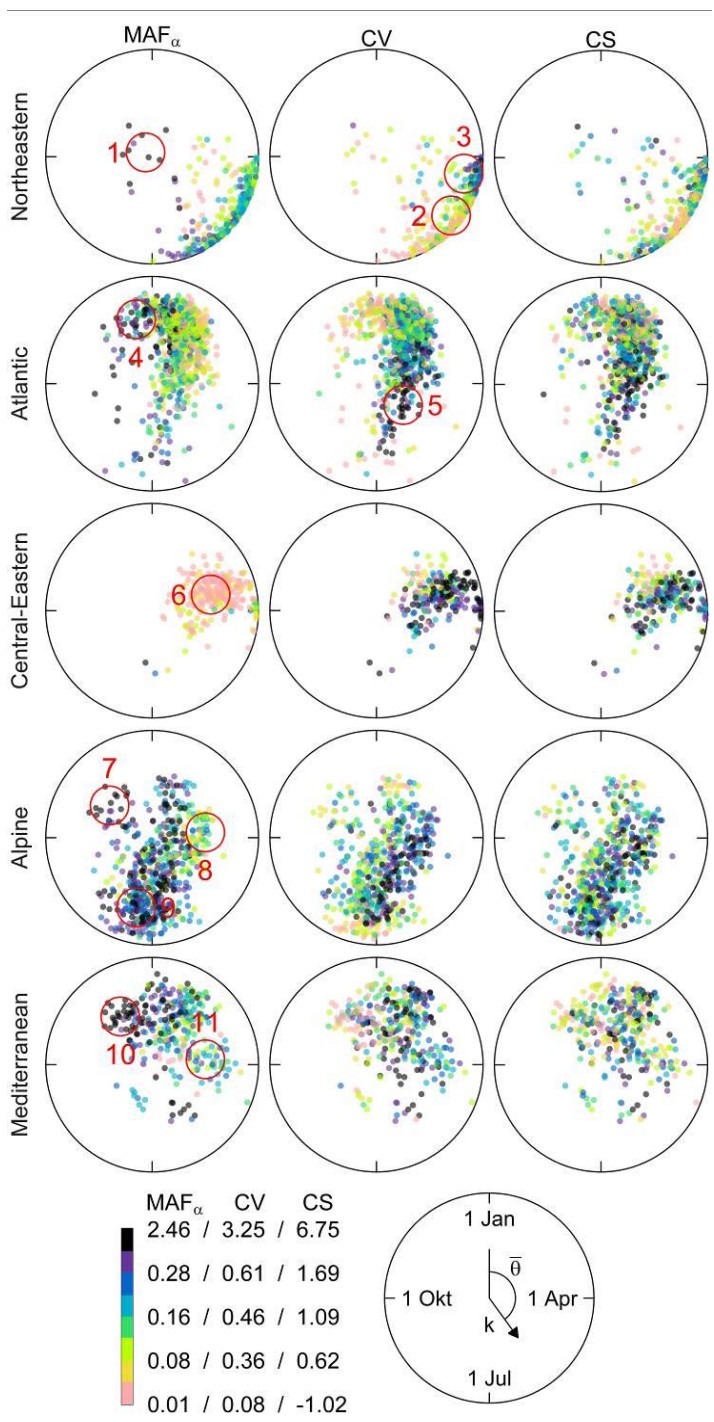

Figure 3: Seasonality of annual flood peaks. Position in circle indicates mean date of occurrence (angle) and variability of the date (inverse of distance from centre). Each point represents one catchment. Colours of the points indicate flood moments as in Figure 2. Small red circles highlight subregions referred to in the text (1: Norwegian coast, 2: Northwestern Russia, 3: Western Russia, 4: Western UK and Southwestern Norway, 5: Southern Germany and North of Czech Republic, 6: Parts of Poland and Ukraine, 7: Southern Austria and Northern Slovenia, 8: Alpine and Carpathian midlands, 9: Alpine region, 10: Slovenia and Southern France, 11: Balkans)

### 3.3 Scaling of flood moments with catchment area

The first control on the flood moments examined here is catchment area, as it is often the dominant and best understood control (Table 3, Fig. 4). The highest decrease in the mean annual flood (MAF) with catchment area occurs in the Mediterranean and the Alpine region with $\beta_{MAF}$ =-0.255 and -0.208, respectively, while the smallest decreases occur in the Northeastern and Central Eastern regions ($\beta_{MAF}$ =-0.163 and -0.108) where snow melt is important. The coefficient of variation (CV) of the flood records decreases with catchment area in most regions. Again, the strongest decrease occurs in the Mediterranean while in the Northeastern and Central Eastern regions there is no significant relationship. There are few small catchments in the Central-Eastern region, which may make the regression with area less robust. Overall, there is a tendency for CS to decrease with catchment area and the strongest decrease occurs again in the Mediterranean.

Table 3: Dependence of the flood moments with catchment area in a double logarithmic relationship Eq. (5) and analogous equations for CV; and a semi logarithmic relationship for CS, i.e. $CS = \log A \; \beta_{CS}$. Last lines show the 5% and 95% quantiles of catchment area (km²). * indicates statistical significance (two-sided t-test) at the 5% level.

|  | Europe | Northeastern | Atlantic | Central-Eastern | Alpine | Mediterranean |
|---|---|---|---|---|---|---|
| $\beta_{MAF}$ | -0.245* | -0.163* | -0.184* | -0.108* | -0.208* | -0.255* |
| $\beta_{CV}$ | -0.030* | -0.015 | -0.042* | 0.025 | -0.020* | -0.072* |
| $\beta_{CS}$ | -0.133* | -0.054 | -0.124* | 0.280* | -0.177* | -0.232* |
| 5% / 95% quantiles of area (km²) | 35/11500 | 28/18010 | 37/5331 | 142/32509 | 26/4212 | 47/27251 |

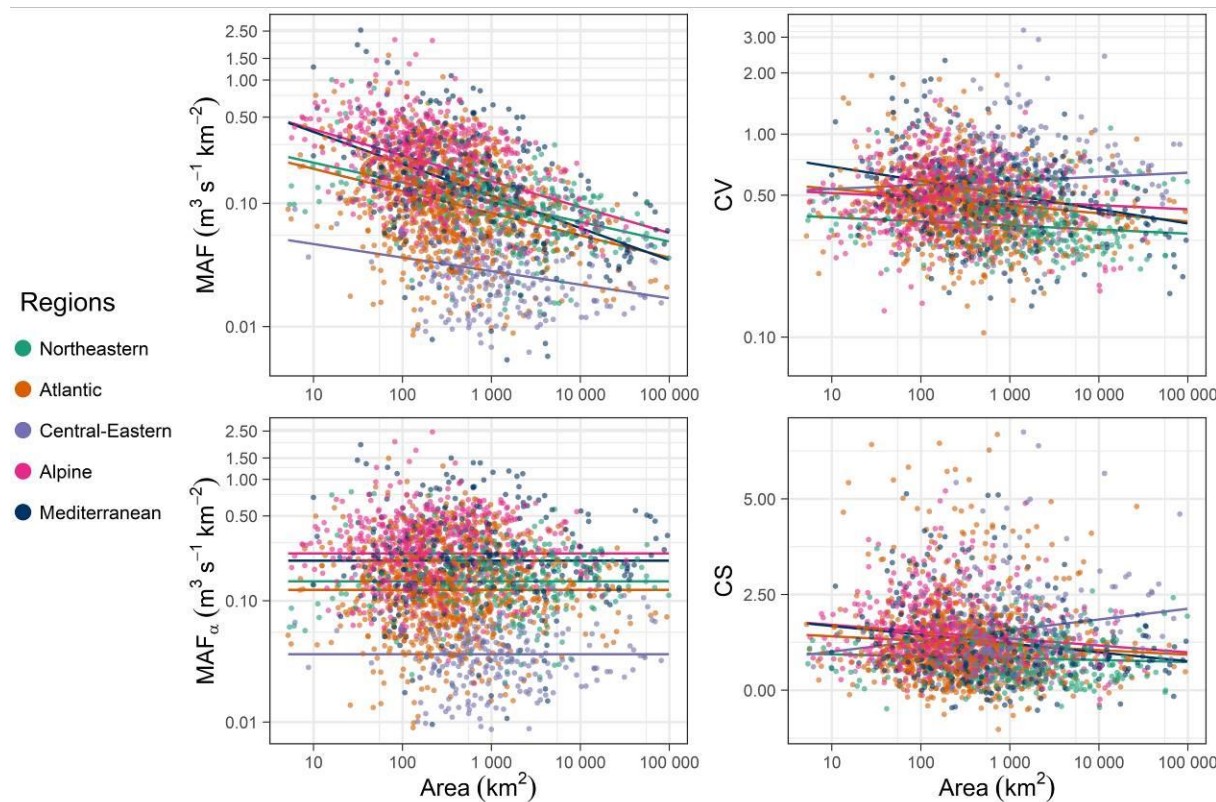

Figure 4: Mean annual specific flood (MAF), normalised mean annual specific flood (MAFₐ), coefficient of variation (CV) and coefficient of skewness (CS) plotted against catchment area (km²). Colours indicate region. Lines are regression lines for each of the regions.

### 3.4 Individual controls on flood moments

When interpreting the association of climate and catchment attributes with flood characteristics, it is important to account for the correlation between the attributes themselves, which may mask causal relationships. Spearman correlation coefficients have therefore been estimated among all explanatory variables (Figure 5). The largest correlations occur among the precipitation characteristics, all of which are at least $r$=0.86. The correlation between long-term mean precipitation (MAP) and daily precipitation not exceeded 95% of the time (P95) is 0.96, indicating that the spatial patterns of these two variables in Europe are almost identical. The correlation between soil moisture (SM) and the precipitation variables is at least 0.78, and the correlation between the aridity index (AI) and the precipitation variables varies between -0.63 and -0.84. The latter may be partly related to the fact that AI is the ratio of potential evaporation (PET) and MAP. PET is related to spring temperature ($r$=0.73). Elevation and slope are closely related to each other ($r$=0.88) and they are also closely related to the precipitation variables with $r$ of at least 0.56, reflecting orographic influences on precipitation. Forest cover (LUF) is related to elevation and slope ($r$= 0.67 and 0.72, respectively) reflecting the presence of forest in mountain areas. The positive correlation between lake area fraction (LUW) and catchment area ($r$=0.45) results from a tendency for large catchments to contain lowlands where lakes are more frequent than in the mountains, and the positive correlation between soil type (Stex) and spring temperature (Tspr) ($r$=0.37) is due to coarse soils prevailing in the (colder) north of Europe. Fig. 5 also shows 2d histograms of the variables as well as their kernel density estimates.

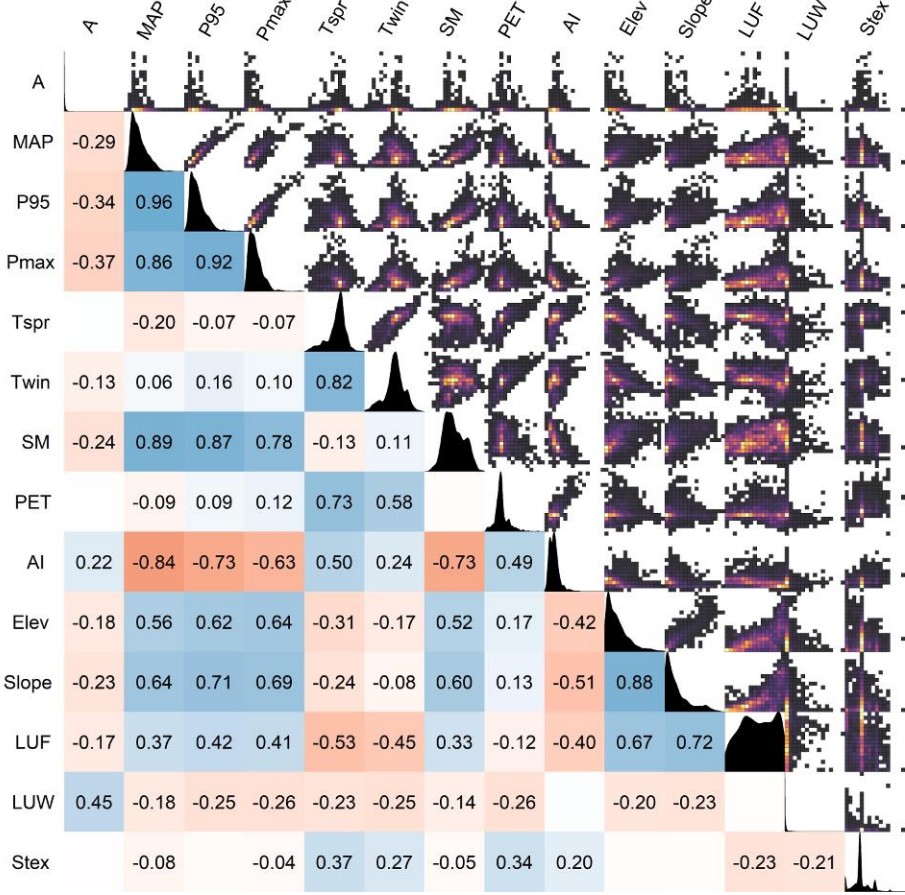

Figure 5: Correlation between explanatory variables as in Table 1. For each variable the data consist of n=2370 values, i.e. the number of catchments. Lower triangle: Spearman correlation coefficients. They are shown if they are statistically significant (α=0.05). Blue and red indicate positive and negative relationships, respectively. Upper triangle: 2d-histograms with colours indicating the frequency of observations in the bins (dark: few; bright: many, separate scale for each panel). Diagonal: kernel density estimate. All scales are linear.

Figure 6 gives the Spearman correlations of the flood moments $MAF_\alpha$ and CV with catchment attributes. We mainly examine $MAF_\alpha$ instead of MAF to minimise the effect of spatial differences in the catchment area on the correlations with the flood moments that may mask the direct effects of other variables. The correlations for CS are often weaker and more difficult to interpret, at least partly due to estimation uncertainty. The corresponding correlations for MAF and CS with catchment attributes are given in Table A.1.5 in the Appendix.

While the correlations between $MAF_\alpha$ and catchment area are inherently small in all regions (Figure 6), the correlations between MAF and catchment area (Table A.1.5, appendix) are significant in all regions ranging between 0.31 and 0.48, with the exception of central Eastern where it is only 0.23, which may be related to the

more important role of snowmelt there, given that snowmelt floods tend to occur at the same time over large areas, so one would expect a smaller reduction of flood peaks due to spatial averaging than for rain floods. Overall, in Europe, the relative large explanatory power of catchment area points to an important role of scale effects, although, in most regions, it is not the variable with the largest correlation.

$MAF_\alpha$ is significantly positively correlated with MAP in all regions (Figure 6) and its correlations with P95 and

Pmax are similar. These high correlations may reflect the effect of not only event rainfalls, but also soil moisture

as well as landscape evolution (Perdigão and Blöschl, 2014), as all rainfall variables as well as soil moisture are highly inter-correlated (Figure 5). The correlations between $MAF_\alpha$ and precipitation (both MAP and P95) are largest in the Atlantic region, reflecting the dominant role of precipitation in explaining the spatial variability of $MAF_\alpha$ in this part of Europe. CV is significantly negatively correlated with MAP, P95 and Pmax, for almost all regions. The strongest relationship is observed in the Alpine and Mediterranean region. In drier catchments, the occurrence of floods is more irregular (large CV, e.g. in Spain) as opposed to wetter catchments where every year rather large floods occur (small CV e.g. in Norway).

As expected, the strongest correlations between flood moments and temperature are in the regions with important snow melt contributions (Northeastern, Central-Eastern regions) (Figure 6). Spring (March-May) and winter (December-February) temperature are negatively correlated with $MAF_\alpha$ and positively correlated with CV.

The correlations between the flood moments and mean annual maximum monthly soil moisture (SM) are very similar to those with MAP as the two covariates are correlated with r=0.89. There are high positive correlations of $MAF_\alpha$, especially in the Atlantic and the Mediterranean, but strongly negative correlations between CV and soil moisture. On the other hand, there is a small negative correlation between $MAF_\alpha$ and the mean annual potential evapotranspiration (PET) in all regions, as a higher PET generally reduces the antecedent wetness conditions of floods and hence the flood discharges. More striking are the CVs that are strongly positively correlated with PET in most regions with r ranging between 0.1 and 0.7. Since the aridity index (AI), defined as the ratio of PET and MAP, it has correlations ranging around those of PET and/or the inverse of MAP. The highest values of AI are observed in the Mediterranean region, where strong negative correlations are observed for all flood moments.

Mean catchment elevation (Elev) and mean topographic slope (Slope) are highly correlated with each other (r= 0.88) and therefore have similar correlations with the flood moments in most regions. $MAF_\alpha$ is highly positively correlated with both elevation and slope across all regions of Europe just as it is with the rainfall variables, which points to an indirect effect of topography on mean floods through precipitation. This is consistent with high correlations of elevation and slope to the precipitation variables (around 0.6 and 0.7, respectively, Fig. 5).

$MAF_\alpha$ is positively correlated with the fraction of area covered by forest (LUF) in all regions and negatively correlated with the fraction of area covered by water bodies, i.e. lakes and reservoirs, (LUW) in most regions, including in the region with the largest fraction of water bodies (the Northeastern region where the median LUW is 5.7%). The positive correlation between $MAF_\alpha$ and LUF is most likely due to an indirect relationship, as areas with high forest cover tend to be high-elevation regions such as mountains, where precipitation is augmented through orographic effects, implying that the positive correlation cannot be interpreted as deforestation reducing floods.

While there is little correlation between $MAF_\alpha$ and soil texture, there is a clear effect of soil texture on CV with finer soils (Stex=5) being associated with higher CVs than coarse soils (Stex=1).

| | MAF$_\alpha$ | | | | | | CV | | | | | |
|---|---|---|---|---|---|---|---|---|---|---|---|---|
| | Europe | Northeastern | Atlantic | Central – Eastern | Alpine | Mediterranean | Europe | Northeastern | Atlantic | Central – Eastern | Alpine | Mediterranean |
| A | -0.12 | | | | | | -0.13 | -0.16 | -0.19 | | | -0.25 |
| MAP | 0.59 | 0.15 | 0.67 | 0.24 | 0.26 | 0.48 | -0.33 | -0.23 | -0.33 | -0.49 | -0.61 | -0.68 |
| P95 | 0.60 | | 0.66 | 0.31 | 0.24 | 0.52 | -0.22 | | -0.22 | -0.19 | -0.58 | -0.59 |
| Pmax | 0.55 | | 0.52 | 0.22 | 0.26 | 0.49 | -0.05 | | 0.14 | -0.18 | -0.40 | -0.36 |
| TSpr | -0.22 | -0.35 | -0.29 | | -0.14 | -0.12 | 0.26 | 0.58 | | 0.50 | 0.38 | 0.17 |
| TWin | -0.07 | -0.20 | | -0.33 | -0.10 | | 0.04 | 0.20 | -0.23 | -0.52 | 0.10 | 0.27 |
| SM | 0.57 | 0.16 | 0.55 | 0.28 | 0.22 | 0.50 | -0.31 | -0.14 | -0.29 | -0.40 | -0.56 | -0.64 |
| PET | | -0.33 | -0.22 | | -0.11 | -0.21 | 0.39 | 0.60 | 0.34 | 0.68 | 0.11 | 0.63 |
| AI | -0.53 | -0.36 | -0.53 | | -0.31 | -0.43 | 0.46 | 0.62 | 0.38 | 0.63 | 0.50 | 0.69 |
| Elev | 0.55 | 0.47 | 0.44 | 0.50 | 0.16 | | 0.08 | -0.31 | 0.35 | 0.21 | -0.47 | 0.20 |
| Slope | 0.65 | 0.39 | 0.62 | 0.30 | 0.24 | 0.37 | | -0.36 | 0.17 | | -0.40 | |
| LUF | 0.45 | 0.28 | 0.45 | | 0.16 | 0.32 | -0.10 | | 0.14 | -0.39 | -0.26 | |
| LUW | -0.16 | -0.21 | | -0.55 | | | -0.27 | -0.22 | -0.20 | -0.50 | -0.19 | -0.19 |
| Stex | -0.06 | | | 0.32 | | -0.18 | 0.29 | 0.37 | 0.17 | 0.53 | 0.23 | 0.26 |

Figure 6: Spearman-Correlation between statistical moments of flood series (mean specific discharge normalized to a catchment area MAF$_\alpha$ of $\alpha$=100km$^2$ and the coefficient of variation CV) and catchment attributes. Correlations are shown if they are statistically significant ($\alpha$=0.05). Blue and red indicate positive and negative relationships, respectively. The correlations for mean specific discharge MAF and coefficient of skewness CS with catchment attributes are given in Table A.1.5.

### 3.5 Multiple controls on the flood moments

While Figure 6 examines the relationships between flood moments and single covariates using Spearman correlation coefficients, in this section we test the relationship between flood moments and multiple covariates with regression models applied for each of the regions separately. Thus the covariates with the largest contributions are those that explain most of the $R^2$ of the spatial variability of the flood moments within each of the regions. We used MAF, rather than $MAF_\alpha$, in order to avoid prior assumptions regarding the role of catchment area. Given that CS was correlated with fewer covariates than the other flood moments we focused here on MAF and CV (Table A.1.5). We represent each of four most important groups of covariates (area, precipitation, temperatures and water balance (i.e. SM, PET and AI)) by one covariate. The contributions to explaining the spatial variability of MAF in terms of the normalised general dominance measure (nGDM) are shown in Figure 7 and Table A.1.3 and are discussed below by region.

In Northeastern Europe, like in all regions, catchment area is an important predictor of MAF. Additionally, P95 and AI play an important role representing an East-West gradient with the largest MAF, largest P95 and the lowest AI in Norway.

In the Atlantic region, P95 is by far the most relevant covariate (nGDM= 0.8) as one would expect in a region where floods are rainfall driven and soil moisture tends to be high in the flood season (winter) (Blöschl, Hall et al., 2019; Kemter et al., 2020).

In the Central-Eastern region MAF is generally low with little spatial variability. The corresponding $R^2$ is small (0.33, Table A.1.1), as it is harder to explain the small spatial contrast. The largest contribution is winter temperature and P95. Negative coefficients for winter temperature suggest that lower temperatures driving deeper snow packs and, in turn, higher floods.

In the Alpine region area is most important and in fact relatively more important than in any other region. However, the $R^2$ of the model in the Alpine region is low (0.27), which may be a reflection of the hydrological heterogeneity of the area, involving snow melt, rain-on-snow and rain driven floods (Merz and Blöschl, 2003).

In the Mediterranean catchment area is important (nGDM= 0.43), which is due to the small coastal catchments exhibiting much larger specific floods than the larger catchments that extend further in-land, e.g. in Catalonia (Spain), Liguria (Italy), and Slovenia (e.g. Gaume et al., 2009). Additionally, P95 plays a relevant role.

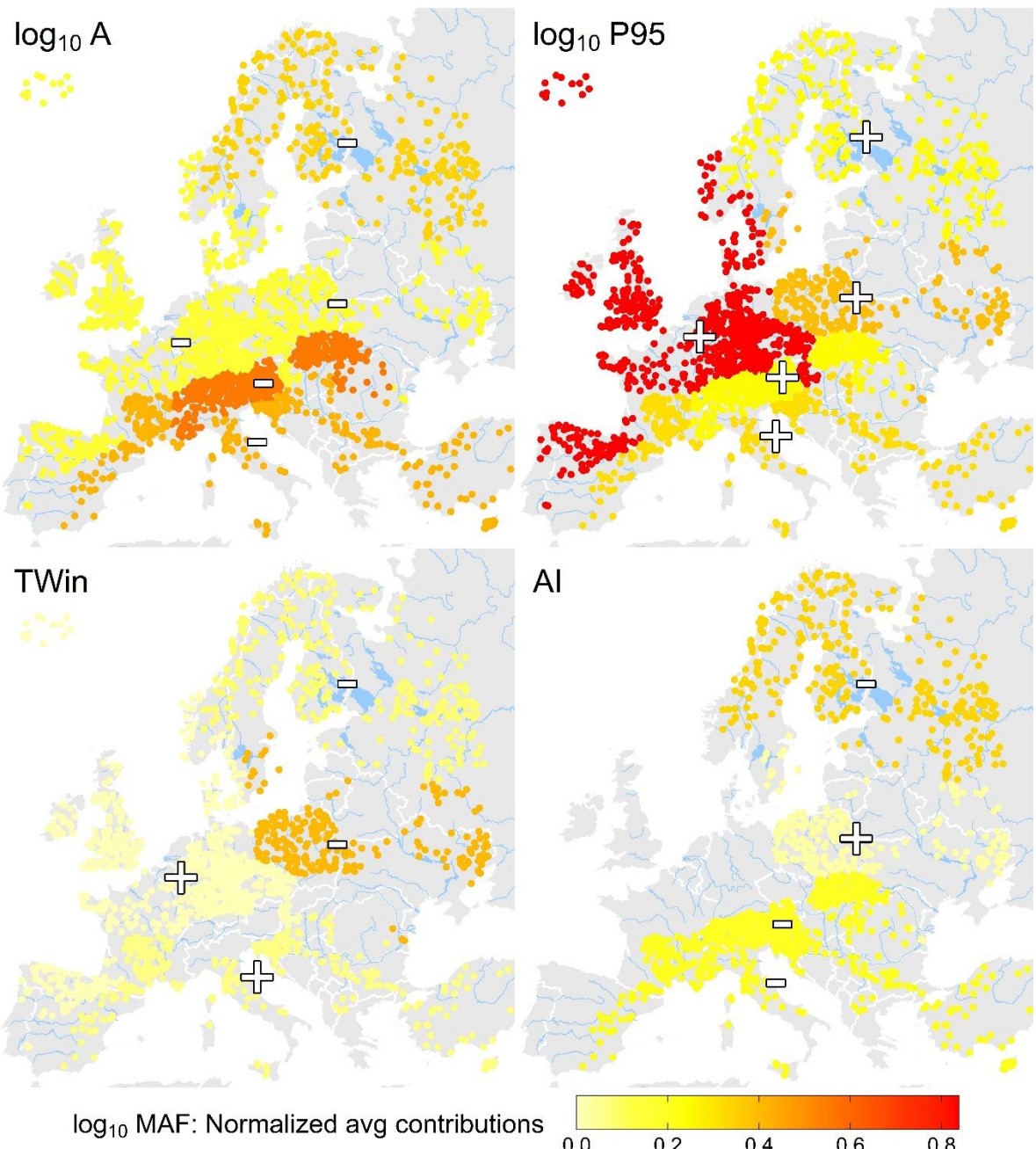

Figure 7: Results of dominance analysis for regional regression models for MAF. Panels depict the average contributions (normalised general dominance measure, nGDM) of the covariates included in the regressions (log of catchment area, log of extreme precipitation index P95, mean winter temperature and aridity index). Plus- and Minus-signs indicate sign of the regression coefficients.

For CV (Figure 8, Table A.1.4), in the Northeastern region the Aridity index (AI) is the most important covariate by far. This is due to Scandinavia being much wetter than northwestern Russia translating into lower CVs.

In the Atlantic region AI is the most important variable for explaining the spatial variability of CV although the overall explanatory power of the regional model is rather low (R² of 0.27, Table A.1.2). The smaller CVs closer to the ocean in the Atlantic region are partly aligned with higher winter temperatures.

In the Central-Eastern region AI dominates again with higher CV in the Ukraine correlated with higher aridity than further in the West, both due to higher PET and lower MAP.

The Alpine region is an exception in that P95 explains more of the spatial variability of CV than AI (nGDM of 0.54 and 0.35, respectively). This is because PET is negatively related to elevation but the flood magnitudes are controlled by the higher orographic rainfall on the windward (NW) side of the Alps.

In the Mediterranean both aridity and P95 are important predictors. For example, low aridity (because of high MAP) in Croatia and Slovenia is associated with low CV, and high P95 in Southern France is associated with moderately low CV.

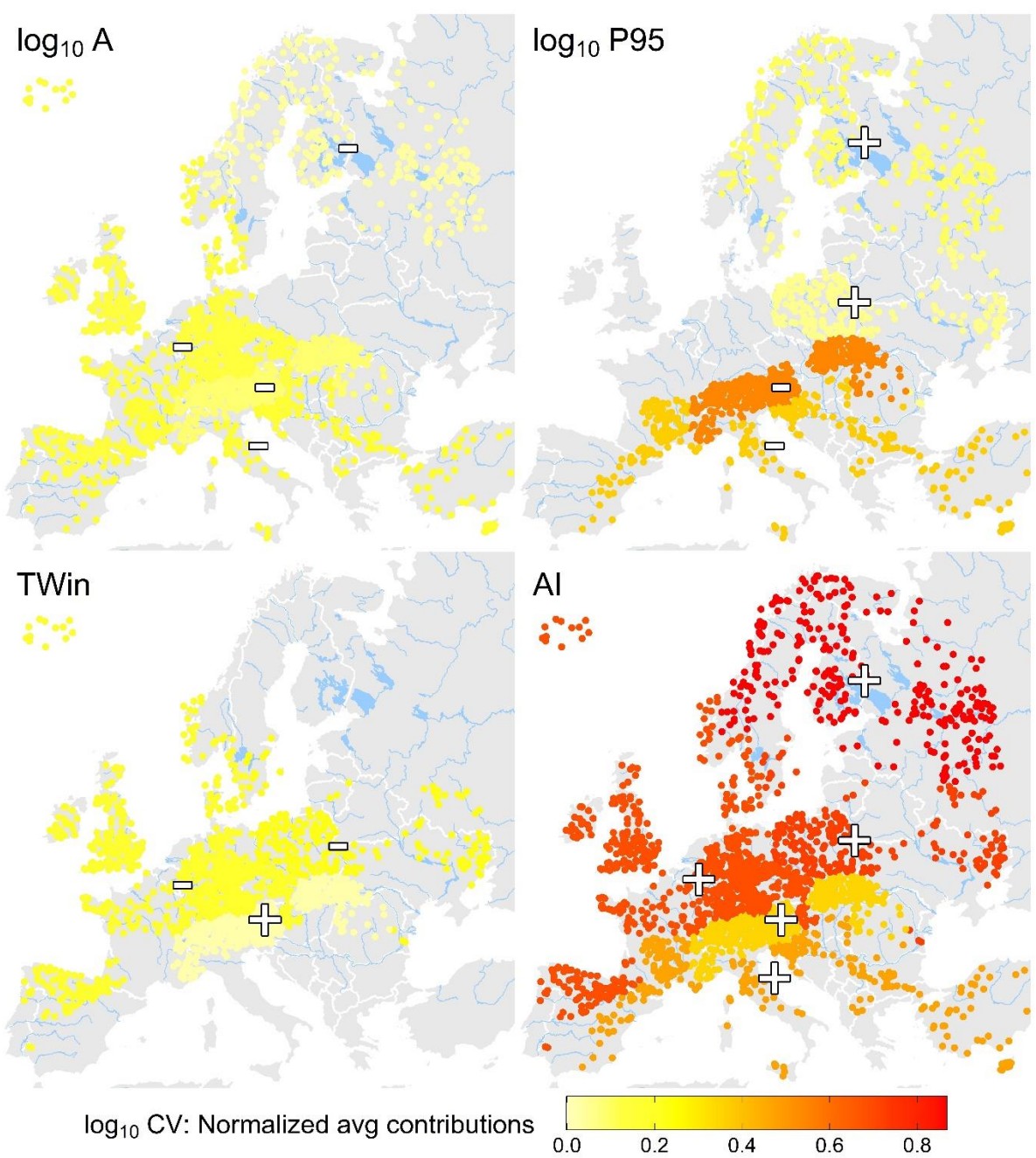

### 3.6 Estimating flood moments from multiple controls

In this section we analyse how well the regression models of the previous section (where A, P95, TWin and AI were used as covariates) are able to predict the moments at any location using a leave-one-out-cross-validation (Figures 9 and 10). Overall, there is a tendency for MAF to be overestimated in those areas where the observed values of MAF are small (e.g. Hungary and Denmark), and underestimated where they are large (e.g. Carpathians, Northern Italy) reflecting the tendency of spatial estimators to underestimate spatial extremes. The overestimation in Finland may also be due to lake retention not being captured adequately in the model. To some degree CV is also overestimated in areas of low CV (e.g. southern Norway and Denmark) and underestimated in areas of high CV (e.g. Ukraine, Ore mountains) although there are also large CV areas it is overestimated (e.g. Southern Spain). The errors are smallest in Russia, Central Germany, British Isles and France where the spatial gradients in CV are relatively smooth.

The median absolute normalized error of MAF and CV, throughout Europe, is 0.37 and 0.18, respectively, with 25%-quantiles of 0.17 and 0.09 and 75%-quantiles of 0.63 and 0.32. This means that the absolute normalized error of CV is about half that of MAF, which seems to be related to the relatively smaller spatial variability of CV as compared to MAF (spatial cvs of 1.11 and 0.49, respectively, Table 2).

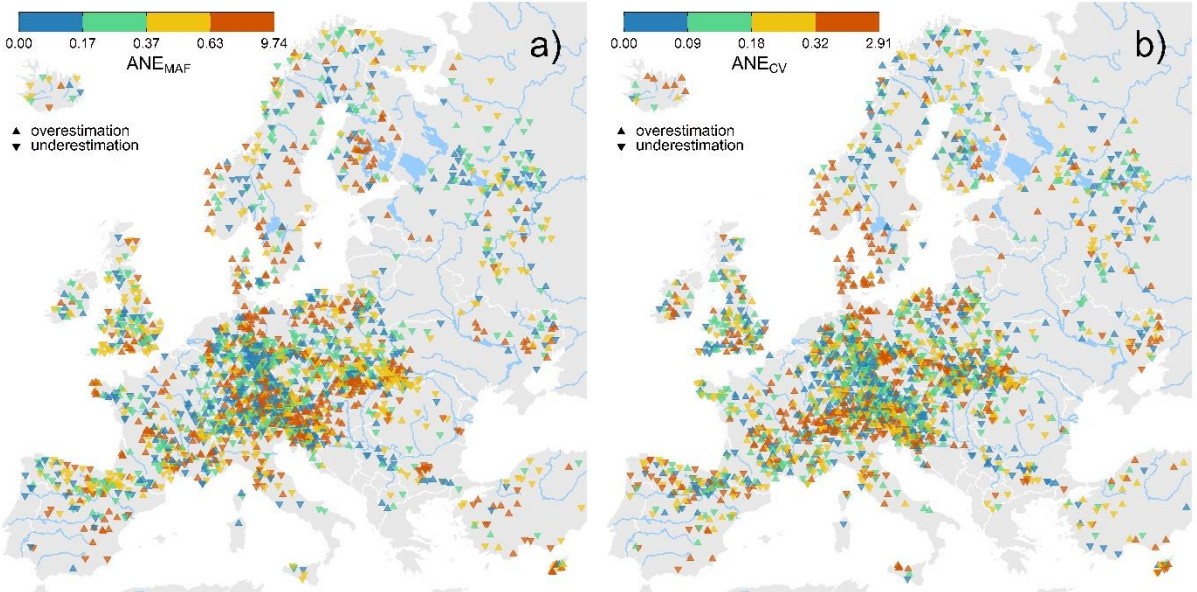

Figure 9: Absolute normalised errors ANE_MAF and ANE_CV of the predictions of the regional regression models for MAF and CV. Errors are evaluated on scale of data (not logarithmised). Colours refer to binned classes of equal frequency. Triangles facing upwards indicate gauges where the model overestimates the moments, triangles facing downwards where the model underestimates.

Figure 10 depicts the predicted (leave-one-out) MAF and CV using the regressions (Circles in Figure 10, Tables A.1.1 and A.1.2) and the predicted MAF and CV by spatial proximity through kriging of the observed moments (background colour). Predictions are shown on the scale of the data (not logarithmised) using the colour scale from

figure 2. Notwithstanding the relatively large ANE (Fig. 9), the overall patterns of the moments are very similar to the observed ones (Figure 2). Overall, the patterns of the moments are consistent with the process reasoning put forward in this paper.

The intention of Figure 10 is to offer a visual comparison between the two regionalization approaches. Before the use of ordinary kriging estimates for applications in ungauged basins, additional cross-validations would be useful in the spirit of Rosbjerg et al. (2013).

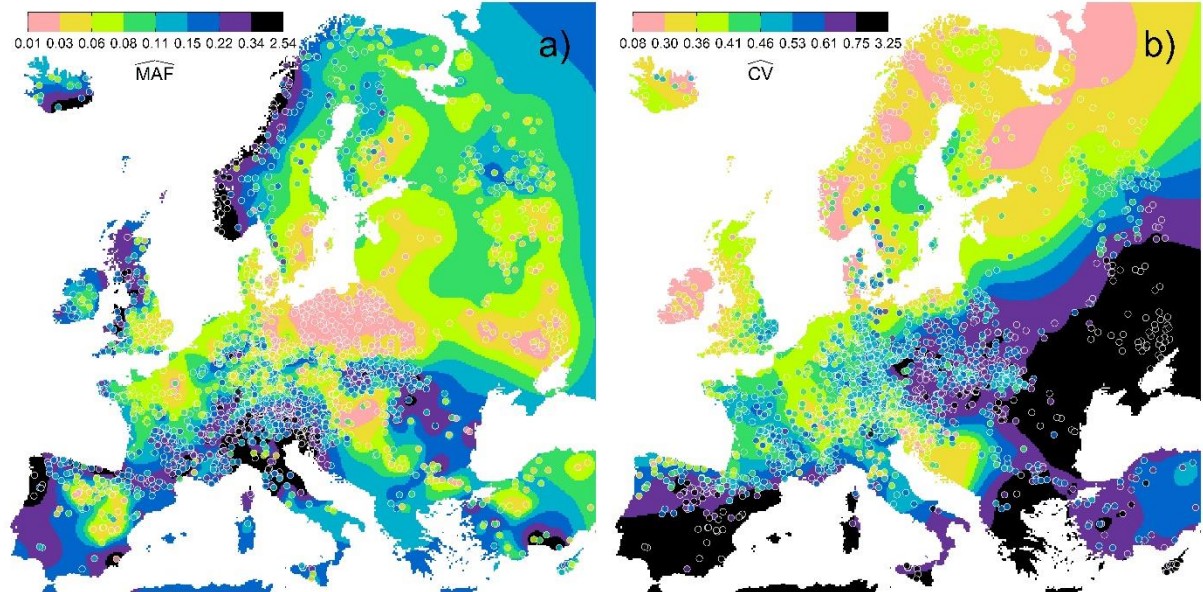

Figure 10: Predicted values of regional regression models and ordinary kriging for MAF (m³ s$^{-1}$ km$^{-2}$) and CV. Circles refer to
predictions of regression models, background refers to predictions of kriging. Colours refer to binned classes of equal frequency for original data (estimated statistical moments) as in Figure 2.

## 4.  Discussion and Conclusions

### 4.1 Patterns of flood moments in Europe

Overall, there are clear patterns in flood moments across Europe. As expected, MAF shows the clearest patterns while they are less clear for CV and particularly CS, at least partly because of sampling variability.

In the Atlantic region, where floods mainly occur in winter as a result of moisture influx from the ocean, MAF$_\alpha$ is very high (above 0.5 m³/s/km² along the western coasts of Norway, Scotland and Galicia, Fig. 2). The CVs, on the other hand, are small (typically around 0.3 at the Norwegian coast and in Scotland and 0.5 in Galicia) as the
atmospheric moisture influx tends to be consistent between years (Giorgi et al., 2004). As one moves towards the continent from the west coast of the British Isles, the MAF tends to decrease and the CV increases because of the decreasing and more variable moisture availability and floods tend to occur later in the winter (i.e. Jan instead of Dec., Fig. 3).

Further inland, various mountain ranges (Pyrenees, Massif Central, Alps Apennines, Ore mountains, Carpathians,
Balkan mountains) stand out with higher MAF than the surrounding areas (mostly above 0.3 m³/s/km²) and summer as the dominant flood season due to their effects of enhancing rainfall and probably shallower soils as well as steeper slopes and smaller watersheds. On the other hand, there are clear differences between the CVs of these

mountain ranges. While CVs in the Alps and the southern Slovenian mountains are low, they are high in the Ore mountains and some of the other mountain ranges, which reflects the stronger influence of Mediterranean storm tracks with high variability of extreme precipitation (Hofstätter et al., 2018) perhaps along with non-linear runoff generation (Viglione et al., 2009).

As one moves inland in northern Europe from the Norwegian coasts towards the North European plain, MAF decreases to values around $MAF_a$=0.1 m³/s/km² due to a decrease in the Atlantic influence and increase of snow processes resulting in late spring events with pronounced seasonality. As one moves southward from the Northeast, CV increases and flood seasonality decreases in line with the increased influence of rain-on-snow and rain floods, which tend to be more irregular than snowmelt floods alone.

Some of the continental regions of Europe (Hungary, Poland, Ukraine) are particularly sheltered by mountain chains, resulting in low precipitation, both at the annual scale and for extreme events which translates into low MAF and mostly high CV due to the more non-linear runoff generation as compared to wetter regions (Nováaky, 1991, Didovets et al., 2017, Ries et al., 2017).

In the Mediterranean, where floods tend to occur in autumn and winter, MAF is generally high, particularly in Southern France, Slovenia and Croatia, Liguria (NW Italy), and parts of northern Italy, due to heavy autumn storms stemming from the warm sea-surface temperatures. In most regions adjacent to the Mediterranean where these storms predominate, annual floods also have high CV due to the interannual variability of these storms. However, this is not the case in Slovenia and Croatia, due to the consistency of these storms between years (Xoplaki et al., 2004, Salinas et al., 2014).

### 4.2 Interpretation of controls of flood moments

The degree to which the moments change with catchment area is a fingerprint of the spatial variability of flood producing processes (Merz et al., 2003, Sivapalan et al., 2005, Merz and Blöschl, 2009, Viglione et al., 2010) (Table 3, Figure 4). As expected, there is a strong scaling effect of MAF with area. The strongest decrease of MAF with area is observed for the Mediterranean region, which points to the important role of small scale, convective storms, patchy runoff generation processes, and more generally flash floods there (Gaume et al., 2009; Marchi et al., 2010; Amponsah et al., 2018). On the other hand, the smallest decrease with area is found in the Central-Eastern and Northeastern regions, where snowmelt is a dominant flood driver. Snowmelt tends to occur over larger regions simultaneously, which results in a smaller reduction of MAF with catchment scale (Blöschl and Sivapalan, 1997; Merz and Blöschl, 2003). The relatively weak decrease in the Atlantic regions suggests an important role of large-scale precipitation from large frontal systems as would be expected (Kemter et al., 2020).

CV decreases with area in most of the regions. Again, the strongest decrease is observed for the Mediterranean region, which can be interpreted in terms of similar aggregation processes as in the case of MAF. Additionally, the degree of non-linearity in runoff generation may decrease with catchment size (Sivapalan, 2003) as threshold processes associated with Hortonian runoff generation or soil storage homogeneity may be more relevant in small catchments while in large catchments these threshold effects may be smoothed out, and spatiotemporal aggregation may introduce additional scale effects (Penna et al., 2011; Rogger et al. 2012). On the other hand, the smallest decrease occurs in the Northeastern region and in the Central Eastern region, where the estimated regression

coefficient is even positive. While both relationships are not significant, they do point towards the larger scale of snowmelt processes relative to other flood generation processes along with more linear runoff generation processes and the larger role of baseflow there (Blöschl and Sivaplan, 1997, Grillakis et al., 2016).

The strongest Spearman correlation (in absolute value) for MAF is observed for the Mediterranean region, while the weakest is observed in the Central-Eastern region (Table A.1.5), in line with the difference in the scale dependence between these two regions (Table 3). For CV (Figure 6) the spatial differences of the correlations between the regions are also consistent with the scaling regressions of Table 3 but, overall, they are smaller than those of MAF. This may be related to possible non-monotonous relationships between CV and area as suggested by Smith (1992), and more complex aggregation effects (Blöschl and Sivapalan 1997), although more research is needed on the transferability of this finding.. The weakest relationships are found in Central-Eastern Europe, where snow is important, and in the Alpine region where the spatial variability of other controls is particularly large (r=0.09 and -0.03, respectively).

As compared to the other controls on the flood moments, area plays an important role for MAF but less so for CV (Table A.1.5, Figures 7 and 8). In the Alpine region the Spearman correlations between MAF and area are larger than those between area and other covariates (Table A.1.5) and it has the large contributions to the fit of the regional model (Figure 7). However, this may not be because of the high explanatory power of area but of the lower explanatory power of the other covariates likely related to the complex topography. For CV, area has some explanatory power in the Atlantic and Mediterranean regions (Figure 6) and is used in most regional regression models, but the role of climate variables such as precipitation and aridity is always much higher than that of area (Figure 6, Figure 8) suggesting that aggregation effects are relatively less important at the European scale than at the regional scale. This finding is likely related to the larger spatial variability of climate variables within Europe than within a region.

Precipitation characteristics are represented by three variables, which are strongly correlated among themselves (Figure 5). The Spearman correlations between MAF and the precipitation characteristics are positive for Europe and in individual regions while for CV and precipitation they are negative (Figure 6). MAP is a surrogate for the combined effects of event precipitation, antecedent soil moisture and the geomorphological processes of landscape evolution that affect runoff generation and routing, whereas P95 and Pmax are more representative of characteristics of event precipitation alone. Pmax is representative of a more extreme part of the daily precipitation distribution than P95. The correlations of MAP and P95 with $MAF_\alpha$ are similar, but generally larger than that of Pmax which may be due to annual maxima precipitation events with low antecedent soil moisture storage. While regional studies have suggested that MAP is a better predictor of MAF than other precipitation variables (Mimikou and Gordios, 1989; Merz and Blöschl, 2009) this does not seem to be the case at the European scale (see e.g. Figure 6).

On the other hand, CV is always better correlated with MAP than with P95 and Pmax, reflecting the decreasing degree with which antecedent soil moisture is captured as one moves from MAP to P95 and Pmax, since MAP better captures soil moisture conditions. Drier catchments can produce larger CVs because the antecedent soil moisture conditions tend to vary more than they do in humid catchments with consistently high soil moisture storage. Consequently, some of the events in drier catchments may be a combination of both large precipitation

and wet initial conditions such producing much larger floods than usual (Farquharson et al., 1992, Viglione et al., 2009; Kemter et a., 2020). This effect is also represented in the negative correlations between CS and MAP (r=-0.35) and CS and P95 (r=-0.34) (Table A.1.5) in the Mediterranean, indicating a decrease in skewness for comparatively wetter catchments, which is related to a particularly large potential for this contrast in initial conditions.

In the context of multiple controls, rainfall (in this case only P95 was considered) is always among the important variables for explaining MAF (Figure 7). However, in the case of CV, aridity is vastly more important, as the combination of evaporation and precipitation better captures the typical initial condition state of the catchments before floods. Also, arid regions tend to have greater interannual precipitation variability (Fatichi et al., 2012), but if a region becomes drier its interannual precipitation variability will not necessarily increase (Pendergrass et al., 2017). On the other hand, drying of a region may imply more non-linear runoff generation processes and thus enhance the CV of floods (Viglione et al., 2009).

Mean spring and winter temperature were used in the analysis to capture snow processes because of the better data availability. Chaoimh (1998) and Bednorz (2003) identified correlations between spring and winter temperature with snowpack-depth and days with snow-cover, and more generally air temperature is often used as an indicator of snowmelt (Ohmura, 2001). Future work could enrich the analysis by using snow data directly, although remote sensing products may have some limitations related to the duration (see e.g. Parajka and Blöschl, 2012). As would be expected, the Spearman correlations between temperature and $MAF_\alpha$ and CV are comparatively high in the Northeastern and Central-Eastern region (higher for spring temperature), where snow-processes are important for floods. Temperatures are negatively correlated with $MAF_\alpha$ and positively correlated with CV. The colder it is, the more precipitation is stored as snowpack in winter, leading to, on average bigger snowmelt floods in spring/early summer, but they are less variable (smaller CV) which may be related to the smaller interannual variability of air temperature as compared to that of precipitation (Giorgi et al. 2004). Snowmelt floods are physically limited by the amount of water stored as snow and solar radiation. This upper limit also contributes to less extreme and more regular floods (Merz and Blöschl, 2003). In these cold regions, temperatures are better correlated with $MAF_\alpha$ and CV than with the precipitation variables and almost as well as with aridity, although this depends on the moment and the region.

Winter temperatures add explanatory power to the regional regression models of MAF in the Central-Eastern region, but a rather small contribution for the other regions (Figure 7). For CV, temperatures explain some of the spatial variability in the Central-Eastern and the Atlantic region, but generally its contribution is low (Figure 8). On the one hand, winter temperature may not be a perfect proxy for the spatial distribution of flood-relevant snowmelt, whereas the other variables may more directly capture runoff generation processes.

Soil moisture, PET and the aridity index (AI) are related to long-term water balance characteristics and are significantly correlated with the estimated flood moments for almost all regions. Soil moisture is positively correlated with $MAF_\alpha$ and negatively correlated with CV, whereas the opposite holds for PET and AI. Soil moisture and MAP are highly correlated, which is related to soil moisture being driven by long term precipitation. The lower the wetness state, the more room for variations in the runoff coefficients between years, and therefore flood peaks and thus high CV (Viglione et al., 2009). For PET and AI, the highest correlations with $MAF_\alpha$ are

observed in the Northeastern, the Atlantic and the Mediterranean region. AI is strongly correlated with CV (Figure 6) and a particularly important variable for capturing the spatial variability of CV in regional regression models (Figure 8). High aridity implies a combination of low precipitation and high evaporation, leading to comparatively dry antecedent conditions. AI may also capture the non-linearity of runoff-generating mechanisms relevant for CV (Blöschl and Sivapalan, 1997). Additionally, precipitation tends to be more variable in the arid regions of Europe (e.g. Giorgi et al., 2004), so there may be both a precipitation and runoff generation effect, the latter being related to the stronger randomness of the runoff coefficient. The effect of the large variability of runoff coefficients between years in the arid catchments of Europe (large AI) is also apparent in the positive correlations with CS in the Mediterranean and Central-Eastern region (r=0.35 and 0.50, respectively) (Table A.1.5)

Topography is included via slope and elevation in the present analysis, but the observed effects of topography on the flood moments are most likely indirect. Precipitation characteristics are highly correlated with topographical indices (Figure 5) and their spatial patterns are very similar (not shown), suggesting little unique effect. Faster routing (flow velocity) due to topography does not seem to be a relevant factor for the spatial patterns of flood moments at the European scale, given that response times may be more closely related to geology than topography at the regional scale (Gaál et al., 2012).

The fraction of area covered by forest (LUF) is positively correlated with $MAF_\alpha$ which is not consistent with the usual expectation of higher infiltration capacities and therefore smaller floods peaks for forest soils (Sun et al., 2018). At the European scale, apparently, this effect is masked by the correlations between forest cover and precipitation. In high elevation regions of Europe forest cover tends to be high as these areas have not been deforested for agricultural purposes, and these are also the areas of high rainfall because of topographic effects on rainfall. Additionally, runoff coefficients may be higher in these high rainfall areas due to shallower soils and water tables notwithstanding the forest cover (Merz et al., 2006, Rogger et al., 2017).

The fraction of area covered by water bodies reduces both $MAF_\alpha$ and CV. The former is consistent with retention effects while the relationship between CV and water body size may be non-linear (increasing CV up to a water body threshold and decreasing CV beyond as shown by Wang et al., 2017 for reservoir effects) which is not captured by Spearman correlation. However, in comparing natural lakes and reservoirs it should be noted that reservoirs tend to introduce more non-linearity in flood frequency behaviour because of a threshold effect when the spillway is activated.

Soil texture, when interpreted in terms of pedotransfer functions (Picciafuoco et al., 2019), is expected to affect infiltration of event rainfall. Coarse soils (Stex=1) are therefore expected to be associated with smaller $MAF_\alpha$ than fine soils (Stex=5) but the data show no consistent relationship. In a similar vein, the data suggest that coarse soils tend to be associated with small CV, which contrasts what one would expect by reasoning in terms of runoff generation processes. For coarse soils one would expect that, for small events, most of the water infiltrates but when a threshold is exceeded the rainfall starts to run off from the surface, thus leading to larger CV relative to soils without threshold behaviour (Rogger et al., 2013). One reason for observing the opposite are the correlations between MAP and Stex within the regions which range between -0.08 and -0.36, i.e., coarse soils would be associated with high precipitation, which would explain large MAF and small CV. Apparently at the scale of an entire continent, the soil characteristics (at least the texture available here) are less important than climate variables.

The rather low explanatory power of soil texture and land use for hydrological response at the regional scale is a general concern that also affects the estimation of other variables in the context of predictions in ungauged basins (Blöschl et al., 2003).

While here we examined monotonic relationships and linear relationships, it would also be worth to explore non-monotonic relationships between flood moments and covariates (see e.g. Blöschl and Sivapalan, 1997; Smith, 1992; Pallard et al., 2009). Possible approaches for modelling non-monotonic relationships include generalized additive models (Rahman et al., 2018, Umlauf and Kneib, 2018) and Random forest regression (Desai et al., 2021).

The properties of the estimators of the investigated correlations and linear regressions depend on assumptions which are only partly met in this analysis. Tables A.1 and A.2 in the appendix report the maximum Variance inflation factors [VIF] for each regional regression model from section 3.5, as well as p-values of hypothesis tests for the homoscedasticity and normality of the residuals. While the VIFs are generally low (indicating a low degree of multicollinearity), the assumption of homoscedasticity and normality of the residuals are generally not met for many models, which may be related to the large number of catchments. Additional diagnostic plots for the regional regression models can be found in the supplementary material. The OLS-estimator still remains unbiased and consistent under these conditions (Hayashi, 2000), but no inferences such as significance tests of individual coefficients should be made from standard properties of the OLS-estimator. In Tables A.1 and A.2 we report the standard errors of the coefficient estimators, which should be interpreted with care and are thus not used for hypothesis tests. The inclusion of additional covariates could help to reduce heteroscedasticity, but would lead to less parsimonious models. Alternatively, heteroscedasticity could be reduced by considering different regional partitions of Europe.

### 4.3 Implications

Even though the main objective of this paper is to investigate process controls on spatial patterns of flood moments in Europe, the results in Section 3.6 may be considered as a benchmark for flood moment estimation in ungauged basins at the European scale. The median absolute normalized error (ANE) of MAF and CV is 0.37 and 0.18, respectively. This is relatively large as compared to similar studies at smaller spatial scales in the literature on flood regionalization, which typically yield ANE of 0.35 for the 100-yr specific flood and smaller values for the MAF (Salinas et al., 2013; Rosbjerg et al., 2013). The fit of the regional models varies between the regions, which reflects differences in the relative importance of the flood-generating processes between the regions. For the case of MAF, $R^2$ is largest in the Atlantic and Mediterranean regions (Table A.1.1) and for CV it is largest in Eastern and Mediterranean regions. Clearly, there is no model applicable to all regions of Europe. The regions here were derived based on previous climatic partitions of Europe and guided by flood seasonality rather than optimal predictive performance of the regional models. The results depend on the regional partitioning of Europe and will look different for different partitioning schemes. If the aim of the study was optimal predictive performance of the regional models, the partitioning could be derived based on the data, for example via cluster analysis or regression trees (see e.g. Laaha and Blöschl, 2006).

Overall, the findings of this paper suggest that, at the continental scale, climate variables dominate over land surface characteristics in their control of the spatial patterns of flood moments. Given the evidence for the coevolution of landscape and climate (Perdigão and Blöschl, 2014, Troch et al., 2015) but the general lack of

predictive power of variables related to land use, soil and geology for hydrological quantities that one would expect to be very relevant at individual sites (Merz and Blöschl, 2009, Rogger et al., 2017), there is a need for new types of land characteristics consistent across countries that can explain spatial differences in flood-generation processes better. Merz and Blöschl (2009) illustrate this need through a comparison of two Austrian catchments that have strikingly similar geological characteristics in terms of percentage of area of certain geological types, but vastly different rainfall-runoff response-behaviour. At the plot, hillslope and catchment scales, runoff generation is strongly controlled by soil properties, including their control infiltration and saturation capacities (Peschke and Sambale, 1999; Scherrer et al., 2007; Rogger et al., 2012; Picciafuoco et al., 2019). There have been attempts to relate or upscale local soil characteristics and regional ones (e.g., Schmocker-Fackel et al., 2007). One successful example is the HOST classification used in the UK (Boorman et al., 1995; Lilly et al., 1998; Maréchal and Holman, 2005), which has been demonstrated to be able to capture runoff generation processes and their spatial variability. Of course, scaling becomes important as well, as land-use may have larger explanatory power in small catchments than in larger ones (Rogger et al., 2017). The finding that climate is the main control for the spatial variability of the flood moments, within the range of the variables considered, also has some implications for quantifying the temporal flood variability. If the spatial patterns of flood behaviour at the continental scale are primarily driven by climatic influences, their temporal fluctuations might be propagated to floods (Šraj et al., 2016, Blöschl, Hall et al., 2019, Bertola et al., 2020, Kemter et al., 2020) On the other hand: flood changes of small local streams may be much more controlled by land use changes, such as urban development and deforestation (Rogger et al., 2017), only a few of which are included in this study (average catchment size of $2,480km^2$). One should however be careful in trading space for time in the context of change, i.e. in assuming that future flood characteristics in one region will be similar to the present ones in another region because the climate in the former will be similar to the present climate in the latter. This is because of the space-time asymmetry discussed in Perdigão and Blöschl (2014), i.e. the fact that, because of the celerity of coevolution, spatial and temporal statistics are not necessarily the same. For example, based on data in Austria, Perdigão and Blöschl (2014) found that a 1% increase in precipitation as one moves in space leads to a 2.3% increase in flood peaks, while the same increase in precipitation as one moves in time leads to an increase of only 0.6%. Overall, this paper is a step toward a better process-based understanding of the statistical properties of annual floods in Europe. The process controls identified here can assist in choosing suitable covariates, both for stationary and nonstationary flood frequency models. A possible extension of the analysis presented here could be the consideration of non-stationarities in flood moments, for example in the spirit of Serago of Vogel (2018). Blöschl, Hall et al. (2019) have found that significant trends do exist in the mean flood of the data set in 28.02% of the stations. Trends affect the estimation of flood moments. For example the detrended data tend to exhibit smaller CVs than the raw data, while the effect on the sample mean may be smaller.

Further mixed-distributions analyses could consider different subpopulations of floods associated with specific generation mechanisms and yield additional insights regarding spatial patterns of process controls (e.g. Fischer et al., 2016, Tarasova et al., 2019), e.g. as indicated by their seasonality (Blöschl et al., 2017). Additionally, we believe that a more comprehensive representation of catchment functioning that goes beyond soil types has the potential to further improve our understanding of process controls on flood probabilities.

## 5. Appendix

Table A.1 Regression coefficients (standard errors), model error variance, $R^2$ and maximum variance inflation factor (VIF) of regional regression models for MAF. Last two columns contain p-values for the Breusch-Pagan (BP) and the Shapiro-Wilk test (SW). Statistically significant results (small p-values) indicate heteroscedasticity and non-normality of the residuals respectively. All numbers are rounded to two digits. For details on VIF see e.g. Weisberg (2005) and for details on the hypothesis tests see e.g. Helsel (2020).

| $Log_{10}$ MAF | (Intercept) | $Log_{10}$ A | $Log_{10}$ P95 | TWin | AI | sigma | R2 | VIF | BP | SW |
|---|---|---|---|---|---|---|---|---|---|---|
| Europe | -2.18(0.09) | -0.16(0.01) | 1.61(0.07) | -0.02(0.00) | -0.04(0.02) | 0.31 | 0.47 | 2.09 | 0.00 | 0.00 |
| Northeastern | -1.76(0.42) | -0.14(0.02) | 1.22(0.31) | -0.05(0.01) | -0.33(0.08) | 0.24 | 0.41 | 3.89 | 0.00 | 0.00 |
| Atlantic | -3.31(0.11) | -0.13(0.01) | 2.54(0.09) | 0.01(0.00) | | 0.26 | 0.51 | 1.04 | 0.00 | 0.00 |
| Central-Eastern | -4.54(0.62) | -0.09(0.03) | 3.33(0.59) | -0.07(0.01) | 0.10(0.07) | 0.25 | 0.33 | 1.61 | 0.00 | 0.00 |
| Alpine | -0.68(0.21) | -0.18(0.02) | 0.47(0.14) | | -0.16(0.06) | 0.27 | 0.27 | 2.46 | 0.00 | 0.02 |
| Mediterranean | -1.09(0.26) | -0.22(0.02) | 0.94(0.19) | 0.02(0.01) | -0.16(0.04) | 0.32 | 0.47 | 3.24 | 0.00 | 0.09 |

Table A.2. Regression coefficients (standard errors), model error variance, $R^2$ and maximum variance inflation factor (VIF) of regional regression models for CV. Last two columns contain p-values for the Breusch-Pagan and the Shapiro-Wilk test. Statistically significant results (small p-values) indicate heteroscedasticity and non-normality of the residuals respectively. All numbers are rounded to two digits. For details on VIF see e.g. Weisberg (2005) and for details on the hypothesis tests see e.g. Helsel (2020).

| $Log_{10}$ CV | (Intercept) | $Log_{10}$ A | $Log_{10}$ P95 | TWin | AI | sigma | R2 | VIF | BP | SW |
|---|---|---|---|---|---|---|---|---|---|---|
| Europe | -0.49(0.05) | -0.05(0.00) | 0.06(0.03) | 0.00(0.00) | 0.22(0.01) | 0.15 | 0.29 | 2.09 | 0.00 | 0.00 |
| Northeastern | -1.31(0.10) | -0.02(0.01) | 0.54(0.08) | | 0.47(0.03) | 0.1 | 0.48 | 1.73 | 0.25 | 0.14 |
| Atlantic | -0.40(0.02) | -0.05(0.01) | | -0.02(0.00) | 0.21(0.01) | 0.14 | 0.28 | 1.05 | 0.01 | 0.00 |
| Central-Eastern | -2.43(0.29) | | 1.42(0.29) | -0.04(0.00) | 0.60(0.04) | 0.13 | 0.61 | 1.37 | 0.10 | 0.06 |
| Alpine | 0.50(0.10) | -0.05(0.01) | -0.64(0.07) | 0.01(0.00) | 0.08(0.03) | 0.13 | 0.37 | 2.76 | 0.19 | 0.08 |
| Mediterranean | 0.23(0.11) | -0.08(0.01) | -0.45(0.08) | | 0.12(0.02) | 0.14 | 0.55 | 2.29 | 0.00 | 0.42 |

Table A.3. Measure for general dominance (additional contributions) for MAF – indicating general dominance. For each row: highest value indicates most important variable in terms of improvement of the model fit for the given regression, second highest indicates second most important, lowest indicates least important. Summing over measures gives $R^2$ of regression.

| $Log_{10}$ MAF | $Log_{10}$ A | $Log_{10}$ P95 | TWin | AI | R2 |
|---|---|---|---|---|---|
| Europe | 0.12 | 0.25 | 0.01 | 0.09 | 0.47 |
| Northeastern | 0.14 | 0.1 | 0.03 | 0.14 | 0.41 |
| Atlantic | 0.08 | 0.43 | 0 | | 0.51 |
| Central-Eastern | 0.06 | 0.13 | 0.13 | 0.01 | 0.33 |
| Alpine | 0.15 | 0.07 | | 0.06 | 0.27 |
| Mediterranean | 0.2 | 0.15 | 0.03 | 0.1 | 0.47 |

Table A.4. Measure for general dominance (additional contributions) for CV – indicating general dominance. For each row: highest value indicates most important variable in terms of improvement of the model fit for the given regression, second highest indicates second most important, lowest indicates least important. Summing over measures gives $R^2$ of regression.

| $Log_{10}$ CV | $Log_{10}$ A | $Log_{10}$ P95 | TWin | AI | R2 |
|---|---|---|---|---|---|
| Europe | 0.03 | 0.04 | 0.01 | 0.21 | 0.29 |
| Northeastern | 0.02 | 0.05 | | 0.42 | 0.48 |
| Atlantic | 0.04 | | 0.05 | 0.19 | 0.28 |
| Central-Eastern | | 0.03 | 0.14 | 0.44 | 0.61 |
| Alpine | 0.03 | 0.2 | 0.01 | 0.13 | 0.37 |
| Mediterranean | 0.09 | 0.2 | | 0.26 | 0.55 |

Table A.5: Spearman-Correlation between statistical moments of flood series, including mean specific discharge MAF, mean specific discharge normalized to a catchment area $MAF_\alpha$ of $\alpha=100km^2$, the coefficient of variation CV, the coefficient of skewness CS, and catchment attributes. Statistically significant estimates (at the 5% level) are printed in bold.

| | Europe | | | | Northeastern | | | | Atlantic | | | | Central-Eastern | | | | Alpine | | | | Mediterranean | | | |
|---|---|---|---|---|---|---|---|---|---|---|---|---|---|---|---|---|---|---|---|---|---|---|---|---|
| | MAF | $MAF_\alpha$ | CV | CS | MAF | $MAF_\alpha$ | CV | CS | MAF | $MAF_\alpha$ | CV | CS | MAF | $MAF_\alpha$ | CV | CS | MAF | $MAF_\alpha$ | CV | CS | MAF | $MAF_\alpha$ | CV | CS |
| A | **-0.44** | **-0.12** | **-0.13** | **-0.13** | **-0.47** | 0.01 | **-0.16** | -0.07 | **-0.31** | 0.01 | **-0.19** | **-0.08** | **-0.23** | -0.01 | 0.09 | **0.17** | **-0.40** | 0.02 | -0.03 | **-0.15** | **-0.48** | 0.03 | **-0.25** | **-0.18** |
| MAP | **0.62** | **0.59** | **-0.33** | -0.01 | **0.18** | **0.15** | **-0.23** | 0.15 | **0.64** | **0.67** | **-0.33** | -0.03 | **0.25** | **0.24** | **-0.49** | **-0.38** | **0.34** | **0.26** | **-0.61** | -0.13 | **0.42** | **0.48** | **-0.68** | **-0.35** |
| P95 | **0.64** | **0.60** | **-0.22** | 0.02 | **0.21** | 0.02 | -0.10 | **0.17** | **0.67** | **0.66** | **-0.22** | 0.00 | **0.39** | **0.31** | **-0.19** | **-0.26** | **0.32** | **0.24** | **-0.58** | -0.14 | **0.49** | **0.52** | **-0.59** | **-0.34** |
| Pmax | **0.61** | **0.55** | -0.05 | **0.12** | **0.19** | -0.04 | -0.09 | **0.20** | **0.53** | **0.52** | 0.14 | **0.17** | **0.30** | **0.22** | **-0.18** | **-0.23** | **0.36** | **0.26** | **-0.40** | -0.09 | **0.57** | **0.49** | **-0.36** | **-0.20** |
| TSpr | **-0.22** | **-0.22** | **0.26** | 0.00 | **-0.31** | **-0.35** | **0.58** | **0.23** | **-0.29** | **-0.29** | 0.02 | -0.14 | **-0.14** | -0.11 | **0.50** | **0.41** | -0.17 | -0.14 | **0.38** | -0.04 | -0.09 | **-0.12** | **0.17** | 0.02 |
| TWin | -0.04 | -0.07 | 0.04 | -0.06 | -0.04 | **-0.20** | **0.20** | **0.30** | -0.02 | -0.03 | **-0.23** | **-0.17** | **-0.28** | **-0.33** | **-0.52** | **-0.31** | -0.11 | -0.10 | 0.10 | -0.12 | 0.12 | -0.04 | **0.27** | 0.14 |
| SM | **0.58** | **0.57** | **-0.31** | -0.03 | **0.17** | **0.16** | -0.14 | **0.19** | **0.52** | **0.55** | **-0.29** | -0.05 | **0.28** | **0.28** | **-0.4** | **-0.33** | **0.26** | **0.22** | **-0.56** | -0.10 | **0.45** | **0.50** | **-0.64** | **-0.37** |
| PET | **-0.05** | -0.03 | **0.39** | **0.14** | **-0.30** | **-0.33** | **0.60** | **0.22** | **-0.23** | **-0.22** | **0.34** | 0.03 | 0.02 | 0.07 | **0.68** | **0.45** | **-0.16** | -0.11 | 0.11 | -0.10 | -0.11 | **-0.21** | **0.63** | **0.31** |
| AI | **-0.55** | **-0.53** | **0.46** | 0.07 | **-0.36** | **-0.36** | **0.62** | 0.12 | **-0.51** | **-0.53** | **0.38** | 0.04 | **-0.16** | -0.12 | **0.63** | **0.50** | **-0.39** | **-0.31** | **0.50** | 0.08 | **-0.37** | **-0.43** | **0.69** | **0.35** |
| Elev | **0.53** | **0.55** | 0.08 | **0.20** | **0.35** | **0.47** | **-0.31** | 0.06 | **0.38** | **0.44** | **0.35** | **0.22** | **0.48** | **0.50** | **0.21** | **0.16** | **0.19** | **0.16** | **-0.47** | 0.00 | 0.06 | 0.02 | **0.20** | **0.20** |
| Slope | **0.63** | **0.65** | -0.01 | **0.16** | **0.34** | **0.39** | **-0.36** | 0.04 | **0.58** | **0.62** | **0.17** | **0.19** | **0.29** | **0.30** | -0.01 | 0.02 | **0.25** | **0.24** | **-0.40** | -0.01 | **0.33** | **0.37** | 0.06 | 0.06 |
| LUF | **0.46** | **0.45** | **-0.10** | 0.06 | **0.30** | **0.28** | -0.01 | 0.05 | **0.43** | **0.45** | **0.14** | **0.14** | 0.08 | 0.01 | **-0.39** | **-0.24** | **0.27** | **0.16** | **-0.26** | 0.03 | **0.40** | **0.32** | 0.03 | -0.03 |
| LUW | **-0.30** | **-0.16** | **-0.27** | **-0.15** | **-0.25** | **-0.21** | **-0.22** | -0.06 | **-0.19** | -0.04 | **-0.20** | -0.06 | **-0.55** | **-0.55** | **-0.50** | **-0.23** | **-0.16** | 0.03 | **-0.19** | **-0.14** | **-0.40** | -0.10 | **-0.19** | -0.10 |
| Stex | **-0.06** | **-0.06** | **0.29** | **0.11** | **-0.12** | -0.07 | **0.37** | **0.25** | -0.06 | -0.05 | **0.17** | -0.01 | **0.29** | **0.32** | **0.53** | **0.31** | -0.02 | -0.06 | **0.23** | 0.01 | **-0.22** | **-0.18** | **0.26** | **0.16** |

Table A.6: $R^2$ of regional regression models of MAF with catchment area in a double logarithmic relationship Eq. (5) and analogous equations for CV; and a semi logarithmic relationship for CS, i.e. $CS = \log A\ \beta_{CS}$.

| Europe | Northeastern | Atlantic | Central-Eastern | Alpine | Mediterranean |
|---|---|---|---|---|---|
| 0.188 | 0.190 | 0.109 | 0.066 | 0.188 | 0.234 |
| 0.015 | 0.009 | 0.028 | 0.007 | 0.007 | 0.084 |
| 0.010 | 0.005 | 0.006 | 0.029 | 0.020 | 0.037 |

**Data Availability**

The flood data used in this paper are available at https://github.com/tuwhydro/europe_floods. The authors acknowledge the E-OBS dataset from the EU-FP6 project UERRA (http://www.uerra.eu) and the Copernicus Climate Change Service, and the data providers in the ECA&D project (https://www.ecad.eu), as well as the CCM River and Catchment database (https://data.europa.eu/), the Global Aridity Index and Potential Evapo-Transpiration Climate Database (https://doi.org/10.6084/m9.figshare.7504448.v3), the CPC Soil Moisture database (https://www.cpc.ncep.noaa.gov/), the data on terrain elevation provided by the USGS (https://www.usgs.gov/), the Corine Land Cover data set (https://land.copernicus.eu/) and the European soil database (https://esdac.jrc.ec.europa.eu).

**Author Contributions**

DL performed the analysis and prepared the paper. GB designed the overall study. All co-authors contributed to the interpretation of the results and writing of the paper.

**Competing Interests**

The authors declare that they have no conflict of interest.

**Acknowledgements**

The authors would like to acknowledge funding from the Austrian Science Funds (FWF) "SPATE" project I 4776 and the German Research Foundation (DFG; grant no. FOR 2416), the FWF Vienna Doctoral Programme on Water Resource Systems (W1219-N28) and the European Union's Horizon 2020 Research and Innovation Programme under the Marie Skłodowska-Curie grant agreement no. 676027. We would like to thank the editor Thomas Kjeldsen, Kolbjorn Engeland and an anonymous referee for their useful comments on the original version of the paper.

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
