# Peer review of "Characteristics and process controls of statistical flood moments in Europe – a data based analysis"

_Hydrology and Earth System Sciences, 2020_

## Referee Comment (RC1) · Kolbjorn Engeland (Referee) · 14 Feb 2021

The paper provides a comprehencive analysis of a datset of annual maximum floods covering all Europe and aims to dicuss how process controls can exolain the spatial pattenrs of mean annual floods and the coeficient of variation (CV) of floods. The paper comes in a line of papers anaysing floods at a European scale (Blöschl et al, 2017; Hall and Blöschl, 2018; Blöschl et al., 2019 and Blöschl et al., 2020). Whereas the previous papers have investigated trnds in time, this paper has a clear focus on the spatial pattenrns. This provides therefore new knowledge and is complentary to the previous papers. The paper is well written and could in my opinion be published after

some minor revisions.

Lines 38:48: This is because a large basin is less likely to be fully covered by a thunderstorm than a small basin which tends to reduce the variance of extreme catchment-average precipitation and thus the MAF (Viglione et al., 2010a, b).

I would suggest to add one sentence discussion that there is a transition form convective thunderstorms to long duration stratisform precipitation as catchment size increases (see e.g. Figure 13 in Merz and Blöchl, 2003). This phenomena is also well studied in literature on area reduction factors for extreme precipitation.

Section 2.1 Data: Some more sentences could be added about the data. 1: Are the data from natural catchments not influenced by river regulations ? Do all flood data represent floods caused by rain and/or snow melt, or are there other types of floods like ice jam floods in this dataset ?

Figure 1: You could discuss more if and how your choice of regions influenced the results. I guess that if the aim as to have the best possible predictions of mean annual flood in ungauged basins, you would investigate more in detail how Europe should be divided into sub-regions.

Line 143: Please specify units of Qi and catchment area.

Equation 5: Since you use multi-letter symbols for variables, it is difficult to see where the multiplication sign is located. Either you should use only single-letter symbols, possibly combined with subscripts, or use the multiplication symbol to make the equation easier to read.

Lines 167-169: What is the equation for calculating the radius ?

Line 180-181 Could you be more specific on which variables were log-transformed and why ?

Line 206-207: Probably better to use past tense here.

Table 1: The regional cv is listed in the table, but not commented in the text. I suggest that you add some comments in the text.

Line 255: 'however' could be removed here.

154: is k the same as the radius defined on lines 167-169 ('The length of the vector from the origin is a measure of the variability of the date of occurrence, ranging from 0 (uniformly distributed across the year) to 1 (all events on the same day).' ? Then maybe k could be defined in the method section.

Kemter et al (2020) is missing in the reference list.

Line 313: 'which may mask causal relationship'. Do you think that also spurious correlations might be a challenge?

Figure 5: Maybe one extra point to add: The sign of the correlations listed in Figure 5 might depend on the domain you investigate, and the sign might change between sub-regions of Europe. E.g in the scandinvian countries, it is a negative correlation between elevation and LUF.

Blöschl, G., Hall, J., Parajka, J., Perdigão, R. A., Merz, B., Arheimer, B., ... & Čanjevac, I. (2017). Changing climate shifts timing of European floods. Science, 357(6351), 588-590.

Blöschl, G., Hall, J., et al. (2019) Changing climate both increases and decreases European river floods. Nature, 573(7772), 108-111.

Blöschl, G., Kiss, A., Viglione, A. et al. Current European flood-rich period exceptional compared with past 500 years. Nature 583, 560–566 (2020). https://doi.org/10.1038/s41586-020-2478-3

Hall, J. and G. Blöschl (2018) Spatial patterns and characteristics of flood seasonality in Europe, Hydrology and Earth System Sciences, 22, pp. 3883-3901, https://doi.org/10.5194/hess-22-3883-2018

---

## Author Comment (AC1) · 23 Feb 2021

**"Characteristics and process controls of statistical flood moments in Europe – a data based"**

by D. Lun, A. Viglione, M. Bertola, J. Komma, J. Parajka, P. Valent and G. Blöschl

Here we reproduce all comments of the Referee in *italic characters*, followed by our answers

**Kolbjorn Engeland (Referee)**

*The paper provides a comprehensive analysis of a dataset of annual maximum floods covering all Europe and aims to discuss how process controls can explain the spatial patterns of mean annual floods and the coefficient of variation (CV) of floods. The paper comes in a line of papers analysing floods at a European scale (Blöschl et al., 2017; Hall and Blöschl, 2018; Blöschl et al., 2019 and Blöschl et al., 2020). Whereas the previous papers have investigated trends in time, this paper has a clear focus on the spatial patterns. This provides therefore new knowledge and is complementary to the previous papers. The paper is well written and could in my opinion be published after some minor revisions.*

> We thank Kolbjorn Engeland for the time he spent on our manuscript and for the useful and constructive comments that will help improve the quality of the manuscript. All his comments are reproduced and addressed in the following paragraphs.

*Lines 38:48: This is because a large basin is less likely to be fully covered by a thunderstorm than a small basin which tends to reduce the variance of extreme catchment average precipitation and thus the MAF (Viglione et al., 2010a, b).*

*I would suggest to add one sentence discussion that there is a transition from convective thunderstorms to long duration stratiform precipitation as catchment size increases (see e.g. Figure 13 in Merz and Blöschl, 2003). This phenomena is also well studied in literature on area reduction factors for extreme precipitation.*

> We plan to add the following sentence:

> Convective events, limited in duration and spatial extent, are most relevant for producing floods in small catchments with fast response times (Gaál et al., 2015), whereas long duration stratiform precipitation becomes more relevant as catchment size increases (Merz and Blöschl, 2009).

*Section 2.1 Data: Some more sentences could be added about the data. 1: Are the data from natural catchments not influenced by river regulations ? Do all flood data represent floods caused by rain and/or snow melt, or are there other types of floods like ice jam floods in this dataset ?*

> The data has been described in Blöschl, Hall et al. (2019) and more extensively in Hall, et al. (2015). A sentence will be added, citing these studies and addressing the referee's comment:

The time series were manually checked for strong human modifications such as reservoirs (Blöschl, Hall et al., 2019 and Hall et al., 2015) and include both rain floods and snowmelt floods (Kemter et al., 2020).

*Figure 1: You could discuss more if and how your choice of regions influenced the results. I guess that if the aim as to have the best possible predictions of mean annual flood in ungauged basins, you would investigate more in detail how Europe should be divided into sub-regions.*

This is a very good point. We plan to add the following sentence to section 2.2:

The aim of the partitioning was to represent a small number of contiguous regions that are to some extent hydro-climatologically homogeneous, without considering their effect on predicting flood moments.

We also plan to add the following sentence to section 4.3:

The results depend on the regional partitioning of Europe and will look different for different regions. If the aim of the study was optimal predictive performance of the regional models, the partitioning could be derived based on the data, for example via cluster analysis or regression trees (see e.g. Laaha and Blöschl, 2006).

*Line 143: Please specify units of Qi and catchment area.*

Units will be added in the text.

*Equation 5: Since you use multi-letter symbols for variables, it is difficult to see where the multiplication sign is located. Either you should use only single-letter symbols, possibly combined with subscripts, or use the multiplication symbol to make the equation easier to read.*

Thank you for pointing this out. We feel that using the three-letter abbreviations for the variables makes it easier to follow, so multiplication symbols will be added.

*Lines 167-169: What is the equation for calculating the radius ?*

We plan to add the following text:

It is calculated as the Euclidean distance between the origin and the mean flood date (mean of the sine and cosine of flood dates in polar coordinates), for more details see e.g. Burn (1997). The point (0,1) on the unit circle refers to January 1st, with clockwise rotation representing progress of the year, therefore polar coordinates as described in Breinl et al. (2020) were used.

*Line 180-181 Could you be more specific on which variables were log-transformed and why ?*

A sentence like the following will be added explaining which variables were log-transformed and motivating the choice:

MAF, CV, A and P95 were log-transformed, as their distributions were skewed.

*Line 206-207: Probably better to use past tense here.*

We will use past tense here.

*Table 2: The regional cv is listed in the table, but not commented in the text. I suggest that you add some comments in the text.*

We plan to add the following comment to the text:

The regional coefficients of variation in Table 2 (every other column) reflect the within-region variability of the observed flood moments. They are generally higher for MAF and $\text{MAF}_\alpha$ than for CV, both within individual regions and for all of Europe.

*Line 225: 'however' could be removed here*

We will remove 'however' from the sentence.

*254: is k the same as the radius defined on lines 167-169 ('The length of the vector from the origin is a measure of the variability of the date of occurrence, ranging from 0 (uniformly distributed across the year) to 1 (all events on the same day).' ? Then maybe k could be defined in the method section.*

Yes. A reference on how the radius was calculated will be added to the methods section, where the radius will be introduced as k.

*Kemter et al (2020) is missing in the reference list.*

Thank you very much for pointing this out. The reference will be added.

*Line 313: 'which may mask causal relationship'. Do you think that also spurious correlations might be a challenge?*

Yes, we think spurious correlations also are a challenge when interpreting correlations between flood moments and their process controls. Spurious correlations that are not meaningful and probably occur purely by chance could be the correlations for soil texture (Stex), because they are inconsistent with the existing literature. Alternative covariates, that are representative of the runoff generation processes, such as the HOST classification in the UK (Lilly et al., 1998), could provide a remedy for this. Spurious correlations that arise due to an indirect relationship between attributes are for example those between the fraction of forested area (LUF) and MAF, given that densely forested areas tend to be high elevation regions with higher rainfall depths (Lines 603-609). We believe we have already addressed these issues in the paper.

*Figure 5: Maybe one extra point to add: The sign of the correlations listed in Figure 5 might depend on the domain you investigate, and the sign might change between sub-regions of Europe. E.g in the Scandinavian countries, it is a negative correlation between elevation and LUF.*

We have analyzed the correlations among attributes also for all regions separately and indeed they vary between regions. However, we have chosen not to add them for space reasons.

**References:**

Blöschl, G., Hall, J., Parajka, J., Perdigão, R. A., Merz, B., Arheimer, B., ... & Canjevac, ˇ I. (2017). Changing climate shifts timing of European floods. Science, 357(6351), 588-590.

Blöschl, G., Hall, J., et al. (2019) Changing climate both increases and decreases European river floods. Nature, 573(7772), 108-111.

Blöschl, G., Kiss, A., Viglione, A. et al. Current European flood-rich period exceptional compared with past 500 years. Nature 583, 560–566 (2020). https://doi.org/10.1038/s41586-020-2478-3

Breinl, K., Di Baldassarre, G., Mazzoleni, M., Lun, D., & Vico, G. (2020). Extreme dry and wet spells face changes in their duration and timing. Environmental Research Letters, 15(7), 074040.

Burn, D. H. (1997). Catchment similarity for regional flood frequency analysis using seasonality measures. Journal of hydrology, 202(1-4), 212-230.

Gaál, L., Szolgay, J., Kohnová, S., Hlavčová, K., Parajka, J., Viglione, A., ... & Blöschl, G. (2015). Dependence between flood peaks and volumes: a case study on climate and hydrological controls. Hydrological Sciences Journal, 60(6), 968-984.

Hall, J., Arheimer, B., Aronica, G. T., Bilibashi, A., Boháč, M., Bonacci, O., ... & Blöschl, G. (2015). A European Flood Database: facilitating comprehensive flood research beyond administrative boundaries. Proceedings of the International Association of Hydrological Sciences, 370, 89-95.

Hall, J. and G. Blöschl (2018) Spatial patterns and characteristics of flood seasonality in Europe, Hydrology and Earth System Sciences, 22, pp. 3883-3901, https://doi.org/10.5194/hess-22-3883-2018

Kemter, M., Merz, B., Marwan, N., Vorogushyn, S., & Blöschl, G. (2020). Joint trends in flood magnitudes and spatial extents across Europe. Geophysical Research Letters, 47(7), e2020GL087464.

Laaha, G., & Blöschl, G. (2006). A comparison of low flow regionalisation methods—catchment grouping. Journal of Hydrology, 323(1-4), 193-214.

Lilly, A., Boorman, D. B., & Hollis, J. M. (1998). The development of a hydrological classification of UK soils and the inherent scale changes. In Soil and Water Quality at Different Scales (pp. 299-302). Springer, Dordrecht.

Merz, R., & Blöschl, G. (2003). A process typology of regional floods. Water Resources Research, 39(12).

Merz, R., & Blöschl, G. (2009). Process controls on the statistical flood moments-a data based analysis. Hydrological Processes: An International Journal, 23(5), 675-696.

Viglione, A., Chirico, G. B., Woods, R., & Blöschl, G. (2010a). Generalised synthesis of space–time variability in flood response: An analytical framework. Journal of Hydrology, 394(1-2), 198-212.

Viglione, A., Chirico, G. B., Komma, J., Woods, R., Borga, M., & Blöschl, G. (2010b). Quantifying space-time dynamics of flood event types. Journal of Hydrology, 394(1-2), 213-229.

---

## Referee Comment (RC2) · Anonymous Referee #2 · 26 Feb 2021

This manuscript has the potential to serve as a strong reference for characterizing the spatial variability of annual peak-flow moments at sites without strong anthropogenic modifications, such as reservoirs, throughout Europe. The leave-one-out cross-validation of a multiple regression model predicting flood moments (mean, Cv, Cs) suggests that, with follow-up efforts, this work could be used to estimate flood moments at ungaged locations with reasonable accuracy in many locations. This work also documents large-scale spatial patterns in controls on flood moments throughout the continent, although the process controls revealed are not especially surprising to people with knowledge of European hydrology. However, numerous technical and presentation improvements detailed below are needed to make this manuscript publish-

[Figure]

able in HESS. In addition, a more compelling case for how this research could benefit both stationary and nonstationary flood-frequency analysis would be helpful. I have also attached a Tracked Changes Word document with more specific writing and presentation suggestions and some more minor technical inquiries.

SOME BASIC CHARACTERISTICS OF THE FLOOD TIME SERIES NEED TO BE CLARIFIED UPFRONT. The authors should state in their abstract whether their set of 2,370 flood series are from stations in anthropogenically impacted basins and whether the "maximum annual flows" they analyze are daily mean flows or instantaneous peak flows. This is important given the small drainage areas of some basins. The authors state that they used the version of the European Flood Database used in Blöschl et al. (2019), which excluded catchments with strong human modifications, such as reservoirs, but did not exclude basins subject to more local anthropogenic perturbations – given their focus on elucidating broad regional patterns. While this dataset contains both [instantaneous?] peak flows and maximum daily mean flows in each year, it seems like the authors might have strictly used peak flows based on descriptions at the beginning of Section 2: "This study uses the data set of European flood discharges of Blöschl, Hall et al. (2019), which . . . consists of 2370 annual maximum peak discharge series from 33 countries". Also, the authors only used 2,370 stations whereas Blöschl et al. (2019) used 3,783. This discrepancy should be explained briefly. Finally, the authors should clarify earlier in the manuscript whether they used calendar years or a designated water year when identifying annual peaks.

MOMENT ESTIMATION BIASES MUST BE ADDRESSED. The authors need to discuss the bias in their estimates of the Cv (coefficient of variation) and Cs (coefficient of skewness). First, with regards to the Cv, Ye et al. (2020) demonstrated the extent to which common Cv estimators can be biased when data are skewed or do not adhere to i.i.d. assumptions. While the degree of bias is not as pronounced as it is with daily flow data, quick calculations using the equations described in this paper demonstrate that Cv of annual peak flows can have a substantial bias. Numerous references have

also demonstrated the bias of skewness estimates from small samples, including their dependence on record length (Wallis et al., 1974; Bobee and Robitaille, 1975; Carney, 2016). In their discussion, the authors should also recognize the literature on regional skewness coefficients as well as the weighted skewness approaches combining at-site and regional information that the U.S. Geological Survey employs.

NONSTATIONARITY AND ITS POTENTIAL IMPACTS ON MOMENTS MUST BE CONSIDERED. Blöschl et al. (2019) reported regional-scale climate-driven trends in northwestern, southern, and eastern Europe (see Fig. 1). Is it worthwhile to describe the sample moments of sites without considering these changes? In my opinion, the authors should either develop a procedure to exclude sites subject to trends or provide a rationale for treating all sites as stationary given their research goals. In making this decision, the authors should consider the ongoing shift from nival to pluvial regimes in 3/5 regions in Europe makes this an important consideration. If they wish to distinguish trends from long-term persistence, an argument often used to refute nonstationary treatments of hydrologic records, the authors could test for trends of a given trajectory against null hypotheses of long-term persistence (see Matalas and Sankarasubramanian, 2003; Cohn et al., 2005). The authors should also note trends in both the mean and variance affect both Cv and Cs estimates [see Serago and Vogel (2018) for some initial guidance for making these adjustments. Hecht and Vogel (2020) offer one approach for modeling trends in variability and reference a handful of other moment-based ones, including Strupczewski et al. (2001).

THE RESIDUAL BEHAVIOR OF THE REGRESSION MODELS MUST BE EVALUATED. The authors do not report the normality, heteroscedasticity, autocorrelation of their residuals. They also do not report the variance inflation factor or alternatives measures of multicollinearity for their multiple regression equations. This is especially important if one is making process-based inferences using covariate matrix-derived statistics from regression models. The authors should consider using a Supporting Information (SI) section to display the residual behavior of their models. Also, the authors

report the tendency for large MAFs to be underestimated and small ones to be over-estimated. This suggests that residuals might not be homoscedastic and that another covariate may be needed to produce a multiple regression model that meets the homoscedasticity requirement for making inferences from standard error-based metrics (Helsel et al., 2020).

THE CHOICE BETWEEN LOG-SPACE VS. REAL-SPACE MOMENTS SHOULD BE RECOGNIZED. The authors should also recognize in their manuscript that moments of log-transformed floods are often used in FFA and clarify that real-space moments are used upfront.

MIXED POPULATIONS SHOULD BE CONSIDERED IN THEIR INTERPRETATION OF RESULTS. While the authors somewhat recognize mixed populations (e.g. description of Alps and Norwegian coast flood-generating processes), they compute moments assuming floods at each site belong to homogenous populations. While statistically evaluating the presence of mixed populations at individual sites lies beyond the scope of this paper, it is important to consider mixed populations explicitly when interpreting results and to caution readers about problems associated with choices to neglect them at individual sites. While the authors use an analysis of flood timing to help identify drivers of floods, they do not specifically check for the presence of multi-modal peaks suggesting mixed distributions in them. This type of quicker analysis could support some of the good observations that authors make about mixed distributions in specific regions. Finally, the authors should communicate an awareness of this 'mixed populations' literature in their discussion of mixed populations.

MORE DETAILS ABOUT THE DRAINAGE AREA-NORMALIZED EQUATION(S) ARE NEEDED. I like the authors' idea of normalizing their analysis to a given drainage area (100 km2) since drainage area is still an important descriptor of flood-generating processes even when specific discharge values are used to express peak flows. However, it would be nice to report goodness-of-fit measures for this model and show the fit graphically, the latter which can be done in the SI section if space constraints remain.

[Figure]

The authors also describe the creation of equations that establish values of the Cv and Cs for 100-km2 drainage areas, but it is unclear if these DA-adjusted values are ever evaluated as response variables in the multiple regression models.

NON-MONOTONIC RELATIONSHIPS WITH COVARIATES SHOULD BE CHCKED - AT LEAST IN AN EXPLORATORY DATA ANALYSIS. The authors raise the possibility of non-monotonic relationships between moments and catchment descriptors in discussions of prior findings, but they only examine monotonic relationships in their linear regression models. In particular, they cite Smith et al. (1992), who found that floods in the Appalachian mountains in the eastern US demonstrated an increase in the CV with catchment area for catchments up to 100 km2 and then exhibited a decrease with catchment area in larger basins. They also recognize that Wang et al. (2017) found a non-monotonic relation between water body size and the Cv. In addition, Pallard et al. (2009) also found that Cv decreases with drainage density in catchments with sparse drainage networks but then increase after a reaching a minimum. I think that if the authors can claim that exploratory data analyses did not demonstrate any non-monotonic trajectories like these, they don't need to formally test hypotheses of non-monotonic change with statistical models, but they should briefly demonstrate that they performed exploratory data analysis (EDA) justifying the monotonic relationships they modeled.

DOES ARIDITY CAUSE FLOOD VARIABILITY? The authors make an important association between the aridity index (AI) and the Cv of annual floods. However, it is important to recognize more succinctly that arid regions tend to have greater interannual precipitation variability, and, for that reason, arid basins tend to have larger Cv's. This is important when considering the implications of these findings under climate change. If a region becomes drier, it's interannual precipitation variability will not necessarily increase. A discussion about the implications of these cross-sectional findings for projecting flood responses of environmental changes at a given location over time would enrich the paper.

ORDINARY KRIGING. Ordinary kriging visualizes broad regional patterns but may be

limited for applications in ungaged basins. The kriging results look visually pleasing and achieve the goal of illustrating broad regional patterns in flood moments. However, what if nearby basins have greatly different drainage areas (since this is stated to be a map of MAF and not MAF[alpha]) or pronounced differences in other catchment characteristics that can change abruptly? In the future, the authors could consider kriging in attribute space instead of geographic space. If the authors retain these kriged maps to display broad regional patterns, they should note the limitations of using these interpolations for characterizing flood regimes at ungaged sites. To me, it seems like the regression equations should work reasonably well for estimating moments at many sites. And if they choose to argue that kriging can be used to estimate moments in ungaged basins, then a more formal cross-validation analysis and more detailed reporting of model performance is necessary. Alternatively, they could turn this kriging exercise into a separate paper.

IMPORTANCE FOR FFA IN PRACTICE. This paper successfully elucidates broad regional patterns in flood moments across Europe. Their leave-one-out cross-validation suggests that flood moments can be reasonably estimated in many regions at sites whose covariate values are known. The implications of these errors for design flood estimates could be made stronger by computing the design floods with a GEV quantile function (noting issues with this distribution in specific regions from prior studies, such as Salinas et al. (2014)) using moments estimated from observations and from the multiple regression models. The authors should also address practical concerns regarding nonstationarity described above. In addition, the authors should note the contribution that their study makes to improve upon other recent prediction in ungaged basins efforts in Europe.

OVERALL PRESENTATION. The paper reads a bit like a lab report in places and generally has the potential to be shortened considerably without losing much content. In some places, starting paragraphs with more topic sentences could help orient the reader better and curtail the 'rambling' nature of some sections, such as the bivariate correlation results. The correlation analysis is important for interpreting regression model results and many of the insights on multicollinearity in the data are good, but the presentation of it should be a bit more focused on supporting the multiple regression model analysis and not a comprehensive review of the entire correlation matrix. The submission also requires more editing for fluidity/conciseness and proper punctuation. While I made some writing and grammar suggestions in the Track Changes document, I did not perform a comprehensive check for these issues and suggest that the authors find someone else who can do that.

REFERENCES. This list contains references not already cited in the submitted manuscript:

Bobee, B., R. Robiataille (1975) Correction of bias in the estimation of the coefficient of skewness, Water Resour. Res., doi: 10.1029/WR011i006p00851

Carney, M.C. (2016), Bias correction to GEV shape parameters used to predict precipitation extremes, doi: 10.1061/(ASCE)HE.1943-5584.0001416

Cohn, T.A., H.F. Lins (2005), Nature's style: naturally trendy, Geophys. Res. Lett., 32, L23402, doi:1029/2005GL024476

Hecht, J.S., R.M. Vogel (2020), Updating urban design floods for changes in central tendency and variability using regression, doi:10.1016/j.advwatres.2019.103484

Helsel, D.R., Hirsch, R.M., Ryberg, K.R., Archfield, S.A., and Gilroy, E.J., 2020, Statistical methods in water resources: U.S. Geological Survey Techniques and Methods, book 4, chapter A3, 458 p., https://doi.org/10.3133/tm4a3. [Supersedes USGS Techniques of Water-Resources Investigations, book 4, chapter A3, version 1.1.]

Matalas, N.C., A. Sankarasubramanian (2003), Effect of persistence on trend detection via regression, Water Resour. Res., doi:10.1029/2003WR002292.

Pallard, B., A. Castellarin, and A. Montanari (2009), A look at the links between drainage density and flood statistics, Hydrol. Earth Syst. Sci., 13, 1019-1029,

doi:10.5194/hess-13-1019-2009

Salinas, J.L., A. Castellarin, A. Viglione, S. Kohnova, and T.R. Kjeldsen (2014), Regional parent flood frequency distributions in Europe – part 1: is the Gev model suitable as a pan-European parent? Hydrol. Earth Syst. Sci., 18, 4381-4389, doi: 10.5194/hess-18-4381-2014

Serago, J.M., R.M. Vogel (2018), Parsimonious nonstationary flood frequency analysis, Adv. Water Resour., 112, 1-16, doi: 10.1016/j.advwatres.2017.11.026

Strupczewski, W., Z. Kaczmarek (2001), Non-stationary approach to at-site flood frequency modelling II. Weighted least squares estimation, J. Hydrol., 248(1-4), 143-151, doi: 10.1016/S0022-1694(01)00398-5

Wallis, J.R., N.C. Matalas, J.R. Slack (1974) Just a moment! Water Resour. Res., doi: 10.1029/WR010i002p00211

Ye, L., X. Gu, D. Wang, and R.M. Vogel. An unbiased estimator of coefficient of variation of streamflow. J. Hydrol., 594, doi: 10.1016/j.jhydrol.2021.125954

I wish the authors good luck with their revision and am happy to discuss any of these issues!

Please also note the supplement to this comment:
https://hess.copernicus.org/preprints/hess-2020-600/hess-2020-600-RC2-supplement.pdf

**Supplement:**

[revised manuscript text omitted]

Commented [A9]: Storm events in general? Not always T-storms.

Commented [A10]: See Pallard et al. (2009) on the effect that drainage density has on the Cv
https://hess.copernicus.org/articles/13/1019/2009/hess-13-1019-2009.pdf

Commented [A11]: Floods from synoptic-scale precip events (e.g. frontal systems)?

Commented [A12]: Many USGS regional flood frequency studies based on observed data have revealed non-climatic controls.

Commented [A13]: Attributing?

Commented [A14]: Consider stating this one first

Commented [A15]: Also, MAP might be better than event precip in places with snowmelt-driven floods

Commented [A16]: Is it aridity itself or does this stem from the tendency of more arid catchments to have greater interannual precipitation variability?

and CV in the lowlands of Austria, which they interpreted in terms of the increasing non-linearity of the rainfall-runoff process

70 with aridity. Iacobellis et al. (2002), found that CV behaviour is controlled mainly by the long-term climate and the abstraction characteristics at the catchment scale. While a decrease in CV with the catchment area is attributed to more arid and impermeable catchments where the infiltration excess (Horton type) mechanism dominates, as in arid and impermeable basins, an increase of CV with the catchment size area is attributed to humid and vegetated catchments with dominant saturation excess runoff generation.

75 The cClimate controls, including rainfall, soil moisture and snowmelt are usually subject to strong seasonality. An analysis of the seasonality of floods (Bayliss and Jones, 1993; Merz and Blöschl, 2003) has therefore been an efficient way to shed light on the interaction of these processes. For example, based on a the seasonality of 4262 catchments in Europe, Blöschl et al. (2017) and Hall et al. (2018) identified extreme winter precipitation in northwestern Europe, snowmelt in northeastern and eastern Europe and summer precipitation and snowmelt in

80 the Alpine regions of Europe to be important flood drivers. Using their data set, Berghuijs et al. (2019) found soil moisture excess to explain flood seasonality better than other variables, such as extreme precipitation, particularly in the western part of Europe, in line with a similar study in the United States (Berghuijs et al., 2016).

While substantial understanding of regional flood controls has been achieved in the past, few studies have

85 analysed large, consistent data sets of flood discharges in terms of their statistical moments. Large data sets provide the opportunity ofto obtaining more robust and generalisable findings than studies containing a smaller number of catchments. The aim of this paper is therefore to identify the continental scale patterns of flood moments and their controls across Europe. We use a data set of flood discharges of 2370 catchments across

Europe for the period 1960-2010 and apply correlation and regression analyses to identify climatic and 90 catchment process controls on the moments.

**2. Data and Methodology**

**2.1 Data**

This study uses the data set of European flood discharges of Blöschl, Hall et al. (2019), which can be found in 95 their supplementary material. It consists of 2370 annual maximum peak discharge series from 33 countries. Catchment areas range from 5 to 100,000 km$^2$, with a median of 383 km². The observation period is 1960 to 2010, and record lengths range from 30 to 51 years, with a median of 51 years.

In order to analyse process controls, a range of catchment attributes and climatic indicators are used. In addition to catchment area, we used catchment-averaged climate indicators, including long-term mean annual

100 precipitation (MAP), long-term mean potential evaporation (PET) and the aridity index, PET/MAP. Extreme precipitation is quantified by the daily rainfall rate that is not exceeded in 95% of the days, and the long-term mean of the maximum 2-day precipitation of each year. As a proxy for snowmelt we used the mean air temperature in spring (MAM) and winter (DJF). Soil moisture was taken from the CPC Soil moisture database, which contains model-calculated soil moisture values. We used the mean of the annual maximum monthly

**Commented [A17]:** The greater the PET, the higher the Cv in lowlands of Austria?

**Commented [A18]:** This sounds very interesting, but could you explain in a sentence or two why Cv becomes lower with higher DA's in basins where infiltration-excess flow dominates and why it becomes higher with DA in basins where saturation-excess flow predominates?

**Commented [A19]:** Please describe the degree to which and the types of anthropogenic perturbations to which basins in your sample are subject.

**Commented [A20]:** Assuming instantaneous peaks?

**Commented [A21]:** Total days or wet days?

**Commented [A22]:** The duration of precipitation to examine varies substantially by region and catchment size. Can you convey an awareness of this in your introduction of these covariates?

**Commented [A23]:** How accurate are these modeled values? Can you add a sentence or so stating this and any places where inaccurate estimates may distort your analyses?

105 values over the observation period. Topographical indicators include the mean catchment elevation and the mean topographical slope of each catchment. Land use was quantified as a percentage of total catchment area. Soil characteristics were quantified in terms of five soil- texture  categories. The data used are summarised in Table 1.

110 Table 1: Data used in this study including quantiles of the variables and source information. Sources:
[1]Data base on European floods: https://github.com/tuwhydro/europe_floods
[2]CCM River and Catchment Database. Vogt et a., 2017. https://data.europa.eu/
[3]E-OBS gridded dataset (v18.0e), 0.1 deg. Cornes at al., 2018. https://www.ecad.eu/
[4]Global Aridity Index and Potential Evapo-Transpiration (ET0) Climate Database V2. Trabucco and Zomer, 2019 115 https://figshare.com/articles/Global_Aridity_Index_and_Potential_Evapotranspiration_ET0_Climate_Database_v2/7504448/3
[5]CPC Soil Moisture (V2), NOAA Climate Prediction Center. Fan and Dool, 2004. https://www.cpc.ncep.noaa.gov/
[6]Global Multi-resolution Terrain Elevation Data GMTED2010, 7.5 arc-seconds. https://www.usgs.gov/
[7]Corine Land Cover (CLC) 2000, Version 20b2, https://land.copernicus.eu/

| Units | 25 / 50 / 75% quantiles | Source |
|---|---|---|
| $m^3 s^{-1}$ | 0.06/0.11/0.22 $km^{-2}$ | Data base on European floods[1] |
| $m^3 s^{-1}$ | 0.08/0.16/0.28 $km^{-2}$ | Data base on European floods[1] |
| - | 0.36/0.46/0.61 | Data base on European floods[1] |
| - | 0.62/1.09/1.69 | Data base on European floods[1] |
| $km^2$ | 135.90/382.80/1264.80 | CCM River and Catchment Database[2] |
| $mm\ yr^{-1}$ | 621.28/798.69/1057.76 | EOBS[3] |
| $mm\ d^{-1}$ | 8.40/10.54/13.45 | EOBS[3] |
| $mm\ d^{-1}$ | 18.00/22.45/28.51 | EOBS[3] |
| □C | 5.03/7.15/8.42 | EOBS[3] |
| □C | -3.35/-1.16/1.05 | EOBS[3] |
| mm | 368.38/424.93/507.20 | CPC Soil Moisture (V2)[5] |
| $mm\ yr^{-1}$ | 749.02/817.73/897.98 | Global Aridity Index and Potential Evapo-Transpiration (ET0) Climate Database V2[4] |

120 | [8]European Soil DataBase (ESDB), Soil Geographical DataBase (SGDB), unitless 0.72/1.00/1.25 TEXT_SRF_DOM, 10x10km. Panagos et al., 2012. https://esdac.jrc.ec.europa.eu

| Variable group | Symbol | Data Description | Unit | Value | Data source |
|---|---|---|---|---|---|
| Flood Moments | MAF | Mean annual specific flood | m a.s.l. | 199.04/472.58/833.12 | Global Aridity Index and Potential Evapo-Transpiration (ET0) Climate Database V2[4] GMTED2010[6] |
| | $MAF_{\alpha}$ | Mean annual specific flood normalized to catchment area of $\alpha = 100 km^2$ | unitless | 0.03/0.08/0.18 | GMTED2010[6] |
| | CV | Coefficient of variation of annual maximum flood peaks | % | 31.66/54.59/79.62 | CORINE[7] |
| | CS | Coefficient of skewness of annual maximum flood peaks | % | 0/0.05/0.57 | CORINE[7] |
| Catchment Area | A | Catchment area | class | 1.77/2.00/2.25 | ESDB[8] |
| Precipitation | MAP | Long-term mean annual precipitation | | | |
| | P95 | Daily precipitation rate, that is higher than what is observed on 95% of days in the observed period | | | |
| | Pmax | Mean of maximum of 2-day precipitation of each year | | | |
| Air temperature | Tspr | Mean daily temperature in MAM (Celsius) | | | |
| | Twin | Mean daily temperature in DJF (Celsius) | | | |
| Soil moisture | SM | Mean of annual maximum monthly soil moisture | | | |
| Evaporation | PET | Long-term mean potential evaporation | | | |
| Aridity | AI | Aridity index (PET/MAP) | | | |
| Topography | Elev | Mean catchment elevations | | | |
| | Slope | Mean topographic slope (mean of tangent of angle of slope) | | | |
| Land use | LUF | Fraction of catchment area covered by forest and seminatural areas | | | |
| | LUW | Fraction of catchment area covered by water bodies | | | |
| Soils | Stex | Dominant surface textural class of the STU (Soil Typological Unit), mean value of categories (1=coarse, 5=fine) | | | |

**2.2 Hydroclimatic regions**

For the statistical analyses, Europe was subdivided into five regions based on  the eleven biogeographic regions of Roekaerts (2002) and guided by the flood seasonalities of Blöschl et al. (2017) and 125 Hall et al. (2018). In the Northeastern region, floods are mainly due to snowmelt during spring and early summer. The Atlantic region is characterised by mild, wet winters and cool, humid summers; floods mainly occur in winter following rain events. The Central-Eastern region has a continental climate with cold winters and warm summers and floods mainly occur in spring with snow-melt contributions. The Alpine region comprises  the Alps and the Carpathians, where floods mainly occur in summer due to summer storms and/or snow melt. The Mediterranean region is

130 characterised by hot, dry summers and mild, wet winters; floods occur in autumn and winter. For simplicity, each catchment was allocated to one of the regions according to the location of its stream gauge.

> **Commented [A27]:** What about lower-altitude streamgages draining primarily alpine catchments? How often is this is an issue?  Generally, it looks like you have a reasonable alpine region

> **Commented [A28]:** Make NE region color standout more. Its shade of blue is too similar to the dark green lowlands and plentiful blue lakes of this region

[Figure]

Figure 1: Location of the 2370 hydrometric stations analyzed. Colours of dots indicate five hydro-climatic regions (Northeastern, Atlantic, Central-Eastern, Alpine, Mediterranean). Background colour is elevation (m a.s.l.).

135

**2.3 Analysis method**

The statistical flood moments, the specific mean annual flood (MAF), the coefficient of variation (CV) and the coefficient of skewness (CS), were estimated from the annual maximum peak discharges series by:

$$MAF = \frac{1}{n}\sum\nolimits_{i=1}^{n} Q_i \tag{1}$$

$$S^2 = \frac{1}{n-1}\sum\nolimits_{i=1}^{n}(Q_i - MAF)^2 \tag{2}$$

$$CV = \frac{S}{MAF} \tag{3}$$

$$CS = \frac{n\sum_{i=1}^{n}(Q_i - MAF)^3}{(n-1)(n-2)S^3} \tag{4}$$

where $Q_i$ is the annual maximum peak discharge of a record in year $i$, divided by the catchment area. Estimation uncertainty of the statistical flood moments decreases with record length and increases with the moment order.

While the estimation uncertainty of the mean is small, the uncertainty of CS can be substantial. For a record length of 50 years and a series with the average estimated moments of the entire dataset (MAF=0.17 m³ s⁻¹ km⁻², CV=0.52, CS=1.28), the standard error of the CS estimate is $\square_{CS}$= 0.56, which is about half of the underlying true moment (assuming a GEV-distribution as the data-generating process). The estimation uncertainties need to be accounted for when interpreting the process controls on the flood moments. Since the specific mean annual

flood is often strongly controlled by catchment area which may mask other controls that vary regionally, we  also considered the specific mean annual flood, $MAF_\alpha$, normalised to a catchment area of $\alpha$=100km²

$$MAF_\alpha = MAF A^{-\beta_{MAF}} \alpha^{\beta_{MAF}} \tag{5}$$

$MAF_\alpha$ and $\beta_{MAF}$ were found by ordinary least squares regression.

First, we estimated what fraction of the spatial variability of the estimated flood moments can be explained by subdividing Europe into five regions (Figure 1), using a simple one-way analysis of variance (ANOVA). This can be interpreted as a simple regression model, where the dependent variables are the estimated flood moments and the only explanatory variables are indicators corresponding to the regional partition. The coefficient of determination of this model corresponds to the fraction of variance explained by the partition over the total variance in estimates of the flood moments.

Second, we evaluated the role of catchment area, since it is almost always the main control on the mean annual flood when examining a sample of catchments varying by orders of magnitude, and it reflects the aggregation behaviour of the floods and  their climatic and catchment controls. Specifically we estimated the dependence of MAF, CV and CS on catchment area in a double logarithmic relationship from Eq. (5) and analogous equations for CV and CS.

Third, we conducted a seasonality analysis to assist in the process interpretations. We represented the

date of occurrence, $D$, of the maximum annual flood as a number from 1 to 365 (Julian dates) in polar coordinates on a unit

circle with angle $\theta = D \frac{2\pi}{365}$. For a flood series, the direction $\theta$ of the average vector from the origin indicates the

**Commented [A29]:** Is there a reference describing the computations you made?

**Commented [A30]:** In your discussion, consider commenting on Salinas et al. (2014) who investigated how well the GEV distribution performed throughout Europe

https://www.researchgate.net/publication/267865581_Regional_parent_flood_frequency_distributions_in_Europe_-_Part_1_Is_the_GEV_model_suitable_as_a_pan-European_parent

**Commented [A31]:** Transforming both sides logarithmically before fitting? How well did this equation fit the data? Please present goodness-of-fit stats and a graphic in the SI.

**Commented [A32]:** Log-log? Where both the independent and dependent variables are log-transformed?

**Commented [A33]:** Nice. Try to emphasize this throughout the paper a bit more.

[revised manuscript text omitted]

**Commented [A38]:** Good. Can you mention earlier that you log-transformed the moment values after computing them in real-space? (At least this is what it sounds like you did)

**Commented [A39]:** Are subscripts missing here or is this a pdf conversion issue?

**Commented [A40]:** The kriging results look visually pleasing and reveal broad spatial patterns, but what if nearby basins have greatly different drainage areas or pronounced diffs in other catchment characteristics that can change abruptly? In the future, you could consider kriging in attribute space instead if all your covariates can be gridded. In this paper (or another one), you should consider stating the limitations of this kriging analysis and perform a split-sample validation experiment to see how well it does at ungaged locations. You could also take this out and make it a separate paper.

**Commented [A41]:** Consider showing histograms

**Commented [A42]:** In other words, most of the variability lies within the regions and not between them.

**Commented [A43]:** Something to think about: how much of the partitioning would the regions explain if you considered of basins different sizes separately?

**Commented [A44]:** Does this mean that you have larger catchments in some regions than others?

[revised manuscript text omitted]

**Commented [A48]:** Is this variability mainly due to snow cover, melt timing or also rain-on-snow events?

**Commented [A49]:** Is their CV large due to interannual snowpack variability? Timing issues with spring thaws? Mixture of spring rains and snowmelt?

**Commented [A50]:** Snow or rain-driven? Or both?

[Figure]

Figure 3: Seasonality of annual flood peaks. Position in circle indicates mean date of occurrence (angle) and variability of the date (inverse of distance from centre). Each point represents one catchment. Colours of the points indicate flood moments as
285 in Figure 2. Small red circles highlight subregions referred to in the text (1: Norwegian coast, 2: Northwestern Russia, 3: Western Russia, 4: Western UK and Southwestern Norway, 5: Southern Germany and North of Czech Republic, 6: Parts of Poland and Ukraine, 7: Southern Austria and Northern Slovenia, 8: Alpine and Carpathian midlands, 9: Alpine region, 10: Slovenia and Southern France, 11: Balkans)

**3.3 Scaling of flood moments with catchment area**

The first control on the flood moments examined here is catchment area, as it is often the dominant and best understood control (Table 3, Fig. 4). The highest decrease in the mean annual flood (MAF) with catchment area

occurs in the Mediterranean and the Alpine region with $\square_{MAF}$ =-0.255 and -0.208, respectively, while the weakest the smallest decreases occur in the Northeastern and Central Eastern regions ($\square_{MAF}$ =-0.163 and -0.108) where

295 snow-melt is important. The coefficient of variation (CV) of the flood records (CV) decreases with catchment area in most regions. Again, the strongest decrease occurs in the Mediterranean while in the Northeastern and Central Eastern there is no significant relationship. There are few small catchments in the Central-Eastern region, which may make the regression with area less robust. Overall, there is a tendency for CS to decrease with catchment area and the strongest decrease occurs again in the Mediterranean.

> **Commented [A53]:** Good obs
>
> **Commented [A54]:** Could the paucity of small basins in Central-Eastern Europe contribute to the positive relation between Cs and basin area there?

300

Table 3: Dependence of the flood moments with catchment area in a double logarithmic relationship Eq. (5) and analogous equations for CV; and a semi logarithmic relationship for CS, i.e. $CS = \log A\ \beta cs$. Last lines show the 5% and 95% quantiles of catchment area (km²). * indicates statistical significance (two-sided t-test) at the 5% level.

|  | Europe | Northeastern | Atlantic | Central-Eastern | Alpine | Mediterranean |
|---|---|---|---|---|---|---|
| $\square_{MAF}$ | -0.245* | -0.163* | -0.184* | -0.108* | -0.208* | -0.255* |
| $\square_{CV}$ | -0.030* | -0.015 | -0.042* | 0.025 | -0.020* | -0.072* |
| $\square_{CS}$ | -0.133* | -0.054 | -0.124* | 0.280* | -0.177* | -0.232* |
| 5% / 95% quantiles of area (km²) | 35/11500 | 28/18010 | 37/5331 | 142/32509 | 26/4212 | 47/27251 |

305

[Figure]

Figure 4: Mean annual specific flood (MAF), normalised mean annual specific flood (MAF$_a$), coefficient of variation (CV) and coefficient of skewness (CS) plotted against catchment area (km²). Colours indicate region. Lines are regression lines for each of the regions.

310

**3.4 Individual controls on flood moments**

When interpreting the association of climate and catchment attributes with flood characteristics, it is important to account for the correlation between the attributes themselves, which may mask causal relationships. Spearman correlation coefficients have therefore been estimated among all explanatory variables (Figure 5). The largest

315 correlations occur among the precipitation characteristics, all of which are at least $r=0.86$. The correlation between long-term mean precipitation (MAP) and daily precipitation not -exceeded 95% of the time (P95) is even 0.96, indicating that the spatial patterns of these two variables in Europe are almost identical. The correlation between soil moisture (SM) and the precipitation variables is at least 0.78, and the correlation between the aridity index (AI) and the precipitation variables varies between -0.63 and 0.84. The latter may be partly related to the

320 fact that AI is the ratio of potential evaporation (PET) and MAP. PET is related to spring temperature ($r=0.73$). Elevation and slope are closely related to each other ($r=0.88$) and they are also closely related to the precipitation variables with $r$ of at least 0.56, reflecting orographic influences on precipitation. Forest cover (LUF) is related to elevation and slope ($r=0.67$ and 0.72, respectively) reflecting the presence of forest in mountain areas. The positive correlation between lake area fraction (LUW) and catchment area ($r=0.45$) results from a tendency for

325 large catchments to contain lowlands where lakes are more frequent than in the mountains, and the positive correlation between soil type (Stex) and spring temperature (Tspr) ($r=0.37$) is due to coarse soils prevailing in the (colder) north of Europe. Fig. 5 also shows 2d histograms of the variables as well as their kernel density estimates.

**Commented [A55]:** 95th percentile daily precipitation

**Commented [A56]:** CHECK correlation between precipitation and forest cover

[Figure]

**Commented [A57]:** In which region are the largest basins?

[revised manuscript text omitted]

**Commented [A65]:** Not normalized by area at all?

**Commented [A66]:** Is it worth including so much about Cs then?

**Commented [A67]:** What is the water balance group? Look at interaction effects?

**Commented [A68]:** Perhaps a proxy variable for snowmelt-driven floods is needed there

Also, do smaller headwater basins in the Alpine regioun have more snow cover and, if so, do they generate larger floods than larger ones with less snow cover?

**Commented [A69]:** Good interpretation

[Figure]

Figure 7: Results of dominance analysis for regional regression models for MAF. Panels depict the average contributions (normalised general dominance measure, nGDM) of the covariates included in the regressions (log of catchment area, log of extreme precipitation index P95, mean winter temperature and aridity index). Plus- and Minus-signs indicate sign of the regression coefficients.

For CV (Figure 8, Table A.1.4), in the Northeastern region the Aridity index (AI) is the most important covariate by far. This is due to Scandinavia being much wetter than  northwestern Russia translating into lower CVs.

In the Atlantic region AI is the most important variable for explaining the spatial variability of CV although the overall explanatory power of the regional model is rather low (R² of 0.27, Table A.1.2). The smaller CV closer to the ocean in the Atlantic region is partly explained by higher winter temperatures.

Commented [A70]: Please put all regional R^2 in parentheses as you describe them in the text. Also, it might help to put this table in the main text.

Check for other instances of this throughout the manuscript.

Commented [A71]: Please tell readers how higher winter temperatures lead to lower Cv's here.

In the Central-Eastern region AI dominates again with higher CV in the Ukraine correlated with higher aridity 430 than further in the West, both due to higher PET and lower MAP.

The Alpine region is an exception in that P95 explains more of the spatial variability of CV than AI (nGDM of 0.54 and 0.35, respectively). This is because PET is negatively related to elevation but the flood magnitudes are controlled by the higher orographic rainfall on the windward (NW) side of the Alps.

In the Mediterranean both aridity and P95 are important predictors. For example, low aridity (because of high 435 MAP) in Croatia and Slovenia is associated with low CV, and high P95 in Southern France is associated with moderately low CV.

Figure 8: Results of dominance analysis for regional regression models for CV. Panels depict the average contributions

(normalised general dominance measure, nGDM) of the covariates included in the regressions (log of catchment area, log of
440 extreme precipitation index P95, mean winter temperature and aridity index). Plus- and Minus-signs indicate sign of the regression
coefficients.

**3.6 Estimating flood moments from multiple controls**

In this section we analyses how well the regression models of the previous section (where A, P95, TWin and AI
were used as covariates) are able to predict the moments at any location in terms ofusing a leave-one-out-cross-
445 validation (Figures 9 and 10). Overall, there is a tendency of for MAF to be overestimated in those areas where the
observed values of MAF are small (e.g. Hungary and Denmark), and underestimated where they are large (e.g.
Carpathians, Northern Italy) reflecting the property tendency of spatial estimators to underestimate spatial
extremes. The overestimation in Finland may also be due to lake retention not being captured adequately in the
model. To some degree CV is also overestimated in areas of low CV (e.g. southern Norway and Denmark) and
underestimated in areas of
450 high CV (e.g. Ukraine, Ore mountains) although there are also large CV areas it is overestimated (e.g. Southern Spain).
The errors are smallest in Russia, Central Germany, British Island Isles and France where the spatial gradients in
CV are relatively smooth.

The median absolute normalized error of MAF and CV is 0.37 and 0.18, respectively, with 25%-quantiles of
0.17 and 0.09 and 75%-quantiles of 0.63 and 0.32. This means that the absolute normalized error of CV is about
455 half that of MAF, which seems to be related to the relatively smaller spatial variability of CV as compared to MAF
(spatial cvs of 1.11 and 0.49, respectively, Table 2).

[Figure]

Figure 9: Absolute normalised errors ANE$_{MAF}$ and ANE$_{CV}$ of the predictions of the regional regression models for MAF and
CV. Errors are evaluated on scale of data (not logarithmised). Colours refer to binned classes of equal frequency. Triangles
460 facing upwards indicate gauges where the model overestimates the moments, triangles facing downwards where the model
underestimates.

Figure 10 depicts the predicted (leave-one-out) MAF and CV using the regressions (Circles in Figure 10, Tables
* * *
**Commented [A73]:** If insignificant no sign shown? Please clarify this.

**Commented [A74]:** This suggests that you have heteroscedastic residuals, and that you haven't explained an important component of the variance in your data.

**Commented [A75]:** You mean LUW was not considered here…not any sort of man-made regulation

**Commented [A76]:** British Isles (Great Britain + Ireland)? Or the island of Great Britain?

**Commented [A77]:** Throughout Europe?

**Commented [A78]:** Consider using a trichromatic legend where one color gets stronger as the errors become more negative, a neutral color indicates where errors are low, and a third color becomes stronger as error get positive.

[revised manuscript text omitted]

**Commented [A84]:** Will the thresholds vary more in larger basins due to their greater environmental heterogeneity? Also, consider spatiotemporal aggregation effects.

**Commented [A85]:** Can you show a figure of this in your regions?

**Commented [A86]:** Because more of the between-region variability is explained by hydroclimatic differences, which makes sense given that the regions are intended to represent distinct hydroclimatic zones in Europe

**Commented [A87]:** What exactly do you mean by landscape evolution?

**Commented [A88]:** This paragraph is comprehensive but a bit too long and reads like a lab report.

[revised manuscript text omitted]

---

## Author Response (AR1)

Response to the interactive comments to the manuscript **hess-2020-600**

**"Characteristics and process controls of statistical flood moments in Europe – a data based"**

by D. Lun, A. Viglione, M. Bertola, J. Komma, J. Parajka, P. Valent and G. Blöschl

We wish to thank the editor and the referees for the time they spent on our manuscript and for their useful and constructive comments. Here we reproduce the comments of the editor and of all referees in *italic characters*, followed by our answers. The line numbers of the referee comments refer to the line numbers of the original submission.
* * *
**Thomas Kjeldsen (Editor)**

*The manuscript has been reviewed by two external experts in the field who have both made meaningful and substantial comments on the study. Both reviewers agree that the manuscript is well written and of interest to HESS. The authors should carefully consider the comments made by the reviewers when submitting a revised version.*

*Additionally, perhaps the authors could consider simplifying the analysis by reducing the number of analysis outlined in Section 2.3. Specifically, I am wondering about the value of the initial ANOVA analysis. I think perhaps a more concise manuscript could drop the ANOVA and focus more on the regression models, thereby also making more space for Reviewer #2 concerns about the validity of the regression models.*

> We thank the editor for his thoughtful comments. We have carefully considered all comments of the reviewers in the revised manuscript and updated the citations.
>
> In most of the cases we adopted the reviewers' suggestions, however we prefer to keep the initial ANOVA analysis. The ANOVA is only a small extension of the regional flood moments (Table 2) and the description of the ANOVA results only comprises the second paragraph of section 3.1. We now include additional diagnostic results of the regression models in the form of hypothesis tests and discuss the validity of the regression models in the light of these results.
* * *
**Kolbjorn Engeland (Referee)**

*The paper provides a comprehensive analysis of a dataset of annual maximum floods covering all Europe and aims to discuss how process controls can explain the spatial patterns of mean annual floods and the coefficient of variation (CV) of floods. The paper comes in a line of papers analysing floods at a European scale (Blöschl et al., 2017; Hall and Blöschl, 2018; Blöschl et al., 2019 and Blöschl et al., 2020). Whereas the previous papers have investigated trends in time, this paper has a clear focus on the spatial patterns. This provides therefore new knowledge and is complementary to the previous papers. The paper is well written and could in my opinion be published after some minor revisions.*

We thank Kolbjorn Engeland for the time he spent on our manuscript and for the useful and constructive comments that helped improve the quality of the manuscript. All his comments are reproduced and addressed in the following paragraphs.

*Lines 38:48: This is because a large basin is less likely to be fully covered by a thunderstorm than a small basin which tends to reduce the variance of extreme catchment average precipitation and thus the MAF (Viglione et al., 2010a, b).*

*I would suggest to add one sentence discussion that there is a transition from convective thunderstorms to long duration stratiform precipitation as catchment size increases (see e.g. Figure 13 in Merz and Blöschl, 2003). This phenomena is also well studied in literature on area reduction factors for extreme precipitation.*

We have added the following sentence:

Convective events, limited in duration and spatial extent, are most relevant for producing floods in small catchments with fast response times (Gaál et al., 2015), whereas long duration stratiform precipitation becomes more relevant as catchment size increases (Merz and Blöschl, 2009).

*Section 2.1 Data: Some more sentences could be added about the data. 1: Are the data from natural catchments not influenced by river regulations ? Do all flood data represent floods caused by rain and/or snow melt, or are there other types of floods like ice jam floods in this dataset ?*

The data has been described in Blöschl, Hall et al. (2019) and more extensively in Hall, et al. (2015). A sentence has been added, citing these studies and addressing the referee's comment:

The time series were manually checked for strong human modifications such as reservoirs (Blöschl, Hall et al., 2019 and Hall et al., 2015) and include both rain floods and snowmelt floods (Kemter et al., 2020).

*Figure 1: You could discuss more if and how your choice of regions influenced the results. I guess that if the aim as to have the best possible predictions of mean annual flood in ungauged basins, you would investigate more in detail how Europe should be divided into sub-regions.*

This is a very good point. We have added the following sentence to section 2.2:

The aim of the partitioning was to represent a small number of contiguous regions that are to some extent hydro-climatologically homogeneous, without considering their effect on predicting flood moments.

We also added the following sentence to section 4.3:

The results depend on the regional partitioning of Europe and will look different for different regions. If the aim of the study was optimal predictive performance of the regional models, the partitioning could be derived based on the data, for example via cluster analysis or regression trees (see e.g. Laaha and Blöschl, 2006).

*Line 143: Please specify units of Qi and catchment area.*

Units have been added in the text.

*Equation 5: Since you use multi-letter symbols for variables, it is difficult to see where the multiplication sign is located. Either you should use only single-letter symbols, possibly combined with subscripts, or use the multiplication symbol to make the equation easier to read.*

Thank you for pointing this out. We feel that using the three-letter abbreviations for the variables makes it easier to follow, so multiplication symbols have been added.

*Lines 167-169: What is the equation for calculating the radius ?*

We have added the following text:

It is calculated as the Euclidean distance between the origin and the mean flood date (mean of the sine and cosine of flood dates in polar coordinates), see e.g. Burn (1997).

*Line 180-181 Could you be more specific on which variables were log-transformed and why ?*

The following sentence has been added explaining which variables were log-transformed and motivating the choice:

MAF, CV, A and P95 were log-transformed, as their distributions were skewed.

*Line 206-207: Probably better to use past tense here.*

We now use past tense here.

*Table 2: The regional cv is listed in the table, but not commented in the text. I suggest that you add some comments in the text.*

We have added the following comment to the text:

The regional coefficients of variation in Table 2 (every other column) reflect the within-region variability of the observed flood moments. They are generally higher for MAF and $MAF_\alpha$ than for CV, both within individual regions and for all of Europe.

*Line 225: 'however' could be removed here*

We have removed 'however' from the sentence.

*254: is k the same as the radius defined on lines 167-169 ('The length of the vector from the origin is a measure of the variability of the date of occurrence, ranging from 0 (uniformly distributed across the year) to 1 (all events on the same day).' ? Then maybe k could be defined in the method section.*

Yes. A reference on how the radius was calculated has been added to the methods section, where the radius will be introduced as k.

*Kemter et al (2020) is missing in the reference list.*

Thank you very much for pointing this out. The reference has been added.

*Line 313: 'which may mask causal relationship'. Do you think that also spurious correlations might be a challenge?*

Yes, we think spurious correlations also are a challenge when interpreting correlations between flood moments and their process controls. Spurious correlations that are not meaningful and probably occur purely by chance could be the correlations for soil texture (Stex), because they are inconsistent with the existing literature. Alternative covariates, that are representative of the runoff generation processes, such as the HOST classification in the UK (Lilly et al., 1998), could provide a remedy for this. Spurious correlations that arise due to an indirect relationship between attributes are for example those between the fraction of forested area (LUF) and MAF, given that densely forested areas tend to be high elevation regions with higher rainfall depths (Lines 603-609). We believe we have already addressed these issues in the paper.

*Figure 5: Maybe one extra point to add: The sign of the correlations listed in Figure 5 might depend on the domain you investigate, and the sign might change between sub-regions of Europe. E.g in the Scandinavian countries, it is a negative correlation between elevation and LUF.*

We have analyzed the correlations among attributes also for all regions separately and indeed they vary between regions. However, we have chosen not to add them for space reasons.
* * *
**Anonymous Referee #2**

*This manuscript has the potential to serve as a strong reference for characterizing the spatial variability of annual peak-flow moments at sites without strong anthropogenic modifications, such as reservoirs, throughout Europe. The leave-one-out cross-validation of a multiple regression model predicting flood moments (mean, Cv, Cs) suggests that, with follow-up efforts, this work could be used to estimate flood moments at ungauged locations with reasonable accuracy in many locations. This work also documents large-scale spatial patterns in controls on flood moments throughout the continent, although the process controls revealed are not especially surprising to people with knowledge of European hydrology. However, numerous technical and presentation improvements detailed below are needed to make this manuscript publishable in HESS. In addition, a more compelling case for how this research could benefit both stationary and nonstationary flood-frequency analysis would be helpful. I have also attached a Tracked Changes Word document with more specific writing and presentation suggestions and some more minor technical inquiries.*

We thank the anonymous referee for the time he or she spent on our manuscript and his or her detailed and constructive comments that helped improve the quality of the manuscript. We are especially thankful for the document with specific writing and presentation suggestions. The writing suggestions were almost entirely included in the revised manuscript, and we reproduce and address the annotated comments from the tracked changes document in this file, after the general review comments. All of the general review comments of the referee are reproduced and addressed in the following paragraphs.

*SOME BASIC CHARACTERISTICS OF THE FLOOD TIME SERIES NEED TO BE CLARIFIED UPFRONT. The authors should state in their abstract whether their set of 2,370 flood series are from stations in anthropogenically impacted basins and whether the "maximum annual flows" they analyze are daily mean flows or instantaneous peak flows. This is important given the small drainage areas of some basins. The authors state that they used the version of the European Flood Database used in Blöschl et al. (2019), which excluded catchments with strong human modifications, such as reservoirs, but did not exclude basins subject to more local anthropogenic perturbations – given their focus on elucidating broad regional patterns. While this dataset contains both [instantaneous?] peak flows and maximum daily mean flows in each year, it seems like the authors might have strictly used peak flows based on descriptions at the beginning of Section 2: "This study uses the data set of European flood discharges of Blöschl, Hall et al. (2019), which . . . consists of 2370 annual maximum peak discharge series from 33 countries". Also, the authors only used 2,370 stations whereas Blöschl et al. (2019) used 3,783. This discrepancy should be explained briefly. Finally, the authors should clarify earlier in the manuscript whether they used calendar years or a designated water year when identifying annual peaks.*

Thank you for pointing this out. We use the exact same data set as in Figure 1 and Extended Data Figure 2 and 8 in Blöschl, Hall el al. (2019). The data set consists of annual maximum discharges, which were derived from both instantaneous peak flows as well as daily flows (this is explained in the section on datasets in Blöschl, Hall et al., 2019). The year refers to calendar years. We have modified the sentence in the abstract about the data in the following way:

"The data consist of maximum annual flood discharge series (instantaneous peaks and daily means) without strong human modifications observed in 2370 catchments in Europe covering the period 1960-2010."

We have modified a paragraph in the data section in the following way

"This study uses the data set of European flood discharges of Blöschl, Hall et al. (2019), which can be found in their supplementary material. The dataset is a subset of the data used in Blöschl, Hall et al. (2019), for which stricter selection criteria applied than for their entire data set (see their section on datasets). It consists of 2370 annual maximum discharge series from 33 countries. Catchment areas range from 5 to 100000 km2, with a median of 383 km². The observation period is 1960 to 2010, and record lengths range from 30 to 51 years with a median of 51 years. The time series were manually checked for strong human modifications such as reservoirs (Blöschl, Hall et al., 2019 and Hall et al., 2015) and include both rain floods and snowmelt floods (Kemter et al., 2020). Annual maximum discharges were derived from instantaneous peak flows and daily mean flows for each calendar year. "

*MOMENT ESTIMATION BIASES MUST BE ADDRESSED. The authors need to discuss the bias in their estimates of the Cv (coefficient of variation) and Cs (coefficient of skewness).*

*First, with regards to the Cv, Ye et al. (2020) demonstrated the extent to which common Cv estimators can be biased when data are skewed or do not adhere to i.i.d. assumptions. While the degree of bias is not as pronounced as it is with daily flow data, quick calculations using the equations described in this paper demonstrate that Cv of annual peak flows can have a substantial bias.*

*Numerous references have also demonstrated the bias of skewness estimates from small samples, including their dependence on record length (Wallis et al., 1974; Bobee and*

*Robitaille, 1975; Carney, 2016). In their discussion, the authors should also recognize the literature on regional skewness coefficients as well as the weighted skewness approaches combining at-site and regional information that the U.S. Geological Survey employs.*

Thank you for pointing this out. We agree, that the uncertainty and bias of estimators should of course be taken into consideration when interpreting the results of this study.

Unfortunately the bias-corrections for the estimator of the CV discussed in Ye et al. (2020) require assumptions on the distribution of the data, which are not in line with the literature on European floods, where a Generalized Extreme Value distribution is most commonly fitted to annual floods.

We have added the following sentence to section 2.3, pointing out, that caution should be used when interpreting spatial patterns of the estimated flood moments, as bias and sampling uncertainty of the respective estimators can be substantial.

"While the estimation uncertainty of the mean is small, the uncertainty and bias of the estimators of CV and CS (equations 3 and 4) can be substantial. Ye et al. (2020) illustrate the uncertainty and bias in the estimation of CV. "

We prefer to use the estimator for CV as given in equation 3 in the manuscript, in order to stay consistent with a large body of hydrological literature and more easily facilitate comparisons in the future.

We agree that statistical estimators of skewness do exhibit substantial bias and are very sensitive with respect to record length. This is among the main reasons why it is so difficult to interpret regional patterns of skewness, as a large portion of these patterns is likely comprised of sampling uncertainty. We have modified the text in the manuscript in the following way.

"For a record length of 50 years and a series with the average estimated moments of the entire dataset (MAF=0.17 m³ s⁻¹ km⁻², CV=0.52, CS=1.28), the standard error and bias of the CS estimate are about 0.56 and 0.22 (simulation), respectively, which is about half and one sixth of the underlying population moment (assuming a GEV-distribution as the data-generating process). This bias and uncertainty for the estimation of skewness are well documented in Wallis et al. (1974), Bobee and Robitaile (1975) and Carney (2016), for example. "

We also recognize the work that has been done in the USA on regional skewness coefficients. Unfortunately we do not have a map of estimated regional skewness coefficients for Europe. We have added the following sentence to the paper.

"Additionally, combining regional with local information can help reduce the estimation uncertainty of statistical moments of flood series, as demonstrated by the weighted skewness approaches of Griffis and Stedinger (2009) and the flood frequency hydrology approach of Viglione et al. (2013), but this is beyond the scope of this paper. "

*NONSTATIONARITY AND ITS POTENTIAL IMPACTS ON MOMENTS MUST BE CONSIDERED. Blöschl et al. (2019) reported regional-scale climate-driven trends in northwestern, southern, and eastern Europe (see Fig. 1). Is it worthwhile to describe the sample moments of sites without considering these changes? In my opinion, the authors should either develop a procedure to exclude sites subject to trends or provide a rationale for treating all sites as stationary given their research goals.*

*In making this decision, the authors should consider the ongoing shift from nival to pluvial regimes in 3/5 regions in Europe makes this an important consideration. If they wish to*

*distinguish trends from long-term persistence, an argument often used to refute nonstationary treatments of hydrologic records, the authors could test for trends of a given trajectory against null hypotheses of long-term persistence (see Matalas and Sankarasubramanian, 2003; Cohn et al., 2005). The authors should also note trends in both the mean and variance affect both Cv and Cs estimates [see Serago and Vogel (2018) for some initial guidance for making these adjustments]. Hecht and Vogel (2020) offer one approach for modeling trends in variability and reference a handful of other moment-based ones, including Strupczewski et al. (2001).*

Thank you for pointing this out. In this paper we have adopted a pragmatic approach. We are interested in the statistical flood moments for the period 1960-2010 and trends are not in the center of our interest, because they have already been comprehensively analyzed by others.

Indeed, the focus of the models in Blöschl, Hall et al. (2019) and this paper are different. We are interested in providing a large-scale analysis of European flood data, using all the data available to use. Of course it would be possible to extend the analysis to non-stationary moments and interpret their behavior with respect to their spatial controls, but this is beyond the scope of this study. Any non-stationarity or persistence in the date will affect the properties of all estimators used in this study. We have added the following sentence to section 2.3, pointing out this issue.

"In interpreting the results, we do not account for any non-stationarities of the flood moments, as the focus is on the aggregated behavior during the observation period. Any autocorrelation that may be present will increase the uncertainty of the estimates, although they are usually small in annual flood data, and are therefore rarely considered in flood frequency estimation (Hosking and Wallis, 2005). "

*THE RESIDUAL BEHAVIOR OF THE REGRESSION MODELS MUST BE EVALUATED. The authors do not report the normality, heteroscedasticity, autocorrelation of their residuals. They also do not report the variance inflation factor or alternatives measures of multicollinearity for their multiple regression equations. This is especially important if one is making process-based inferences using covariate matrix-derived statistics from regression models. The authors should consider using a Supporting Information (SI) section to display the residual behavior of their models. Also, the authors report the tendency for large MAFs to be underestimated and small ones to be overestimated. This suggests that residuals might not be homoscedastic and that another covariate may be needed to produce a multiple regression model that meets the homoscedasticity requirement for making inferences from standard error-based metrics (Helsel et al., 2020).*

Thank for pointing this out. These are all interesting analysis, but given that the paper is already long, we prefer to focus on the physical interpretation of the spatial patterns of the moments.

Of course, checking the assumptions of a linear regression model is important for making accurate statistical inferences. However, the aim of the regression models in this study is not to perform statistical inferences or provide optimal predictions, but rather to serve as a baseline for more sophisticated analysis. If the assumption of homoscedasticity, no autocorrelation or normality of the error term is not met, the OLS-estimator remains unbiased and consistent (e.g. Proposition 1.1 and Proposition 2.1 in Hayashi, 2000). Of course the violation of these assumptions will affect the distribution of the OLS-estimator and therefore temper with inferences, such as hypothesis tests.

To address this more explicitly in the manuscript we have added the maximum variance inflation factors for each regional regression model, as well as p-values of Breusch-

Pagan and Shapiro-Wilk tests to tables A.1 and A.2 in the Appendix. We added a paragraph at the end of section 4.2 to discuss these results. Collinearity between potential explanatory variables for the regression models is also investigated in section 3.4, where interpretations of these relationships are discussed, which guided the selection of variables for the regression analysis to minimize collinearity. The following paragraph was added to section 4.2:

"The properties of the estimators of the investigated correlations and linear regressions depend on assumptions which are only partly met in this analysis. Tables A.1 and A.2 in the appendix report the maximum Variance inflation factors [VIF] for each regional regression model from section 3.5, as well as p-values of hypothesis tests for the homoscedasticity and normality of the residuals. While the VIFs are generally low (indicating a low degree of multicollinearity), the assumption of homoscedasticity and normality of the residuals are generally not met for many models, which may be related to the large number of catchments. Additional diagnostic plots for the regional regression models can be found in the supplementary material. The OLS-estimator still remains unbiased and consistent under these conditions (Hayashi, 2000), but no inferences such as significance tests of individual coefficients should be made from standard properties of the OLS-estimator. The inclusion of additional covariates could help to reduce heteroscedasticity, but would lead to less parsimonious models. Heteroscedasticity could be reduced by considering different regional partitions of Europe."

In addition we now provide diagnostic plots for the regression models in a Supporting Information (SI) section.

*THE CHOICE BETWEEN LOG-SPACE VS. REAL-SPACE MOMENTS SHOULD BE RECOGNIZED. The authors should also recognize in their manuscript that moments of log-transformed floods are often used in FFA and clarify that real-space moments are used upfront.*

We have added the following sentence to section 2.3, emphasizing that real-space moments of flood series are analyzed in this manuscript.

"While in some cases log-transformed variables are used in flood frequency analysis (Griffis and Stedinger, 2007), here we analyze real space moments of flood series, in line with European practice (e.g. Merz and Blöschl, 2009). "

*MIXED POPULATIONS SHOULD BE CONSIDERED IN THEIR INTERPRETATION OF RESULTS. While the authors somewhat recognize mixed populations (e.g. description of Alps and Norwegian coast flood-generating processes), they compute moments assuming floods at each site belong to homogenous populations. While statistically evaluating the presence of mixed populations at individual sites lies beyond the scope of this paper, it is important to consider mixed populations explicitly when interpreting results and to caution readers about problems associated with choices to neglect them at individual sites. While the authors use an analysis of flood timing to help identify drivers of floods, they do not specifically check for the presence of multi-modal peaks suggesting mixed distributions in them. This type of quicker analysis could support some of the good observations that authors make about mixed distributions in specific regions. Finally, the authors should communicate an awareness of this 'mixed populations' literature in their discussion of mixed populations.*

Thank you for pointing this out. Indeed, analyzing mixed populations of floods unfortunately is beyond the scope of this manuscript, as the data is simply lacking. We have added the following sentence to the manuscript, discussing this issue

"Further analyses could consider different subpopulations of floods associated with specific generation mechanisms (Tarasova et al., 2019), e.g. as indicated by their seasonality (Blöschl et al., 2017). An approach based on mixed distributions (e.g. Fischer et al., 2016), could yield additional insights into the spatial patterns of flood moments of mixed populations. "

*MORE DETAILS ABOUT THE DRAINAGE AREA-NORMALIZED EQUATION(S) ARE NEEDED. I like the authors' idea of normalizing their analysis to a given drainage area (100 km2) since drainage area is still an important descriptor of flood-generating processes even when specific discharge values are used to express peak flows. However, it would be nice to report goodness-of-fit measures for this model and show the fit graphically, the latter which can be done in the SI section if space constraints remain. The authors also describe the creation of equations that establish values of the Cv and Cs for 100-km2 drainage areas, but it is unclear if these DA-adjusted values are ever evaluated as response variables in the multiple regression models.*

Thank you for pointing this out. Goodness-of-fit measures for this model are now reported in the appendix in Table A.6. The fit of the models is shown graphically in figure 4.

The equations for CV and CS are established alongside the equation for MAF, as pointed out by the referee. We believe that we state that MAF instead of $MAF_\alpha$ is used for the regression models in line 397: 'We used MAF, rather than $MAF_\alpha$, in order to avoid prior assumptions regarding the role of catchment area.'

*NON-MONOTONIC RELATIONSHIPS WITH COVARIATES SHOULD BE CHECKED - AT LEAST IN AN EXPLORATORY DATA ANALYSIS. The authors raise the possibility of non-monotonic relationships between moments and catchment descriptors in discussions of prior findings, but they only examine monotonic relationships in their linear regression models. In particular, they cite Smith et al. (1992), who found that floods in the Appalachian mountains in the eastern US demonstrated an increase in the CV with catchment area for catchments up to 100 km2 and then exhibited a decrease with catchment area in larger basins. They also recognize that Wang et al. (2017) found a non-monotonic relation between water body size and the Cv. In addition, Pallard et al. (2009) also found that Cv decreases with drainage density in catchments with sparse drainage networks but then increase after a reaching a minimum. I think that if the authors can claim that exploratory data analyses did not demonstrate any non-monotonic trajectories like these, they don't need to formally test hypotheses of non-monotonic change with statistical models, but they should briefly demonstrate that they performed exploratory data analysis (EDA) justifying the monotonic relationships they modeled.*

Thank you for pointing this out. While we agree, that non-monotonic relationships with covariates could be present in the data and would be interesting to analyze, we feel that this would require a separate subsection in section 3 and go beyond the scope of the paper. We therefore have included the possibility of non-monotonous relationships in the discussion section instead.

"While here we examined monotonic relationships and linear relationships, it would also be worth to explore non-monotonic relationships between flood moments and covariates (see e.g. Blöschl and Sivapalan, 1997; Smith, 1992; Pallard et al., 2009). "

*DOES ARIDITY CAUSE FLOOD VARIABILITY? The authors make an important association between the aridity index (AI) and the Cv of annual floods. However, it is important to recognize more succinctly that arid regions tend to have greater interannual precipitation variability, and, for that reason, arid basins tend to have larger Cv's. This is important when considering the implications of these findings under climate change. If a region becomes drier, it's interannual precipitation variability will not necessarily increase. A discussion about the implications of these cross-sectional findings for projecting flood responses of environmental changes at a given location over time would enrich the paper.*

Thank you for pointing this out. We have modified the text in the introduction in the following way.

"Based on data from around the world, Farquharson et al. (1992) found CV to increase with the Aridity Index (the ratio of potential evaporation and MAP). This dependency may be the result of at least two processes. On the one hand, low and variable runoff coefficients tend to increase the flood CV far beyond that of rainfall (Viglione et al., 2009). On the other hand, the CV of rainfall (variability between years) is also sometimes larger in arid regions than in more humid regions (Fatichi et al., 2012). "

*ORDINARY KRIGING. Ordinary kriging visualizes broad regional patterns but may be limited for applications in ungauged basins. The kriging results look visually pleasing and achieve the goal of illustrating broad regional patterns in flood moments. However, what if nearby basins have greatly different drainage areas (since this is stated to be a map of MAF and not MAF[alpha]) or pronounced differences in other catchment characteristics that can change abruptly? In the future, the authors could consider kriging in attribute space instead of geographic space. If the authors retain these kriged maps to display broad regional patterns, they should note the limitations of using these interpolations for characterizing flood regimes at ungauged sites. To me, it seems like the regression equations should work reasonably well for estimating moments at many sites. And if they choose to argue that kriging can be used to estimate moments in ungauged basins, then a more formal cross-validation analysis and more detailed reporting of model performance is necessary. Alternatively, they could turn this kriging exercise into a separate paper.*

Thank you for pointing this out. Ordinary kriging for regionalizing floods has already been extensively cross-validated in different areas of the world (e.g. Rosbjerg et al., 2013). We therefore prefer not to add a cross-validation of the kriging results for space reasons. The intention of Figure 10 is to offer a quick visual comparison between the two regionalization approaches. We agree that, before the use of ordinary kriging estimates for applications in ungauged basins, additional cross-validations would be useful. We have added the following sentence to section 3.6 to discuss the limitations of this result.

"The intention of Figure 10 is to offer a visual comparison between the two regionalization approaches. Before the use of ordinary kriging estimates for applications in ungauged basins, additional cross-validations would be useful in the spirit of Rosbjerg et al. (2013). "

*IMPORTANCE FOR FFA IN PRACTICE. This paper successfully elucidates broad regional patterns in flood moments across Europe. Their leave-one-out cross-validation suggests that flood moments can be reasonably estimated in many regions at sites whose covariate values are known. The implications of these errors for design flood estimates could be made stronger by computing the design floods with a GEV quantile function (noting issues with this distribution in specific regions from prior studies, such as Salinas et al. (2014)) using moments estimated from observations and from the multiple regression models. The authors should also address practical concerns regarding nonstationarity described above. In addition, the authors should note the contribution that their study makes to improve upon other recent prediction in ungauged basins efforts in Europe.*

Thank you for pointing this out. We agree that the regression results can provide a baseline for more sophisticated studies on regional flood frequency analyses in Europe, which however, is not the aim of the current paper. Estimating flood quantiles would be a natural subsequent step after the regional estimation of moments, but also lies beyond the scope of the present study. Regarding the nonstationarity we have added the following sentence to section 4.3:

"The process controls identified here can assist in choosing suitable covariates, both for stationary and nonstationary flood frequency models. "

*OVERALL PRESENTATION. The paper reads a bit like a lab report in places and generally has the potential to be shortened considerably without losing much content. In some places, starting paragraphs with more topic sentences could help orient the reader better and curtail the 'rambling' nature of some sections, such as the bivariate correlation results. The correlation analysis is important for interpreting regression model results and many of the insights on multicollinearity in the data are good, but the presentation of it should be a bit more focused on supporting the multiple regression model analysis and not a comprehensive review of the entire correlation matrix. The submission also requires more editing for fluidity/conciseness and proper punctuation. While I made some writing and grammar suggestions in the Track Changes document, I did not perform a comprehensive check for these issues and suggest that the authors find someone else who can do that.*

Thank you for the detailed suggestions, which we really appreciate. We followed most of the suggestions, which we believe have strengthened the paper. Additionally we have condensed the section on the correlation results slightly and have had the paper proof-read. In addition, Hess papers are all copy-edited, so any small English inaccuracies will be taken care of.

First column is the number of comment, second column is the line in the annotated pdf, third
is the comment of the referee, fourth is the response, fifth are the text changes

| | | Review comments | Response | Text change |
|---|---|---|---|---|
| A1 | 14 | Instantaneous or daily flows? Minimally impacted basins? | Suggestion adopted | The data consist of maximum annual flood discharge series (instantaneous peaks and daily means) without strong human modifications observed in 2370 catchments in Europe covering the period 1960-2010. |
| A2 | 15 | Mention that these values vary widely due to catchment size, climate and other covariates. In my opinion, it's not necessary to state this in the abstract but there's nothing wrong with it either. | Suggestion adopted | The estimated moments MAF, CV and CS vary due to catchment size, climate and other controls, their averages across Europe are 0.17 m³ s-1 km-2, 0.52 and 1.28, respectively. |
| A3 | 19 | Due to the greater sensitivity of sampling variability to Cs in short records? See general comments about regional skewness. | Explanation added | The pattern of CS is similar, albeit more erratic, in line with the greater sampling variability of CS. |
| A4 | 21 | Can you state why briefly? | Explanation added | .. weaker mainly due to the effect of snow melt. |
| A5 | 26 | Do you need to describe this here? You already did above. | Sentence deleted as suggested | |
| A6 | 26 | Is it aridity itself or do arid regions happen to greater interannual precipitation variability? | Both rainfall and runoff generation are considered as relevant and this is discussed in more detail in the main part of the paper | |
| A7 | 27 | You already said that they are relevant in most of Europe earlier on in this paragraph | Sentence deleted as suggested | |
| A8 | 35 | Scientific? | Changed to scientific | |
| A9 | 38 | Storm events in general? Not always T-storms. | Changed thunderstorm to storm | |
| A10 | 44 | See Pallard et al. (2009) on the effect that drainage density has on the Cv https://hess.copernicus.org/articles/13/1019/2009/hess-13-1019-2009.pdf | We are discussing here a slightly different point, i.e. CV as a function of area rather than drainage density | |
| A11 | 48 | Floods from synoptic-scale precip events (e.g. frontal systems)? | Changed as suggested | such as floods from synoptic-scale precipitation events (e.g. frontal systems) and snowmelt |

| A12 | 53 | Many USGS regional flood frequency studies based on observed data have revealed non-climatic controls. | Suggestion adopted | While USGS regional flood frequency studies based on observed data have revealed non-climatic controls (e.g. England et al., 2019) most knowledge |
|---|---|---|---|---|
| A13 | 58 | Attributing? | Suggestion adopted | Attributing |
| A14 | 64 | Consider stating this one first | Order of sentences changed | |
| A15 | 66 | Also, MAP might be better than event precip in places with snowmelt-driven floods | Yes, probably, but we did not look at this specifically. | |
| A16 | 68 | Is it aridity itself or does this stem from the tendency of more arid catchments to have greater interannual precipitation variability? | In general, probably both increased rainfall CV and smaller (and more variable) runoff coefficients may be relevant. The latter process is clearly sufficient to produce high flood CVs (e.g. Viglione et al., 2009), and larger rainfall CV may further contribute to increasing flood CVs. | Farquharson et al. (1992) found CV to increase with the aridity Index (the ratio of potential evaporation and MAP). This dependency may be the result of at least two processes. On the one hand, low and variable runoff coefficients tend to increase the flood CV far beyond that of rainfall (Viglione et al., 2009). On the other hand, the CV of rainfall (variability between years) is also sometimes larger in arid regions than in more humid regions (Fatichi et al, 2012). Merz and Blöschl (2009) found … |
| A17 | 69 | The greater the PET, the higher the Cv in lowlands of Austria? | Suggestion adopted | and CV in the lowlands of Austria (the greater the PET, the higher the CV), which they interpreted in terms |
| A18 | 71 | This sounds very interesting, but could you explain in a sentence or two why Cv becomes lower with higher DA's in basins where infiltration-excess flow dominates and why it becomes higher with DA in basins where saturation-excess flow predominates? | We have reworded the sentence slightly to make it clearer. The findings result from the model structure and is not directly apparent from the reference cited. | Iacobellis et al. (2002), found that CV behaviour is controlled mainly by the long-term climate and the infiltration characteristics at the catchment scale. Specifically, they suggest that in arid and impermeable catchments CV tends to decrease with area because the infiltration excess (Horton type) mechanism dominates while in humid and vegetated catchments CV tends to increase with area because the saturation excess mechanism dominates. |
| A19 | 94 | Please describe the degree to which and the types of anthropogenic | More detailed description has been added | The time series were manually checked for strong human modifications such as reservoirs |

| | | | | |
|---|---|---|---|---|
| | | perturbations to which basins in your sample are subject. | | (Blöschl, Hall et al., 2019 and Hall et al., 2015) and include both rain floods and snowmelt floods (Kemter et al., 2020). |
| A20 | 95 | Assuming instantaneous peaks? | More detailed description has been added | Annual maximum discharges were derived from instantaneous peak flows and daily mean flows for each calendar year. |
| A21 | 101 | Total days or wet days? | More detailed description has been added | Extreme precipitation is quantified by the daily rainfall rate that is not exceeded in 95% of the days of the year, … |
| A22 | 102 | The duration of precipitation to examine varies substantially by region and catchment size. Can you convey an awareness of this in your introduction of these covariates? | Yes, in smaller, flashier catchments one would expect shorter rainstorms to be more relevant. | and the long-term mean of the maximum 2-day precipitation of each year. While the duration of event precipitation to examine varies with catchment size and characteristics due to differences in response times (Gaál et al, 2012) we chose a constant value of two days here for consistency. |
| A23 | 103 | How accurate are these modeled values? Can you add a sentence or so stating this and any places where inaccurate estimates may distort your analyses? | Modified as suggested | Fan and Van Den Dool (2004) discuss any biases of the soil moisture data set, which may distort some of the findings here. |
| A24 | 106 | Can you mention forest and water body categories here? | Modified as suggested | Land use was quantified as the percentage of total catchment area and includes forest areas and water bodies |
| A25 | 111 | Proper name? | Yes | Data Base on European Floods |
| A26 | 117 | Check for change? | URL is correct as of now | |
| A27 | 131 | What about lower-altitude streamgages draining primarily alpine catchments? How often is this an issue? Generally, it looks like you have a reasonable alpine region | Low altitude streamgauges draining primarily alpine catchments are an issue in the largest river basins analysed here, for example the Danube at Vienna (elevation 150m a.s.l) while the mean catchment elevation is 785 m due to alpine tributaries such as the Inn. Only for 65 catchments the difference between mean elevation and catchment elevation is more than 1000m so this is not generally considered an issue. One possibility to address this issue would be | its stream gauge. The latter is usually representative of the entire catchment, as only for 65 catchments the difference between stream gauge elevation and mean catchment elevation is more than 1000m. |

| | | | | |
|---|---|---|---|---|
| | | | to exclude the largest basins, but then the analysis with respect to catchment area would be less meaningful. | |
| A28 | 132 | Make NE region color standout more. Its shade of blue is too similar to the dark green lowlands and plentiful blue lakes of this region | We have modified the colours to make the points stand out more | |
| A29 | 147 | Is there a reference describing the computations you made? | No, these figures were obtained by a simple simulation, so should be easily reproducible by the reader | standard error and bias of the CS estimate are about … (simulation) |
| A30 | 148 | In your discussion, consider commenting on Salinas et al. (2014) who investigated how well the GEV distribution performed throughout Europe | We believe that the focus of this paper is somewhat different, i.e. on analyzing the flood moments rather than the choice of distribution function. | |
| A31 | 152 | Transforming both sides logarithmically before fitting? How well did this equation fit the data? Please present goodness-of-fit stats and a graphic in the SI. | Yes.

The goodness-of-fit statistics of this equation are now presented in the appendix (Table A.6). Additionally the goodness-of-fit can be seen from Figure 4. | were found by ordinary least squares regression in the logarithmic space. |
| A32 | 163 | Log-log? Where both the independent and dependent variables are log-transformed? | Yes | Specifically, we estimated the dependence of MAF, CV and CS on catchment area from Eq. (5) and analogous equations for CV and CS, transforming all variables logarithmically. |
| A33 | 164 | Nice. Try to emphasize this throughout the paper a bit more. | We have checked the potential of emphasizing this more throughout the paper and believe we already have the right balance. | |
| A34 | 169 | Nice description. Did you formally test any hypotheses related to seasonality using circular stats? | We did not. Hall and Blöschl (2018) are testing this. | |
| A35 | 174 | And are a limitation of the regression analysis? | Generally speaking, one of the potential limitations of a linear regression analysis. | |

| A36 | 180 | Just the explanatory variables? Or also the peak flows? | Explanation added as suggested | MAF, CV, A and P95 were log-transformed, as their distributions were skewed. |
|-----|-----|--------------------------------------------------------|--------------------------------|------------------------------------------------------------------------------|
| A37 | 187 | Can you note which ones were excluded in SI or Git repo? | These catchments are distributed throughout Europe, but given they are few, listing them is perhaps not needed. | |
| A38 | 201 | Good. Can you mention earlier that you log-transformed the moment values after computing them in real-space? (At least this is what it sounds like you did) | Has now been mentioned earlier | |
| A39 | 203 | Are subscripts missing here or is this a pdf conversion issue | This seems to be a Pdf conversion issue | |
| A40 | 205 | The kriging results look visually pleasing and reveal broad spatial patterns, but what if nearby basins have greatly different drainage areas or pronounced diffs in other catchment characteristics that can change abruptly? In the future, you could consider kriging in attribute space instead if all your covariates can be gridded. In this paper (or another one), you should consider stating the limitations of this kriging analysis and perform a split-sample validation experiment to see how well it does at ungaged locations. You could also take this out and make it a separate paper. | Yes. Comparisons of ordinary kriging with alternative regionalization methods have been performed in the past (e.g. Merz and Blöschl, 2005). It would certainly be worth to conduct similar comparisons at the European scale. | |
| A41 | 214 | Consider showing histograms | Instead we are giving the spatial distribution of the moments. Some information on the distribution can also be inferred from Figure 4. | |
| A42 | 225 | In other words, most of the variability lies within the regions and not between them. | Sentence modified as suggested | i.e. much of the variability lies within the regions and not between them |

| A43 | 226 | Something to think about: how much of the partitioning would the regions explain if you considered of basins different sizes separately? | We conducted preliminary analysis of this question and the statistics do not change much if one stratifies by catchment area. | |
|---|---|---|---|---|
| A44 | 230 | Does this mean that you have larger catchments in some regions than others? | This comment does not refer to regions but to the patterns across all of Europe | slightly more homogeneous spatial patterns across Europe than MAF, as the effect of catchment |
| A45 | 234 | Flashy mountainous watersheds with high rainfall | Comment added | Large as well, partly because of flashy mountainous catchments with high rainfall |
| A46 | 235 | Consider mentioning the extremely low R^2 here | We are mentioning it now. | (which can also be seen from the low R2 in Table 2) |
| A47 | 239 | Interesting since Cs is normalized by sd^3. An increase in sd raises Cv but lowers Cs. | We believe this is already clear. | |
| A48 | 264 | Is this variability mainly due to snow cover, melt timing or also rain-on-snow events? | This variability is probably due to the majority of floods being driven by snow melt and some by rain-on-snow events, but further analyses would be required to ascertain these processes in detail | |
| A49 | 264 | Is their CV large due to interannual snowpack variability? Timing issues with spring thaws? Mixture of spring rains and snowmelt? | The large CV is probably due to the majority of floods being driven by snow melt and some by rain-on-snow events, but further analyses would be required to ascertain these processes in detail | |
| A50 | 280 | Snow or rain-driven? Or both? | Again, this would require more detailed analyses. | |
| A51 | 283 | Nice figure | | |
| A52 | 287 | I like these circled areas | | |
| A53 | 297 | Good obs | | |
| A54 | 299 | Could the paucity of small basins in Central-Eastern Europe contribute to the positive relation between Cs and basin area there? | Yes, this is a possibility (and one of the reasons we are giving basin area in Table 3), another reason are snow processes. | |
| A55 | 316 | 95th percentile daily precipitation | We believe that 'daily precipitation not exceeded 95% of the time' is sufficiently clear | |

| A56 | 322 | Check correlation between precipitation and forest cover | We did, and correlations on the order of 0.4 are typical in Europe, mainly because forests are mainly left in the mountains that tend to have higher precipitation (Fig. 5) | |
| --- | --- | --- | --- | --- |
| A57 | 330 | In which region are the largest basins? | There is a tendency for the Central-Eastern region to have larger basins (Table 3) although the largest basins occur in various regions (Figure 4) | |
| A58 | 336 | Good | | |
| A59 | 345 | Explain a bit more. How might snowmelt temper this correlation? | We are now explaining the argument in more detail. | more important role of snowmelt there, given that snowmelt floods tend to occur at the same time over large areas, so one would expect a smaller reduction of flood peaks due to spatial averaging than for rain floods. |
| A60 | 368 | Not sure this is necessary to report in the text. | This brief text (i.e. –r). may perhaps clarify the argument for some readers | |
| A61 | 379 | Good observation. I suggest adding a sentence saying that this should NOT suggest that deforestation reduces floods. | Sentence added | through orographic effects, implying that the positive correlation cannot be interpreted as deforestation reducing floods. |
| A62 | 381 | Explain this in terms of infiltration being greater in coarser soils, which tend to be more permeable | We are explaining this later in the discussion section but an explanation is not straightforward as the permeability will affect both the mean and the standard deviation of the flood peaks. | |
| A63 | 384 | Is it worth going to into so much detail about the bivariate correlations when there are so many confounding factors, as you seem to recognize? | We believe that the bivariate correlations are a first step to support the analysis. While there are indeed many confounding factors that bivariate correlations have the additional advantage of usually more robust estimates as compared to multivariate correlations. | |
| A64 | 388 | Suggest adding dark vertical lines to separate each region's results if possible | Vertical grey lines have been added | |

| | | | | |
|---|---|---|---|---|
| A65 | 397 | Not normalized by area at all? | Normalized by area, i.e. specific flow, rather than allowing for an areal dependence. MAF is given in Equation (1) | |
| A66 | 398 | Is it worth including so much about Cs then? | The treatment of CS in the paper is a compromise in that we are providing some of the results, but not to the same extent as for MAF and CV for the reasons stated | |
| A67 | 400 | What is the water balance group? Look at interaction effects? | The water balance group has now been explained | water balance (i.e. SM, PET and AI) by one covariate |
| A68 | 413 | Perhaps a proxy variable for snowmelt-driven floods is needed there Also, do smaller headwater basins in the Alpine region have more snow cover and, if so, do they generate larger floods than larger ones with less snow cover? | Not all the floods in the Alpine region are necessarily snow melt driven. Yes, smaller headwater basins tend to have more snow cover because of higher elevations, but this does not generally translate into higher floods (Merz and Blöschl, 2003). | the R² of the model in the Alpine region is low, which may be a reflection of the hydrological heterogeneity of the area, involving snow melt, rain-on-snow and rain driven floods (Merz and Blöschl, 2003). |
| A69 | 417 | Good interpretation | | |
| A70 | 427 | Please put all regional R^2 in parentheses as you describe them in the text. Also, it might help to put this table in the main text. Check for other instances of this throughout the manuscript. | We followed this suggestion and put the R2 in parenthesis where mentioned | |
| A71 | 428 | Please tell readers how higher winter temperatures lead to lower Cv's here | The explanation is complex and probably related to the rainfall regime. We have reworded the sentence to make it more general. | Atlantic region are partly aligned with higher winter temperatures. |
| A72 | 432 | Good analysis | | |
| A73 | 441 | If insignificant no sign shown? Please clarify this. | The signs are shown if the variable is included in the model in the stepwise selection procedure | |
| A74 | 444 | This suggests that you have heteroscedastic residuals, and that you haven't explained an important component of the variance in your data. | Yes, the model does not explain the full variance. It would be interesting to test more complex models in order to avoid these estimation biases. | |
| A75 | 448 | You mean LUW was not considered here…not any | We did not consider land use here to render the model | |

| | | sort of man-made regulation | more parsimonious. The data set consists of catchments with minimum man-made regulations; analyses with an extended data set (including regulated catchments) would be an interesting extension of this work. | |
|---|---|---|---|---|
| A76 | 451 | British Isles (Great Britain + Ireland)? Or the island of Great Britain? | This should read British Isles | British Isles |
| A77 | 453 | Throughout Europe? | Added as suggested | throughout Europe |
| A78 | 458 | Consider using a trichromatic legend where one color gets stronger as the errors become more negative, a neutral color indicates where errors are low, and a third color becomes stronger as error get positive. | We tested numerous possibilities and for the given map the colour scale chosen seemed optimal to us as it emphasizes the absolute values of the error and yet allows identification of the sign with lower priority. | |
| A79 | 467 | List these in SI. I assume these were left out of the kriging analysis as well. | These catchments are distributed throughout Europe, but given they are few, listing them is perhaps not needed. We have removed the associated sentence. | |
| A80 | 492 | Are these storm tracks more regular? | They produce more variable precipitation | influence of Mediterranean storm tracks associated with high variability of extreme precipitation (Hofstätter et al., 2018) perhaps along |
| A81 | 500 | Explain a bit more | We have added more information | mostly high CV due to the more non-linear runoff generation as compared to wetter regions. |
| A82 | 515 | Good commentary about consistency with other studies | | |
| A83 | 516 | From large frontal systems? | Changed as suggested | large-scale precipitation from large frontal systems as would be expected |
| A84 | 522 | Will the thresholds vary more in larger basins due to their greater environmental heterogeneity? Also, consider spatiotemporal aggregation effects. | They will be smoothed out according to this reasoning. Yes, spatiotemporal aggregation effects may perhaps increase the rainfall return period at which the thresholds become relevant. | as threshold processes associated with Hortonian runoff generation or soil storage homogeneity may be more relevant in small catchments while in large catchments these threshold effects may be smoothed out, and spatiotemporal aggregation |

| | | | | may introduce additional scale effects (Penna et al., 2011; Rogger et al. 2012). |
|---|---|---|---|---|
| A85 | 531 | Can you show a figure of this in your regions? | Figure 4 gives an indication of this relationship | |
| A86 | 543 | Because more of the between-region variability is explained by hydroclimatic differences, which makes sense given that the regions are intended to represent distinct hydroclimatic zones in Europe | We believe this is not so much a result of a subdivision into regions as climate explains the flood moments both through the region subdivision and the covariates. Rather it has to do with the continental scale where climate differences can be much larger than those in a region. | important at the European scale than at the regional scale. This finding is likely related to the larger spatial variability of climate variables within European than within a region. |
| A87 | 548 | What exactly do you mean by landscape evolution? | Explained as suggested | soil moisture and the geomorphological processes of landscape evolution that affect runoff generation and routing, whereas |
| A88 | 554 | This paragraph is comprehensive but a bit too long and reads like a lab report. | Reworded as suggested | capturing antecedent soil moisture less. While regional studies in Greece and Austria have suggested that MAP is a better predictor of MAF than other precipitation variables (Mimikou and Gordios, 1989; Merz and Blöschl, 2009) this does not seem to be the case at the European scale. CV is always a better correlated |
| A89 | 558 | Because the antecedent soil moisture conditions tend to vary more than they do in humid catchments? | Yes. Explanation added | because the antecedent soil moisture conditions tend to vary more than they do in humid catchments, so some of the events may be a combination of both large precipitation and wet initial conditions such producing much larger floods than usual |
| A90 | 602 | Good pt | | |
| A91 | 611 | Show this? | Yes. Explanation added | decreasing CV beyond as shown by Wang et al., 2017) |
| A92 | 618 | Nice analysis. | | |
| A93 | 632 | What values did they obtain? For which regions? | We have added the requested information | similar, but smaller scale studies in the literature of flood regionalization, that typically give ANE of 0.35 for the 100yr specific flood and smaller values for the MAF (Salinas et al., |

| | | | | 2013; Rosbjerg et al., 2013). The fit of the |
|---|---|---|---|---|
| A94 | 635 | Change term | Changed as suggested | model applicable to all regions of Europe. |
| A95 | 637 | If I understood you correctly, didn't you say earlier that these regions reflected general hydroclimatic properties rather than flood-generating processes in particular? | Wording adjusted to make consistent with that in section 2.2 | previous climatic partitions of Europe and guided by flood seasonalities rather than optimal predictive performance |
| A96 | 637 | Mention this earlier | We now mention this in section 2.2 | |
| A97 | 639 | Is it worth mentioning that the lack of importance of land-surface characteristics in explaining the spatial variability of floods over large regions of Europe should not be construed to mean that land-surface perturbations have a second-order effect at individual sites compared to climate? | Mentioned as suggested | predictive power of variables related to land use, soil and geology for hydrological quantities that one would expect to be very relevant at individual sites (Merz and Blöschl, 2009, Rogger et al., 2017), |
| A98 | 659 | Can you describe this in simpler language a bit so a wider range of readers who are not familiar with Perdigao and Bloschl can understand this? | We have added more detail in simpler language | This is because of the space-time asymmetry discussed in Perdigao and Blöschl (2014), i.e. the fact that, because of the celerity of coevolution, spatial and temporal statistics are not necessarily the same. For example, based on data in Austria, Perdigao and Blöschl (2014) found that a 1% increase in precipitation as one moves in space leads to a 2.3% increase in flood peaks, while the same increase in precipitation as one moves in time leads to an increase of only 0.6%. |
| A99 | 662 | Good pt…expand on it a bit more | We have added more detail in simpler language (see above) | |
| A100 | 669 | Can you give an example of such a coevolutionary index earlier on?

I would add a few more of your key accomplishments in this paper to the last | We have explained the asymmetry associated with the coevolutionary index in more detail above (A98). | |

| | | paragraph as well as a sentence or two regarding more general future directions, and not a specific focus on coevolution. See ideas about nonstationarity and mixed distributions. | The key accomplishments of this paper are summarized earlier in the same section | |
|------|-----|---|---|---|
| A101 | 680 | Residual normality? | We have added to Tables A.1 and A.2 the variance inflation factor which we consider more important in the context of this paper. | |
| A102 | 709 | Consider using CREDIT system. | Given this is a simple paper, the CRediT (Contributor Roles Taxonomy) of 14 roles is perhaps not needed. | |

**References**

Blöschl, G. and M. Sivapalan (1997) Process controls on regional flood frequency: Coefficient of variation and basin scale. Water Resources Research, 33 (12), pp. 2967-2980.

Blöschl, G., Hall, J., Parajka, J., Perdigão, R. A., Merz, B., Arheimer, B., ... & Canjevac, ˇ I. (2017). Changing climate shifts timing of European floods. Science, 357(6351), 588-590.

Blöschl, G., Hall, J., et al. (2019) Changing climate both increases and decreases European river floods. Nature, 573(7772), 108-111.

Blöschl, G., Kiss, A., Viglione, A. et al. Current European flood-rich period exceptional compared with past 500 years. Nature 583, 560–566 (2020). https://doi.org/10.1038/s41586-020-2478-3

Bobee, B., R. Robiataille (1975) Correction of bias in the estimation of the coefficient of skewness, Water Resour. Res., doi: 10.1029/WR011i006p00851

Burn, D. H. (1997). Catchment similarity for regional flood frequency analysis using seasonality measures. Journal of hydrology, 202(1-4), 212-230.

Carney, M.C. (2016), Bias correction to GEV shape parameters used to predict precipitation extremes, doi: 10.1061/(ASCE)HE.1943-5584.0001416

Cohn, T.A., H.F. Lins (2005), Nature's style: naturally trendy, Geophys. Res. Lett., 32, L23402, doi:1029/2005GL024476

England, J.F., Jr., Cohn, T.A., Faber, B.A., Stedinger, J.R., Thomas, W.O., Jr., Veilleux, A.G., Kiang, J.E., and Mason, R.R., Jr., 2019, Guidelines for determining flood flow frequency—Bulletin 17C (ver. 1.1, May 2019): U.S. Geological Survey Techniques and Methods, book 4, chap. B5, 148 p., https://doi.org/10.3133/tm4B5.

Fan, Y., & Van Den Dool, H. (2004). Climate Prediction Center global monthly soil moisture data set at 0.5 resolution for 1948 to present. Journal of Geophysical Research: Atmospheres, 109(D10).

Farquharson, F. A. K., Meigh, J. R., & Sutcliffe, J. V. (1992). Regional flood frequency analysis in arid and semi-arid areas. Journal of Hydrology, 138(3-4), 487-501.

Fatichi, S., Ivanov, V. Y., & Caporali, E. (2012). Investigating interannual variability of precipitation at the global scale: Is there a connection with seasonality?. Journal of climate, 25(16), 5512-5523.

Fischer, S., Schumann, A., & Schulte, M. (2016). Characterisation of seasonal flood types according to timescales in mixed probability distributions. Journal of Hydrology, 539, 38-56.

Gaál, L., J. Szolgay, S. Kohnová, J. Parajka, R. Merz, A. Viglione and G. Blöschl (2012) Flood timescales: Understanding the interplay of climate and catchment processes through comparative hydrology, Water Resources Research, 48, W04511, doi:10.1029/2011WR011509.

Gaál, L., Szolgay, J., Kohnová, S., Hlavčová, K., Parajka, J., Viglione, A., ... & Blöschl, G. (2015). Dependence between flood peaks and volumes: a case study on climate and hydrological controls. Hydrological Sciences Journal, 60(6), 968-984.

Griffis, V. W., & Stedinger, J. R. (2007). Log-Pearson type 3 distribution and its application in flood frequency analysis. I: Distribution characteristics. Journal of Hydrologic Engineering, 12(5), 482-491.

Griffis, V. W., & Stedinger, J. R. (2009). Log-Pearson Type 3 distribution and its application in flood frequency analysis. III: Sample skew and weighted skew estimators. Journal of Hydrologic Engineering, 14(2), 121-130.

Hall, J., Arheimer, B., Aronica, G. T., Bilibashi, A., Boháč, M., Bonacci, O., ... & Blöschl, G. (2015). A European Flood Database: facilitating comprehensive flood research beyond administrative boundaries. Proceedings of the International Association of Hydrological Sciences, 370, 89-95.

Hall, J. and G. Blöschl (2018) Spatial patterns and characteristics of flood seasonality in Europe, Hydrology and Earth System Sciences, 22, pp. 3883-3901, https://doi.org/10.5194/hess-22-3883-2018

Hayashi, F., (2000), Econometrics, Princeton University Press.

Hecht, J.S., R.M. Vogel (2020), Updating urban design floods for changes in central tendency and variability using regression, doi:10.1016/j.advwatres.2019.103484

Helsel, D.R., Hirsch, R.M., Ryberg, K.R., Archfield, S.A., and Gilroy, E.J., 2020, Statistical methods in water resources: U.S. Geological Survey Techniques and Methods, book 4, chapter A3, 458 p., https://doi.org/10.3133/tm4a3. [Supersedes USGS Techniques of Water-Resources Investigations, book 4, chapter A3, version 1.1.]

Hofstätter M., Lexer A., Homan M. and G. Blöschl (2018) Large-scale heavy precipitation over central Europe and the role of atmospheric cyclone track types. International Journal of Climatology, 38, pp. e497–e517, https://doi.org/10.1002/joc.5386

Hosking, J. R. M., & Wallis, J. R. (2005). Regional frequency analysis: an approach based on L-moments. Cambridge university press.

Kemter, M., Merz, B., Marwan, N., Vorogushyn, S., & Blöschl, G. (2020). Joint trends in flood magnitudes and spatial extents across Europe. Geophysical Research Letters, 47(7), e2020GL087464.

Laaha, G., & Blöschl, G. (2006). A comparison of low flow regionalisation methods—catchment grouping. Journal of Hydrology, 323(1-4), 193-214.

Lilly, A., Boorman, D. B., & Hollis, J. M. (1998). The development of a hydrological classification of UK soils and the inherent scale changes. In Soil and Water Quality at Different Scales (pp. 299-302). Springer, Dordrecht.

Matalas, N.C., A. Sankarasubramanian (2003), Effect of persistence on trend detection via regression, Water Resour. Res., doi:10.1029/2003WR002292.

Merz R. and G. Blöschl (2003) A process typology of regional floods. Water Resources Research, 39 (12), article number 1340.

Merz, R., & Blöschl, G. (2005). Flood frequency regionalisation—spatial proximity vs. catchment attributes. Journal of Hydrology, 302(1-4), 283-306.

Merz, R., & Blöschl, G. (2009). Process controls on the statistical flood moments-a data based analysis. Hydrological Processes: An International Journal, 23(5), 675-696.

Mimikou, M., & Gordios, J. (1989). Predicting the mean annual flood and flood quantiles for ungauged catchments in Greece. Hydrological sciences journal, 34(2), 169-184.

Pallard, B., A. Castellarin, and A. Montanari (2009), A look at the links between drainage density and flood statistics, Hydrol. Earth Syst. Sci., 13, 1019-1029, doi:10.5194/hess-13-1019-2009

Penna, D., Tromp-van Meerveld, H. J., Gobbi, A., Borga, M., & Dalla Fontana, G. (2011). The influence of soil moisture on threshold runoff generation processes in an alpine headwater catchment. Hydrology and Earth System Sciences, 15(3), 689-702.

Rogger, M., H. Pirkl, A. Viglione, J. Komma, B. Kohl, R. Kirnbauer, R. Merz, and G. Blöschl (2012), Step changes in the flood frequency curve: Process controls, Water Resour. Res., 48, W05544, doi:10.1029/2011WR011187.

Rogger, M., M. Agnoletti, A. Alaoui, J.C. Bathurst., G. Bodner, M. Borga, V. Chaplot, F. Gallart, G. Glatzel, J. Hall, J. Holden, L. Holko, R. Horn, A. Kiss, S. Kohnova, G. Leitinger, B. Lennartz, J. Parajka, R. Perdigão, S. Peth, L. Plavcová, J.N. Quinton, M. Robinson, J.L. Salinas, A. Santoro, J. Szolgay, S. Tron, J.J.H. van den Akker, A. Viglione and G. Blöschl (2017) Land-use change impacts on floods at the catchment scale: Challenges and opportunities for future research. Water Resources Research, 53, 5209–5219, doi:10.1002/2017WR020723.

Rosbjerg, D., G. Blöschl, D. H. Burn, A. Castellarin, B. Croke, G. DiBaldassarre, V. Iacobellis, T. R. Kjeldsen, G. Kuczera, R. Merz, A. Montanari, D. Morris, T. B. M. J. Ouarda, L. Ren, M. Rogger, J. L. Salinas, E. Toth, A. Viglione (2013) Prediction of floods in ungauged basins. Chapter 9 in: G. Blöschl, M. Sivapalan, T. Wagener, A. Viglione, H. Savenije (Eds.) Runoff Prediction in Ungauged Basins - Synthesis across Processes, Places and Scales. Cambridge University Press, Cambridge, UK, pp. 135-162.

Salinas, J.L., G. Laaha, M. Rogger, J. Parajka, A. Viglione, M. Sivapalan and G. Blöschl (2013) Comparative assessment of predictions in ungauged basins; Part 2: Flood and low flow studies. Hydrology and Earth System Sciences, 17, 2637-2652, doi: 10.5194/hess-17-2637-2013

Salinas, J.L., A. Castellarin, A. Viglione, S. Kohnova, and T.R. Kjeldsen (2014), Regional parent flood frequency distributions in Europe – part 1: is the Gev model suitable as a pan-European parent? Hydrol. Earth Syst. Sci., 18, 4381-4389, doi: 10.5194/hess-18-4381-2014

Serago, J.M., R.M. Vogel (2018), Parsimonious nonstationary flood frequency analysis, Adv. Water Resour., 112, 1-16, doi: 10.1016/j.advwatres.2017.11.026

Smith, J. A. (1992). Representation of basin scale in flood peak distributions. Water Resources Research, 28(11), 2993-2999.

Strupczewski, W., Z. Kaczmarek (2001), Non-stationary approach to at-site flood frequency modelling II. Weighted least squares estimation, J. Hydrol., 248(1-4), 143-151, doi: 10.1016/S0022-1694(01)00398-5

Tarasova, L., Merz, R., Kiss, A., Basso, S., Blöschl, G., Merz, B., ... & Wietzke, L. (2019). Causative classification of river flood events. Wiley Interdisciplinary Reviews: Water, 6(4), e1353.

Viglione, A., R. Merz and G. Blöschl (2009) On the role of the runoff coefficient in the mapping of rainfall to flood return periods, Hydrology and Earth System Sciences, 13 (5) 577 - 593.

Viglione, A., Chirico, G. B., Woods, R., & Blöschl, G. (2010a). Generalised synthesis of space–time variability in flood response: An analytical framework. Journal of Hydrology, 394(1-2), 198-212.

Viglione, A., Chirico, G. B., Komma, J., Woods, R., Borga, M., & Blöschl, G. (2010b). Quantifying space-time dynamics of flood event types. Journal of Hydrology, 394(1-2), 213-229.

Viglione, A., Merz, R., Salinas, J. L., & Blöschl, G. (2013). Flood frequency hydrology: 3. A Bayesian analysis. Water Resources Research, 49(2), 675-692.

Wallis, J.R., N.C. Matalas, J.R. Slack (1974) Just a moment! Water Resour. Res., doi: 10.1029/WR010i002p00211

Wang, W., H.-Y. Li, L. R. Leung, W. Yigzaw, J. Zhao, H. Lu, Z. Deng, Y. Demisie and G. Blöschl (2017) Nonlinear filtering effects of reservoirs on flood frequency curves at the regional scale. Water Resources Research, 53, 8277–8292, doi: 10.1002/2017WR020871

Ye, L., X. Gu, D. Wang, and R.M. Vogel. An unbiased estimator of coefficient of variation of streamflow. J. Hydrol., 594, doi: 10.1016/j.jhydrol.2021.125954

---

## Author Response (AR2)

Response to the review comments to the manuscript **hess-2020-600**

**"Characteristics and process controls of statistical flood moments in Europe – a data based analysis"**

by D. Lun, A. Viglione, M. Bertola, J. Komma, J. Parajka, P. Valent and G. Blöschl

We want to thank the editor and the referees for their useful and constructive comments. Here we reproduce the comments of the editor and of all referees in *italic characters*, followed by our answers. The line numbers of the referee comments refer to the line numbers of the revised manuscript, if not stated otherwise.
* * *
**Thomas Kjeldsen (Editor)**

*The reviewers have considered the revised version of the manuscript. While they are generally impressed with the study they have both suggested further, relatively minor, revisions. In particular reviewer #2 have asked for more clarification on aspects raised in the first review. In particular, asked the authors to consider shortening the paper to below 10,000 words. Please consider the comments in detail and submit a revised manuscript for consideration.*

> We thank Thomas Kjeldsen for the useful and constructive comments. We have addressed all comments of the referees below. The main text of the paper, excluding tables, figure captions and the appendix is currently 10,262 words. We have expanded the text following the suggestion of the referees regarding more clarification and it is now 10,800 words, which we believe is close to the target of 10,000.
* * *
**Kolbjorn Engeland (Referee)**

*I think the paper is suitable for publication following some minor clarifications.*

> We thank Kolbjorn Engeland for the valuable comments on the second revision of the manuscript that helped improve the quality of the manuscript. All his comments are reproduced and addressed in the following paragraphs, the line numbers refer to line numbers in the revised manuscript with tracked changes accepted.

*1: Are peak floods or daily floods used in the analysis? The sentence on line 12 makes it unclear: 'Annual maximum discharges were derived from instantaneous peak flows and daily mean flows for each calendar year.'*

> About 25% of the 2,370 flood peak series are instantaneous annual peaks and the rest are annual maxima of daily mean flows, depending on data availability. Given that the average catchment size is about 2,500km$^2$, we consider the effect of this inhomogeneity small relative to the spatial contrasts of floods in Europe. For example, Merz et al. (1999) found that the ratio of annual maxima of instantaneous peaks and daily flows for a catchment size of 2,500km$^2$ is on average 1.2, which is small relative to the spatial contrasts in Europe.

*2: Figure 6: It might help to first show MAF for all sub-regions and then CV for all sub-regions. Then the visual interpretation of the heat map is easier.*

We changed the ordering of the columns of Figure 6, as suggested.

*3: Lines 665-670 I would be careful to make a direct link to the paper by Wang et al., 2017 since you in the current paper uses data from catchments that are only limited influenced by reservoir operations. What you analyse is the effect of natural lakes, whereas Wang et al (2017) consider the influence of reservoirs. Reservoirs introduce much more non-linearity than natural lakes (reservoirs often introduces thresholds in the system response when the dam is overtopped, whereas for lakes, there is a much more gradual transition). In addition, LUW does not account for the location of the lake in the catchment and does not directly tell how large part of the catchment runoff that has to flow through the lakes.*

We fully agree with this assessment, and have therefore modified the sentence to emphasize that reservoirs and natural lakes tend to have different response characteristics.

"The former is consistent with retention effects while the relationship between CV and water body size may be non-linear (increasing CV up to a water body threshold and decreasing CV beyond as shown by Wang et al., 2017 for reservoir effects) which is not captured by Spearman correlation. However, in comparing natural lakes and reservoirs it should be noted that reservoirs tend to introduce more non-linearity in flood frequency behaviour because of a threshold effect when the spillway is activated."

*4: Lines 685 : You could add one or two sentences suggesting non-linear approaches that could be used, e.g. generalized additive models GAM (Rahman et al, 2018, Umlauf & Kneib, 2018) and Random forest (e.g. Desai eta al, 2021) that*

We have added a sentence on possible non-linear modelling procedures, as suggested by the referee.

"While here we examined monotonic relationships and linear relationships, it would also be worth exploring non-monotonic relationships between flood moments and covariates (see e.g. Blöschl and Sivapalan, 1997; Smith, 1992; Pallard et al., 2009). Possible approaches for modelling non-monotonic relationships include generalized additive models (Rahman et al., 2018, Umlauf and Kneib, 2018) and Random forest regression (Desai et al., 2021)."

**References**

Blöschl, G. and M. Sivapalan (1997) Process controls on regional flood frequency: Coefficient of variation and basin scale. Water Resources Research, 33 (12), pp. 2967-2980.

Desai, S., Ouarda, T.B.M.J. (2021) Regional hydrological frequency analysis at ungauged sites with random forest regression, Journal of Hydrology, 594, https://doi.org/10.1016/j.jhydrol.2020.125861.

Merz, R., G. Blöschl und U. Piock-Ellena (1999) Zur Anwendbarkeit des Gradex-Verfahrens in Österreich (Applicability of the Gradex-Method in Austria). Österreichische Wasser- und Abfallwirtschaft, 51, (11/12), pp. 291-305.

Pallard, B., Castellarin, A., and Montanari, A.: A look at the links between drainage density and flood statistics, Hydrol. Earth Syst. Sci., 13, 1019–1029, https://doi.org/10.5194/hess-13-1019-2009, 2009.

Rahman, A., Charron, C., Ouarda, T.B.M.J. et al. Development of regional flood frequency analysis techniques using generalized additive models for Australia. Stoch Environ Res Risk Assess 32, 123–139 (2018). https://doi-org.ezproxy.uio.no/10.1007/s00477-017-1384-1

Smith, J. A. (1992). Representation of basin scale in flood peak distributions. Water Resources Research, 28(11), 2993-2999.

Umlauf, N. and Kneib, T. (2018) A primer on Bayesian distributional regression, Statistical modelling, 18(3.4): 219-247

Wang, W., H.-Y. Li, L. R. Leung, W. Yigzaw, J. Zhao, H. Lu, Z. Deng, Y. Demisie and G. Blöschl (2017) Nonlinear filtering effects of reservoirs on flood frequency curves at the regional scale. Water Resources Research, 53, 8277–8292, doi: 10.1002/2017WR020871
* * *
**Anonymous Referee #2**

*The authors have improved the article and incorporated some suggestions from reviewers. However, further improvements described below are needed before this article can be accepted for publication. I have also attached a Tracked Changes document to this round of revisions for writing suggestions and minor technical comments. I've incorporated my feedback on the authors' responses to my initial comments below, including some places where I agreed with their responses.*

> We thank the anonymous referee for the valuable comments on the second revision of the manuscript. All writing suggestions have been adopted. The technical comments are summarised and addressed in a table at the end of this document.

*Effects of nonstationarity on flood series moments ---------------------------------------------------*

*The authors described nonstationarity as being outside the scope of their paper in their response to my initial review. While I agree that it is not the goal of their paper, I still think a quick investigation as to whether trends might affect estimates of sample moments is imperative, especially given that snowmelt comprises a major control on flood generation in 3/5 regions they examine.*

*This could be limited to an analysis of the effects of trends in the MAF on estimates of the CV following the conditional moments framework of Serago and Vogel (2018). If a trend is not accounted for, the CV can be overestimated since the overall variance of the peak flows will also include the variance explained by the trend.*

*The effects of trends in the mean and variance on CS can be mathematically derived but given general estimation challenges with at-site skewness arising from sampling variability, it seems like a less essential endeavor for this study.*

*I strongly recommend that any choice to retain the assumption of stationarity for any region should be supported with at-site trend analyses of these site records as well as any prior literature, including studies that examine trends over periods of record of more than 50 years*

*(to avoid confounding apparent trends with artifacts of inter-decadal variability) and studies specifically focused on snow trends.*

*Finally, I agree with the authors that the autocorrelation of the annual flood series is often weak and, consequently, that adjusting significance inferences for persistence is of second-order importance for their continental-scale investigation. However, it should be noted very briefly to avoid any inappropriate uptake of this work.*

We agree that the presence of trends is a relevant aspect in the analysis of flood moments. Trends of exactly the same data set have already been analysed and published previously (Blöschl, Hall et al., 2019).

We prefer not to include a trend analysis and/or an analysis of the moments of the residuals to a trend, as this would completely change the focus of the paper. Our approach is pragmatic in the sense that we are interested in the statistical flood moments for the period 1960-2010. We use the standard product moment estimators to make the results comparable with a large body of literature.

We now do acknowledge in the discussion section the existence of trends in the data and the possible effect on floods moments. We also state that flood moments could be investigated by using a framework including trends in some of the moments, such as the one suggested by Serago and Vogel (2018).

"A possible extension of the analysis presented here could be the consideration of non-stationarities in flood moments, for example in the spirit of Serago of Vogel (2018). Blöschl, Hall et al. (2019) have found that significant trends do exist in the mean flood of the data set in 28.02% of the stations. Trends affect the estimation of flood moments. For example the detrended data tend to exhibit smaller CVs than the raw data, while the effect on the sample mean may be smaller."

We believe we already refer to the issue regarding autocorrelation in line 194.

*Sample moment estimation biases under stationarity ----------------------------------------------*

*While the authors point out some good reasons for not pursuing further efforts to correct common product moment estimators for bias, I think that the authors should pay a little more attention to this. In their revised manuscript, the authors write "while the estimation uncertainty of the mean is small, the uncertainty and bias of the estimators of CV and CS (equations 3 and 4) can be substantial. Ye et al. (2020) illustrate the uncertainty and bias in the estimation of CV". This does not provide readers with an idea of the magnitude of this bias nor a sense of how to determine it should they be concerned about it for a practical application.*

*The authors are correct in observing that the specific methods for bias correcting CV estimates that Ye et al. (2020) employ (recommended in prior round of review) assume distributions other than the GEV, a distribution whose prominence in many parts of Europe has been previously established. However, Ye et al. (2020) provide these methods as examples and make it clear that the bias of the common product common CV estimator is not specific to any theoretical probability distribution.*

*Ye et al. (2020) cite the following general relation between the bias of the product moment estimator of the CV and the population CV, population CS and record length from Breunig (2011):*

*$Bias(CV\_est) = CV\_true^{(3/2)}/N * [3*sqrt(CV\_true) – 2*CS\_true]$*

*Indeed, this equation is difficult to apply for CV bias correction without knowing the true value of the CV unless Monte Carlo experiments requiring distribution assumptions are simulated.*

*Yet, it is possible to use this equation to assess the general magnitude of CV estimation bias by examining ranges of plausible values of the true CV and true CS based on a priori knowledge of sites in a region.*

*For instance, using the 75% values of the estimated CV (0.61) and CS (1.69) as true values, one obtains the following correction factor for a 50-year peak-flow series:*

*(0.61)^(3/2)/50*[3*sqrt(0.61) – 2(1.69)] = -0.009 = -0.9%*

*For the 25% estimated CS (0.62), this rises to just 1.1%. Unfortunately, I cannot compute the bias over the full range of estimated CV and CS values (since only 25%/50%/75% are reported in Table 1). The authors may also want to expand this range given that CS and CV informing this range are estimated values, not true ones.*

*However, it could end up that the bias in the CV is relatively minor for the range of CV and CS in the study. In this case, the authors could state that after testing plausible CV_true and CS_true values reflecting the range of sites in their study, the adjustments to the CV general did not exceed a low percentage (e.g. 10%), therefore making it reasonable to use common product moment estimators of the CV for the sake of comparing their work with the body of literature that uses this estimator.*

*It would also be nice to mention that future work should involve the generation of GEV-based bias correction factors using an approach similar to the one that Ye et al. (2020) undertook with the lognormal, kappa, and Wakeby distributions.*

*With regards to skewness estimation bias, the authors could explore using the GEV-based bias correction method from Carney (2016), although this is really a second-order issue given the pronounced effects that sampling variability can have on skewness coefficient estimates.*

In order to inform readers about the magnitude of the bias in the estimation of CV in more detail, we have modified the text in section 2.3 in the following way, using the equations suggested by the referee:

"While the estimation uncertainty of the mean is small,  the uncertainty and bias of the estimators of CV and CS (equations 3 and 4) can be substantial. Ye et al. (2020) illustrate the uncertainty and bias in the estimation of CV. The bias in the estimation of CV is relatively small for ranges of CV and CS as in this study (using their equation 2: the bias is at most 0.065 in absolute value, in the case of CV ranging from 0.25 to 0.97 and CS ranging from 0.09 to 3.18, which encompasses 90% of the observed values in this study) making it reasonable to use the common product moment estimator of the CV. "

Regarding skewness, we agree with the referee that the sampling variability is very pronounced for records as short as in the present study. We believe we inform the readers about the estimation uncertainty and bias in the estimation of CS and prefer not to pursue this issue in more detail, as it is not at the heart of the manuscript. Carney (2016) investigates bias in the estimation of parameters of the GEV-distribution via the method of L-moments, whereas this paper does not fit a GEV distribution and focusses on product moments. Extrapolating their results to a bias-correction for the estimation of skewness would require additional attention.

*OLS regression model assumptions ------------------------------------------------------------------*

*In their response to initial feedback, the authors are correct in stating that regression coefficient estimates are unbiased even when the assumptions of normality and heteroscedasticity are violated. I viewed the need for these assumptions to be evaluated as requisite for making hypothesis testing-based inferences using p-values and other standard*

*error-based criteria. However, if the goal is to understand the range of coefficient magnitudes without making hypothesis-oriented inferences, then ignoring these assumption evaluations is less critical. However, the authors should make an explicit statement if this is a scope limitation that they would like to establish. If the take this approach, it is yet another reason for them to apply an all subsets modeling strategy in lieu of the stepwise one that they reported, which presumably uses a statistical significance-based criterion in adding and removing variables from the multivariate regression models (see below).*

*I appreciate the information in the appendices, and the importance of including the variance inflation factors to prevent excessive multicollinearity among explanatory variables as well.*

> We have now added a sentence stating, that we do not look at significance tests for coefficient estimates:

> "The OLS-estimator still remains unbiased and consistent under these conditions (Hayashi, 2000), but no inferences such as significance tests of individual coefficients should be made from standard properties of the OLS-estimator. In Tables A.1 and A.2 we report the standard errors of the coefficient estimators, which should be interpreted with care and are thus not used for hypothesis tests. "

> The stepwise selection is based on an information criterion (Mallow's Cp) and aims at a good predictive performance respectively a good model fit, but does not rely on a significance-based criterion. We therefore choose to retain the stepwise procedure to derive meaningful covariates for the regional regression models. We also performed additional analyses comparing the results of the stepwise selection procedure and an all subsets modelling strategy, as suggested by the referee. Both procedures select the same variables for the regional regression models for MAF and CV for the data in the present study.

*Model building and variable selection --------------------------------------------------------------------*

*The authors used a stepwise [forward] selection process to build their multivariate regression models. The leaps R package has an all subsets routine that they could use to check if they missed any strongly performing models by using a stepwise selection process, which does not evaluate all possible combinations of explanatory/predictor variables.*

> We now performed additional analysis using the R-package 'leaps', as suggested by the referee. The best performing models for MAF and CV were the same for the stepwise selection and the all subset selection using the leaps-package and Mallow's Cp as the selection criterion.

*Temperature as a proxy for snowmelt -------------------------------------------------------------------*

*The authors include two temperature variables (min winter temp and min spring temp) as proxies for snowmelt impacts on annual peak flows. Negative coefficients on their relationship with peak flows assume that colder winters and springs lead to greater snowmelt contributions to flooding. However, how well correlated are winter/spring temperatures with both seasonal snowpack and the rate at which it melts?*

*At a minimum, the authors should describe some constraints regarding the collection of consistent snowpack depth and snow cover data in Europe as well as limiting assumptions of their use as proxies and efforts to circumvent them. Other examples of regional regression studies using temperature variables as snowmelt proxies would also be helpful.*

We now include the following sentence in the discussion, pointing out the limited information content of temperatures as proxies for snowmelt.

"Mean spring and winter temperature were used in the analysis to capture snow processes because of the better data availability. Chaoimh (1998) and Bednorz (2003) identified correlations between spring and winter temperature with snowpack-depth and days with snow-cover and more generally air temperature is often used as an indicator of snowmelt (Ohmura, 2001). Future work could enrich the analysis by using snow data directly, although remote sensing products may have some limitations related to the duration (see e.g. Parajka and Blöschl, 2012). "

*Soil moisture data biases* --------------------------------------------------------------------------------

*The authors write that "Soil moisture (SM) was taken from the CPC Soil moisture database, which contains model-calculated soil moisture values. Fan and Van Den Dool (2004) discuss some biases of the soil moisture data set, which may distort some of the findings here." However, the authors do not comment further on any of these biases/distortions and their implications regarding inferences on process controls of annual peak flows.*

The discussion of the soil moisture biases of Fan and Van Den Dool (2004) is rather vague. For example they state: "The results show that the Climate Prediction Center (CPC) global soil moisture data, in spite of its simplicity, simulates the seasonal to interannual variability of observed soil moisture reasonably well in many places." It is therefore difficult to be more explicit about the potential biases and their effects on the results in this paper.

*Seasonality analysis not well integrated into manuscript narrative* --------------------------------

*The seasonality analysis is interesting its own right, but it could be better integrated into the manuscript. A phrase or sentence in the abstract should mention this purpose considering the amount of text devoted to this component of the study.*

We have modified the following sentence in the abstract adding information about the the purpose of the seasonality analysis:

"The process controls on the flood moments in five predetermined hydroclimatic regions are identified through correlation and multiple linear regression analyses with a range of covariates and the interpretation is aided by a seasonality analysis."

*Writing* ------------------------------------------------------------------------------------------------

*The authors should aim to reduce the length of the article to roughly 10,000 words by combining sentences, getting rid of unnecessary phrases and wordy language and possibly reducing the discussion of CS given the challenges that sample variability poses to at-site skewness estimates. See the Tracked Changes document for additional writing suggestions.*

We are especially grateful for the specific writing suggestions. They are all adopted in the revised manuscript. The main text of the paper, excluding tables, figure captions and the appendix is currently 10,262 words. We have expanded the text following the suggestion of the referees regarding more clarification and it is now 10,800 words, which we believe is close to the target of 10,000.

*A minor comment on reproducibility ------------------------------------------------------------------*

*I still think it is valuable from a reproducibility perspective to identify the 22 catchments you omitted from your study due to insufficient covariate data. This does not have to be overemphasized, as you could add a quick list to your supplemental material or data repository.*

> We have added a table to the supplementary material indicating the row number of the series in the data of the supplemental material from Blöschl, Hall et al. (2019), which were omitted from the regional regression models.

Table 1: Row-indices of omitted catchments for regional regression models. The row-indices refer to the supplementary material of Blöschl, Hall et al. (2019). Only catchments that were used in their Figure 1 are used in this paper.

| |
|---|
| 221 |
| 320 |
| 415 |
| 416 |
| 566 |
| 601 |
| 630 |
| 632 |
| 969 |
| 2154 |
| 2232 |
| 3382 |
| 3456 |
| 3467 |
| 3468 |
| 3511 |
| 3570 |
| 3593 |
| 3610 |
| 3620 |
| 3640 |
| 3679 |

**References**

Bednorz, E. (2004). Snow cover in eastern Europe in relation to temperature, precipitation and circulation. International Journal of Climatology: A Journal of the Royal Meteorological Society, 24(5), 591-601.

Carney, M.C. (2016), Bias correction to GEV shape parameters used to predict precipitation extremes, doi: 10.1061/(ASCE)HE.1943-5584.0001416

Chaoimh, Ú. N. (1998). European snow cover and its influence on spring and summer temperatures. Geographical Journal, 41-54.

Hayashi, F., (2000), Econometrics, Princeton University Press.

Ohmura, A. (2001). Physical basis for the temperature-based melt-index method. Journal of applied Meteorology, 40(4), 753-761.

Parajka, J., & Blöschl, G. (2012). MODIS-based snow cover products, validation, and hydrologic applications. Multiscale Hydrologic Remote Sensing: Perspectives and Applications, edited by: Chang, N.-B. and Hong, Y.

Serago, J. M., & Vogel, R. M. (2018). Parsimonious nonstationary flood frequency analysis. Advances in Water Resources, 112, 1-16.

Ye, L., Gu, X., Wang, D., & Vogel, R. M. (2021). An unbiased estimator of coefficient of variation of streamflow. Journal of Hydrology, 594, 125954.

**Technical comments**

First column is the number of comment, second column is the line in the annotated word-file provided by the referee, third is the comment of the referee, fourth is the response, fifth are the text changes

|  |  | **Review comments** | **Response** | **Text change** |
|---|---|---|---|---|
| A1 | 11 | Consider using introductory sentences like this to make the scope and contribution of your work clear from the get-go | Thank you for this suggestion. | Text adopted. |
| A2 | 35 | Add something about how this can support future work? | Future work is supported by providing a baseline for local studies, as mentioned in the text. | |
| A3 | 35 | Abstract doesn't say anything about seasonality analysis. | The abstract now includes the seasonality analysis. | The process controls on the flood moments in five predetermined hydroclimatic regions are identified through correlation and multiple linear regression analyses with a range of covariates and the interpretation is aided by a seasonality analysis. |
| A4 | 60 | More studies than this. | We have added two references. | While USGS regional flood-frequency studies based on observed data have revealed non-climatic controls (Parrett et al., 2011, Paretti et al., 2014, England et al., 2019), most |

| | | | | knowledge on these effects comes from process-based simulation studies. |
|---|---|---|---|---|
| A5 | 62 | Do you mean process-based simulation based? | We refer to process-based simulation studies. | Sentence adopted. |
| A6 | 66 | Including nonstationary ones | Suggestion added. | The role of these variables, can to some extent be inferred from their use as covariates in flood frequency regionalization models (see, e.g. Zaman et al., 2012; Rosbjerg et al., 2013; Miller and Brewer, 2018 ), including nonstationary ones. |
| A7 | 74 | This co-evolution doesn't always favor flood generation though. For instance, a wet year could increase vegetation, which could increase transpiration during the following year, reducing runoff generation. | Thank you for pointing this out, we agree and modified the sentence. | Moreover, climate, vegetation, soils and land forms may co-evolve with MAP, thus exerting a longer-term influence which may increase or decrease floods (Gaál et al., 2012, Perdigão and Blöschl, 2014). |
| A8 | 75 | Of annual peak flows? | Yes, clarification added. | Farquharson et al. (1992) found CV of annual peak flows (variability between years) to increase with the Aridity Index (the ratio of potential evaporation and MAP). |
| A9 | 79 | Clarify this earlier | Clarification moved to | |

| | | | | previous sentence of paragraph. | |
|---|---|---|---|---|---|
| A10 | 83 | Pallard drainage area studies: https://hess.copernicus.org/articles/13/1019/2009/ | We are discussing here a slightly different point, i.e. CV as a function of area rather than drainage density. | |
| A11 | 86 | What about the infiltration excess mechanism causes this decrease with area? What about the increase in CV with area in basins where saturation excess overflow dominates? Adding a sentence or so to explain these mechanisms would be helpful. | The authors are not clear about the physical process controlling the scaling of CV. We have therefore chosen to remove this sentence | |
| A12 | 120 | Check to see if this is referenced later on in paper. | This is not referenced later in the paper. | |
| A13 | 120 | See General Comment | We address this in the general comment regarding temperature as a proxy for snowmelt. | |
| A14 | 123 | How biased is it? Discuss this in the discussion? | See our answer to the comment regarding Soil moisture data biases. | |
| A15 | 150 | So mixed rainfall and snowmelt? | Yes, added clarification. | The Central-Eastern region has a continental climate with cold winters and warm summers and floods mainly occur in spring with snow-melt contributions (resulting in a mixture of rainfall and snowmelt). |

| A16 | Table1 | Consider creating three columns for these values | We adopted the suggestion and modified the table accordingly. | |
|---|---|---|---|---|
| A17 | Table1 | Evapotranspiration? | Yes, thank you. | Changed to evapotranspiration |
| A18 | Table1 | Note this article in your discussion: https://www.researchgate.net/publication/352322291_A_note_on_some_uncertainties_associated_with_Thornthwaite's_aridity_index_introduced_by_using_different_potential_evapotranspiration_methods | While this is interesting in its own right, we feel this would go beyond the scope of the paper. | |
| A19 | 161 | See general comment about adjustments for nonstationarity | We address this in the general comment regarding nonstationarity. | |
| A20 | 174 | What do they find? How big of an issue is it? | This is also addressed in the response to the referee's comment regarding the sample moment estimation biases under stationarity. A More detailed calculation is now presented in the manuscript. | While the estimation uncertainty of the mean is small, the uncertainty and bias of the estimators of CV and CS (equations 3 and 4) can be substantial. Ye et al. (2020) illustrate the uncertainty and bias in the estimation of CV. The bias in the estimation of CV is relatively small for ranges of CV and CS as in this study (using their equation 2: the bias is at most 0.065 in absolute value, in the case of CV ranging from 0.25 to 0.97 and CS ranging from 0.09 to 3.18, which corresponds to roughly 90% of |

| | | | | observed values in this study). |
|---|---|---|---|---|
| A21 | 175 | Please describe. | We have clarified the text. | Based on a simulation study. |
| A22 | 183 | See General Comment. Consider stating this at the beginning of the paragraph before getting into estimation biases. | We have a slight preference for leaving this statement where it is. | |
| A23 | 191 | Any model performance stats to report briefly? | The model performance stats are reported in Table A.6. | |
| A24 | 192 | Consider enumerating each of these steps | We are now enumerating the steps of our analysis. | |
| A25 | 210 | Interesting analysis in its own right, but you should describe how you used these results to inform your interpretations of process controls | We added a sentence clarifying the use of the results from the seasonality analysis in the paper. | In the spirit of Blöschl et al. (2017) we used the seasonality of floods to identify dominant flood-generating mechanisms, e.g. spring snowmelt vs winter storms which to some extent explain variations in the flood moments (Merz and Blöschl, 2003). |
| A26 | 223 | What about all possible subsets approach? | We performed additional analyses and the all possible subsets approach yields exactly the same selection of covariates for all regional regression models for MAF and CV. | |
| A27 | 247 | Did you also use Cs as a response variable? | No, this is stated at the beginning of section 3.5. | |

| A28 | 248 | Does this tie in with your analysis of process controls? It seems like it could be a second paper if you validate your ordinary kriging model. | No process controls are used for the ordinary kriging model. It merely serves as comparison for predictions of the regional regression models. | |
|-----|-----|------|------|------|
| A29 | 254 | Worth showing other descriptive stats of these values in a table? For instance, sd, min, max? | We believe that the quantiles (25%/50%/75%) give a comprehensive picture of the characteristics of the distribution and the moments and the extremes are perhaps not needed. | |
| A30 | 261 | How large is the MAF? | We added the relevant information. | On the other hand, the Northeastern region has the smallest average CV and CS and below average MAF (0.39, 0.82 and 0.13 respectively). |
| A31 | 285 | Consider the inherent relationship between these two here:

CS = (X-M)^3/S^3 while CV = S/M

If S goes up, \|CS\| goes down and CV goes up

Can you discuss why CS and CV are positively correlated across sites despite the expected relationship between CS and CV at a single site described above? | Yes, a detailed answer is given below the table. | |
| A32 | Fig 4 | Consider moving to supplement | We have a total of 10 Figures which we do not consider an excessive number, so | |

| | | | | perhaps moving Figure 4 to the supplement is not needed. | |
|---|---|---|---|---|---|
| A33 | 382 | Check for repetition | | We checked for repetitions of this statement and they have been removed in the previous version of the document. | |
| A34 | 540 | Steeper slopes? Smaller watersheds? | | Suggestion adopted. | Further inland, various mountain ranges (Pyrenees, Massif Central, Alps Apennines, Ore mountains, Carpathians, Balkan mountains) stand out with higher MAF than the surrounding areas (mostly above 0.3 m³/s/km²) and summer as the dominant flood season due to their effects of enhancing rainfall and probably shallower soils as well as steeper slopes and smaller watersheds. |
| A35 | 541 | Slovenia is close to the Mediterranean Sea (Adriatic Sea) while the Ore mountains are on the German-Czech border. How do highly variable Mediterranean storm tracks drive this pattern? | | This is explained in more detail in Hofstätter et al. (2018). Flood-generating storms, such as Vb events tend to have preferred pathways (see e.g. Figure 5 and 6 in Hofstätter et al., 2018), which | |

| | | | both affect Slovenia and the Ore mountains. These tracks often extend over more than 1,000 km. | |
|---|---|---|---|---|
| A36 | 552 | Are you just assuming that they have more nonlinear runoff generation processes because they lie in a region with less mean annual and extreme precip? To support this speculation, could you please add a citation about this general tendency in Europe? Or better, could you cite any work demonstrating nonlinear runoff generation processes in this specific region? | We now cite papers discussing the non-linearity of runoff generation in arid and semi-arid regions of Europe, including Ukraine and Hungary. | Some of the continental regions of Europe (Hungary, Poland, Ukraine) are particularly sheltered by mountain chains, resulting in low precipitation, both at the annual scale and for extreme events, which translates into low MAF and mostly high CV due to the more non-linear runoff generation as compared to wetter regions (Nováaky, 1991, Didovets et al., 2017, Ries et al., 2017). |
| A37 | 578 | Evidence? | Blöschl and Sivapalan (1997) discuss the process controls on CV. The non-linearity of runoff generation is generally related to the soil moisture status and the reasoning is that during snowmelt floods the soil tends to be wetter than for other types | They |

| | | | | |
|---|---|---|---|---|
| | | | of floods (Grillakis et al., 2016). The reference has been added to the paper. | |
| A38 | 583 | How spatially transferrable is this finding? Should recognize need for more research on this given that this study was conducted in just one location. | We have included the referee's suggestion in the manuscript. | This may be related to possible non-monotonous relationships between CV and area as suggested by Smith (1992), and more complex aggregation effects (Blöschl and Sivapalan 1997), although more research is needed on the transferability of this finding. |
| A39 | 607 | Can you support this sentence with some of your findings? | We now support this sentence by referring to Figure 6. | While regional studies have suggested that MAP is a better predictor of MAF than other precipitation variables (Mimikou and Gordios, 1989; Merz and Blöschl, 2009) this does not seem to be the case at the European scale (see e.g. Figure 6). |
| A40 | 609 | Since MAP is a better indicator of soil moisture than P95 and Pmax? | Yes, we added the referee's sentence for clarification. | On the other hand, CV is always a better correlated with MAP than with P95 and Pmax, reflecting the decreasing degree with which antecedent soil moisture is captured as one moves from MAP |

| | | | | to P95 and Pmax, since MAP better captures soil moisture conditions. |
|---|---|---|---|---|
| A41 | 613 | Explain better | We further clarified our statement. | This effect is also represented in the negative correlations between CS and MAP (r=-0.35) and CS and P95 (r=-0.34) (Table A.1.5) in the Mediterranean, indicating a decrease in skewness for comparatively wetter catchments, which is related to a particularly large potential for this contrast in initial conditions. |
| A42 | 615 | Why? | Given that AI also contains MAP, we decided on P95 as a representative variable for precipitation characteristics to avoid this overlap. | |
| A43 | 616 | See this note: https://www.researchgate.net/publication/352322291_ A_note_on_some_uncertainties_associated_with_ Thornthwaite's_aridity_index_introduced_by_ using_different_potential_evapotranspiration_ methods | We feel a discussion of the estimation procedures of the aridity index would distract from the results of the paper, in particular given that we are not including a similar discussion for the other | |

| | | | | variables. Within the levels of correlations obtained and the accuracy of the discharge measurements the effect of different approaches to estimating AI is probably minor. | |
|---|---|---|---|---|---|
| A44 | 623 | Both winter and spring air temperatures? | | We added more information. | As would be expected, the Spearman correlations between temperature and $MAF_\alpha$ and CV are comparatively high in the Northeastern and Central-Eastern region (higher for spring temperature), where snow-processes are important for floods. |
| A45 | 628 | What about the spatial scale of snowmelt vs. rain events? | | The spatial scale of snowmelt floods is usually larger than that of rain-floods, but according to our opinion this aspect does not tie in closely with the argument. | |
| A46 | 635 | Whereas? | | Suggestion adopted. | |
| A47 | 638 | General comment: variables that explain spatial variability might not be best for explaining temporal variability | | While we fully agree with this comment, there might be some symmetry which is worth looking into (Perdigão | |

| | | | | |
|---|---|---|---|---|
| | | | and Blöschl, 2014). | |
| A48 | 706 | Didn't you use pre-determined regions from another study? | The regions are also used in Lun et al. (2020). | |
| A49 | 725 | Citation regarding the ability to capture these processes? | This is discussed in the cited literature i.e. section 3.4 in Boorman et al. (1995), Lilly et al. (1998) and section 3 in Maréchal and Holman (2005). | |
| A50 | 733 | Your study also excluded sites with pronounced anthropogenic impacts, including these | We excluded sites with heavy urbanization, but not those with deforestation/ afforestation. We have added a comment to explain. | On the other hand: flood changes of small local streams may be much more controlled by land use changes, such as urban development and deforestation (Rogger et al., 2017), only a few of which are included in this study (average catchment size of 2,480 km$^2$). |

*A31: Consider the inherent relationship between these two here:*

*CS = (X-M)^3/S^3 while CV = S/M*

*If S goes up, |CS| goes down and CV goes up*

*Can you discuss why CS and CV are positively correlated across sites despite the expected relationship between CS and CV at a single site described above?*

> The referee is correct in pointing out that in the above equations a ceteris paribus increase in S would result in a decrease of |CS| and an increase of CV. However, from this fact we cannot infer the correlation between these estimators. For small samples exact formulas for the correlation between these two estimators are hard to derive. Instead we use a limit theorem and check its validity for a sample size of 50, which is representative of our data.

> For the asymptotic correlation between these two estimators, we use a multivariate central limit theorem for the estimators of the first three non-centralized moments. Using

$$\mu_n' = \mathbb{E}[X^n]; \; \mu_n = \mathbb{E}[(X - \mu_1')^n]; \; \tilde{\mu}_n = \frac{\mathbb{E}[(X - \mu_1')^n]}{(\mathbb{E}[(X - \mu_1')^2])^{n/2}}; CV = \frac{\mu_2^{1/2}}{\mu_1'}$$

$$m_n' = \frac{1}{n}\sum_{i=1}^{n} X_i^n; \; m_n = \frac{1}{n}\sum_{i=1}^{n}(X_i - \bar{X})^n; \; cv = \frac{\sqrt{m_2}}{m_1'}; \; g = \frac{m_3}{m_2^{3/2}}$$

According to a multivariate CLT (assuming iid-observations and 6-th order moments exist, see e.g. example 2.18 in Vaart, 2000)

$$\sqrt{n}\begin{pmatrix} m_1' - \mu_1' \\ m_2' - \mu_2' \\ m_3' - \mu_3' \end{pmatrix} \overset{d}{\Rightarrow} N(0, \Sigma)$$

$$\Sigma = \begin{pmatrix} \mu_2' - \mu_1'^2 & \mu_3' - \mu_1'\mu_2' & \mu_4' - \mu_1'\mu_3' \\ \mu_3' - \mu_1'\mu_2' & \mu_4' - \mu_2'^2 & \mu_5' - \mu_2'\mu_3' \\ \mu_4' - \mu_1'\mu_3' & \mu_5' - \mu_2'\mu_3' & \mu_6' - \mu_3'^2 \end{pmatrix}$$

Here $\overset{d}{\Rightarrow}$ refers to convergence in distribution. By using the delta method (e.g. Theorem 3.1 in Vaart, 2000) we can obtain the following limit theorem (we omit the calculation steps)

$$\sqrt{n}\begin{pmatrix} m_1' - \mu_1' \\ cv - \mu_2^{1/2}/\mu_1' \\ g - \mu_3/\mu_2^{3/2} \end{pmatrix} \to N(0, \Sigma^*)$$

$$\Sigma^* = \begin{pmatrix} \mu_2 & \frac{\mu_2}{\mu_1}\left[\frac{\tilde{\mu}_3}{2} - CV\right] & \mu_2^{1/2}\left[\tilde{\mu}_4 - \frac{3}{2}\tilde{\mu}_3^2 - 3\right] \\ \frac{\mu_2}{\mu_1}\left[\frac{\tilde{\mu}_3}{2} - CV\right] & CV^2\left[\frac{\tilde{\mu}_4 - 1}{4} + CV^2 - \tilde{\mu}_3 CV\right] & CV^2\left[-\frac{5\tilde{\mu}_3}{4\,CV} - \frac{3\tilde{\mu}_3\tilde{\mu}_4}{4\,CV} - (\tilde{\mu}_4 - 3) + \binom{3}{2}\tilde{\mu}_3^2 + (1/2)\frac{\tilde{\mu}_5}{CV}\right] \\ \mu_2^{1/2}\left[\tilde{\mu}_4 - \frac{3}{2}\tilde{\mu}_3^2 - 3\right] & CV^2\left[-\frac{5\tilde{\mu}_3}{4\,CV} - \frac{3\tilde{\mu}_3\tilde{\mu}_4}{4\,CV} - (\tilde{\mu}_4 - 3) + \binom{3}{2}\tilde{\mu}_3^2 + (1/2)\frac{\tilde{\mu}_5}{CV}\right] & 9 - 6\tilde{\mu}_4 + \frac{9}{4}\tilde{\mu}_4\tilde{\mu}_3^2 - 3\tilde{\mu}_3\tilde{\mu}_5 + \frac{35}{4}\tilde{\mu}_3^2 + \tilde{\mu}_6 \end{pmatrix}$$

The correlation between the estimators of CV and CS depends on higher-order moments of the underlying distribution. $\tilde{\mu}_n$ refers to standardized moments (skewness, kurtosis, etc.). From this result, which is also documented in Bobee (1973), we can calculate the correlation between the estimators of CV and CS from the asymptotic distribution (simply as $\frac{\Sigma_{2,3}^*}{\sqrt{\Sigma_{2,2}^* \cdot \Sigma_{3,3}^*}}$), which we can use as a proxy for small-sample results.

A quick calculation (higher-order moments of the GEV can be calculated with the formulas from Muraleedharan et al., 2011) reveals that these results predict a correlation of about 0.64 for the average parameter configuration of the study data, assuming a GEV-distribution as the data-generating process. Here we use the parametrization of the GEV as in chapter 18 of Maidment (1993). Using the average moments of the data as the population moments ($\mu_1' = 0.17, CV = 0.52, \tilde{\mu}_3 = 1.28$) the parameters of the GEV correspond to $\xi = 0.13, \alpha = 0.07, \kappa = -0.02$. The higher-order moments correspond to $\tilde{\mu}_4 = 6.13$, $\tilde{\mu}_5 = 24.11$, $\tilde{\mu}_6 = 132.61$. All numbers here are rounded to two digits.

A quick simulation study (100,000 times generating 50 observations from a GEV with the parameters as specified above and calculating the correlation between the estimates

of CV and CS) results in an empirical correlation that is very close to the value predicted above (around 0.66 instead of 0.64 in the simulation). This result indicates that the asymptotic result is suitable for the sample size and parameter configuration considered here.

Considering this, the positive correlation between estimates of CV and CS across sites, assuming a GEV as the data-generating process and considering the average empirical moments of the study data, does seem plausible.

**References**

Bobee, B. (1973). Sample error of T-year events commuted by fitting a Pearson type 3 distribution. Water Resources Research, 9(5), 1264-1270.

Boorman, D. B., Hollis, J. M., & Lilly, A. (1995). Hydrology of soil types: a hydrologically-based classification of the soils of United Kingdom. Institute of Hydrology, Wallingford, UK.

Didovets, I., Lobanova, A., Bronstert, A., Snizhko, S., Maule, C. F., & Krysanova, V. (2017). Assessment of climate change impacts on water resources in three representative Ukrainian catchments using eco-hydrological modelling. Water, 9(3), 204.

England, J.F., Jr., Cohn, T.A., Faber, B.A., Stedinger, J.R., Thomas, W.O., Jr., Veilleux, A.G., Kiang, J.E., and Mason, R.R., Jr., 2019, Guidelines for determining flood flow frequency—Bulletin 17C (ver. 1.1, May 2019): U.S. Geological Survey Techniques and Methods, book 4, chap. B5, 148 p., https://doi.org/10.3133/tm4B5.

Grillakis, M. G., Koutroulis, A. G., Komma, J., Tsanis, I. K., Wagner, W., & Blöschl, G. (2016). Initial soil moisture effects on flash flood generation–A comparison between basins of contrasting hydro-climatic conditions. Journal of Hydrology, 541, 206-217.

Hofstätter M., Lexer A., Homan M. and G. Blöschl (2018) Large-scale heavy precipitation over central Europe and the role of atmospheric cyclone track types. International Journal of Climatology, 38, pp. e497– e517, https://doi.org/10.1002/joc.5386

Lilly, A., Boorman, D. B., & Hollis, J. M. (1998). The development of a hydrological classification of UK soils and the inherent scale changes. In Soil and Water Quality at Different Scales (pp. 299-302). Springer, Dordrecht.

Lun, D., Fischer, S., Viglione, A., & Blöschl, G. (2020). Detecting flood-rich and flood-poor periods in annual peak discharges across Europe. Water Resources Research, 56(7), e2019WR026575.

Maidment, D. R. (1993). Handbook of hydrology (No. 631.587). McGraw-Hill,.

Maréchal, D., & Holman, I. P. (2005). Development and application of a soil classification-based conceptual catchment-scale hydrological model. Journal of Hydrology, 312(1-4), 277-293.

Muraleedharan, G., Soares, C. G., & Lucas, C. (2011). Characteristic and moment generating functions of generalised extreme value distribution (GEV). In Sea Level Rise, Coastal Engineering, Shorelines and Tides (pp. 269-276). Nova.

Nováaky, B. (1991). Climatic effects on the runoff conditions in Hungary. Earth surface processes and landforms, 16(7), 593-599.

Parrett, C., Veilleux, A., Stedinger, J. R., Barth, N. A., Knifong, D. L., & Ferris, J. C. (2011). Regional skew for California, and flood frequency for selected sites in the Sacramento-San Joaquin River Basin, based on data through water year 2006. U. S. Geological Survey.

Paretti, N. V., Kennedy, J. R., Turney, L. A., & Veilleux, A. G. (2014). Methods for estimating magnitude and frequency of floods in Arizona, developed with unregulated and rural peak-flow data through water year 2010 (No. 2014-5211). US Geological Survey.

Perdigão, R. A. P., and G. Blöschl (2014) Spatiotemporal flood sensitivity to annual precipitation: Evidence for landscape-climate coevolution, Water Resour. Res., 50, 5492-5509, doi:10.1002/ 2014WR015365.

Ries, F., Schmidt, S., Sauter, M., & Lange, J. (2017). Controls on runoff generation along a steep climatic gradient in the Eastern Mediterranean. Journal of Hydrology: Regional Studies, 9, 18-33.

Smith, J. A. (1992). Representation of basin scale in flood peak distributions. Water Resources Research, 28(11), 2993-2999.

Van der Vaart, A. W. (2000). Asymptotic statistics (Vol. 3). Cambridge university press.